# Ground-based MAX-DOAS observations of tropospheric formaldehyde VCDs and comparisons with the CAMS model at a rural site near Beijing during APEC 2014

Xin Tian [1,2], Pinhua Xie *[1,2,4], Jin Xu *[2], Yang Wang *[3], Ang Li [2], Fengcheng Wu [2], Zhaokun Hu [2], Cheng Liu [2,4,5,6], Qiong Zhang [2]

1. School of Environmental Science and Optoeclectronic Technology, University of Science and Technology of China, Hefei, 230026,China;
2. Key laboratory of Environmental Optical and Technology, Anhui Institute of optics and Fine Mechanies, Chinese Academy of Science, Hefei, 230031,China;
3. Max Planck Institute for Chemistry, Mainz, 55128, Germany;
4. CAS Center for Excellence in Urban Atmospheric Environment, Institute of Urban Environment, Chinese Academy of Sciences, Xiamen, 361021, China;
5. School of Earth and Space Sciences, University of Science and Technology of China, Hefei, 230026, China;
6. Anhui Province Key Laboratory of Polar Environment and Global Change, USTC, Hefei, 230026, China;

*Correspondence to:* Pinhua Xie (phxie@aiofm.ac.cn); Jin Xu (jxu@aiofm.ac.cn); Yang Wang (y.wang@mpic.de)

**Abstract.** Formaldehyde (HCHO), a key aerosol precursor, plays a significant role in atmospheric photo-oxidation pathways. In this study, HCHO column densities were measured using a Multi-AXis Differential Optical Absorption Spectroscopy (MAX-DOAS) instrument at the University of Chinese Academy of Science (UCAS) in Huairou District, Beijing, which is about 50 km away from the city center. Measurements were taken during the period of October 1, 2014 to December 31, 2014, and the Asia-Pacific Economic Cooperation (APEC) summit was organized on November 5–11. Peak values of HCHO vertical column densities (VCDs) around noon and a good correlation coefficient $R^2$ of 0.73 between HCHO VCDs and surface $O_3$ concentration during noontime indicated that the secondary sources of HCHO through photochemical reactions of volatile organic compounds (VOCs) dominated the HCHO values in the area around UCAS. Dependences of HCHO VCDs on wind fields and backward trajectories were identified and indicated that the HCHO values in the area around UCAS were considerably affected by the transport of pollutants (VOCs) from polluted areas in the south. The effects of control measures on HCHO VCDs during the APEC period were evaluated. During the period of the APEC conference, the average HCHO

VCDs were ~38% ± 20%, and ~30% ± 24% lower than that during the pre-APEC and post-APEC periods calculated at 95% confidence limit, respectively. This phenomenon could be attributed to both the effects of prevailing northwest wind fields during APEC and strict control measures. We also compared the MAX-DOAS results with the Copernicus Atmosphere Monitoring Service (CAMS) model. The HCHO VCDs of the CAMS model and MAX-DOAS were generally consistent with a correlation coefficient $R^2$ greater than 0.68. The peak values were consistently captured by both data datasets, but the low values were systematically underestimated by the CAMS model. This finding may indicate that the CAMS model can adequately simulate the effects of the transport and the secondary sources of HCHO, but underestimates the local primary sources.

## 1 Introduction

The 2014 Asia-Pacific Economic Cooperation (APEC) conference was held in the Huairou District of Beijing from November 5–11, 2014. To improve the air quality in Beijing during the APEC conference, a group of collaborators on atmospheric pollution prevention and control in the Beijing-Tianjin-Hebei (Jing-Jin-Ji) region and surrounding areas compiled the "The APEC conference air quality assurance policy" (Liu et al., 2015). Some provinces, including Beijing, Tianjin, Hebei, Shanxi, Inner Mongolia, and Shandong implemented different emission reduction strategies in accordance with the air quality assurance plan (Wang et al., 2016a). Since November 1, 2014, parts of the Jing-Jin-Ji region and surrounding areas had begun to implement an emission reduction plan according to the APEC conference air quality assurance policy. Formal emission reduction measures were implemented in the Jing-Jin-Ji region and surrounding areas from November 3 and included limiting the production of factories, shutting down construction sites, implementing traffic restrictions based on even- and odd- numbered license plates, and improving road cleaning procedures (Wang et al., 2016a). In response to the possible adverse weather conditions from November 8–10, the "enhanced emission reduction measures" were implemented in the Jing-Jin-Ji region and surrounding areas from November 6. These various efforts coupled with relatively favorable weather conditions compared with previous years resulted in the emission reduction measures having significant effects. Based on estimations, all types of main pollutants were reduced by over 40% in Beijing and by over 30% in other provinces, through these measures (Wang et al., 2016a). From November 1–12, 2014, the air quality was at an

excellent level, referred to as "APEC blue" reference.

Presently, many studies have analyzed the effects of emission reduction measures during the APEC summit. Ground-based observations were taken to investigate the air quality changes associated with a series of stringent emission-reduction measures (Fan et al., 2016; Li et al., 2016; Liu et al., 2016; Tang et al., 2015; Chen et al., 2015; Wang et al., 2016a; Wang et al., 2016b; Wang et al., 2017a). Wang et al, (2016a) selected five representative *in situ* stations in different locations in Beijing, which were Miyun Reservoir Station (City Background Station), Yuzhan Station (Regional Station), Changping Station (Suburban Station), Olympic Sports Center Station (City Station) and Xizhimen North Street Station (Transport Station), and found that average concentrations of $SO_2$, $NO_2$, $PM_{10}$, and $PM_{2.5}$ decreased by 62%, 41%, etc, respectively, whereas the average $O_3$ level approximately doubled over this period than the same period over the last five years ($PM_{2.5}$ since 2013). $O_3$ production rate depends on the ratios of volatile organic carbon (VOC) and NOx. The urban and suburban areas of Beijing are controlled by the NOx saturated condition of $O_3$ production. Since emission control measures are mainly focus on NOx, but not VOCs, decrease of NOx can cause significant increases of $O_3$ concentration (Wang et al., 2016a). Although the traffic and urban stations produce a lot of pollution due to motor vehicle emissions, the $NO_2$ concentrations of suburban and regional stations significantly dropped (47%) compared with the traffic and urban stations (23%) as a result of the control measures. The $NO_2$ emitted by motor vehicles in the Beijing urban area remained high even under the measures taken to limit the number of vehicles (Wang et al., 2016a). Space observations were also used to evaluate the effect of emission control measures on the changes in $NO_2$ tropospheric vertical column densities (VCDs) and aerosol optical depth (AOD) in Beijing and its surroundings based on the Ozone Monitoring Instrument (OMI) and Moderate Resolution Imaging Spectroradiometer (MODIS) retrieval. The results showed that $NO_2$ VCD and AOD were mostly reduced by 47% and 34% in Beijing, respectively (Huang et al., 2015; Wei et al., 2016; Meng et al., 2015). The analytical results of the Chemical Mass Balance (CMB) model showed that the contributions of coal-fired boilers, dust, and motor vehicles to $PM_{2.5}$ in Beijing were around 2%, 7%, and 30%, respectively, during the APEC summit (Cheng et al., 2016). Zhang et al (2017) analyzed the characteristics of aerosol size distribution and the vertical backscattering coefficient profile during the 2014 APEC summit using lidar observation. Particles with larger sizes were better controlled during the APEC period, with the number concentration of accumulation mode and coarse mode particles experiencing more significant decreases of 47% and 68%

than before and after the APEC period (Zhang et al., 2017). Published studies have focused mainly on the effects of commonly measured gas pollutants, particulate matter, and aerosols, but not HCHO (Cheng et al., 2016; Fan et al., 2016; Huang et al., 2015; Li et al., 2016; Liu et al., 2016; Meng et al., 2015; Tang et al., 2015; Chen et al., 2015; Wang et al., 2016a; Wang et al., 2016b; Wang et al., 2017a; Wei et al., 2016).

As an abundant product of the oxidation of many volatile organic compounds (VOCs), HCHO is known to harm human health for instance, by damaging oral epithelial cells (Nilsson et al., 1998; Pinardi et al., 2013). A variety of other hydrocarbons generally determine the concentration of HCHO. Thus, HCHO is used as an indicator of VOCs (Fried et al., 2011). Tropospheric formaldehyde mainly originates from two sources. The primary emissions emanate from incomplete combustion, such as anthropogenic (e.g., industrial emissions) and pyrogenic (mainly biomass burning) sources. Secondary

sources originate from the photo-oxidation process of many VOCs. In addition, a small fraction of HCHO originates from the direct emissions of biogenic sources (e.g., vegetation). The variability in HCHO over continents is particularly dominated by the distributions of local emissions of non-methane volatile organic compounds (NMVOCs) (Chance et al., 2000). Being a short lifetime oxidation product, long-living VOCs, such as methane ($CH_4$), contribute to the background levels of HCHO (Pinardi et al., 2013; Stavrakou et al., 2009; Vrekoussis et al., 2010). The monitoring of NMVOC emissions is essential not

only for the hydroxyl radical OH, but also for the formation and transport of secondary organic aerosols (Palmer et al., 2006; Stavrakou et al., 2009; De Smedt et al., 2015). HCHO is an important indicator of atmospheric photochemical reactions. As an active gas, HCHO can be photolyzed to generate $HO_2$ free radicals. $HO_2$ rapidly and radically reacts with NO to generate OH, which can influence the oxidation ability of the atmosphere. All of the photolysis equations of HCHO to form OH radical at wavelengths below 370 nm are listed as follows:

$$HCHO + h\nu \rightarrow H + HCO(\lambda \leq 370nm) \rightarrow H_2 + CO \tag{1}$$

$$H + O_2 \rightarrow HO_2 \tag{2}$$

$$HCO + O_2 \rightarrow HO_2 + CO \tag{3}$$

$$HO_2 + NO \rightarrow OH + NO_2 \tag{4}$$

Therefore, HCHO can reflect anthropogenic VOC emissions and VOC emissions through the fast production of short-lived

NMVOCs. Identifying the major sources of HCHO is essential for quantifying the photolysis sources of OH and their

contributions to aerosol formation and for effectively controlling photochemical pollution (Bauwens, et al., 2016; Chang, et al., 2016; Ling, et al., 2017; Ma, et al., 2016; Tanaka, et al., 2016).

The implementation of a series of temporary reduction measures for atmospheric pollutant emissions in major international events in China is relatively rare. The APEC summit provided an opportunity to study the relationship between environmental concentrations and pollutant emissions (Cheng et al., 2016). Studying the influences of control measures on HCHO is crucial for improving air quality.

A type of passive differential optical absorption spectroscopy system, called Multi-AXis Differential Optical Absorption Spectroscopy (MAX-DOAS), has been used over the past decade to measure tropospheric trace gases (Honninger et al., 2004; Wagner et al., 2004; Sinreich et al., 2005; Wagner et al., 2007; Vigouroux wt al., 2009). The information obtained from MAX-DOAS measurements includes tropospheric column densities, surface mixing ratios, and vertical profiles of aerosol extinction and trace gas mixing ratios. HCHO can be measured in the ultraviolet spectral range using the MAX-DOAS technique (Vigouroux et al., 2009; Wagner et al., 2011; Pinardi et al., 2012, 2013; Li et al., 2013; Borovski et al., 2014; Cheung et al., 2014; Franco et al., 2015; Lee et al., 2015; Schreier et al., 2016; Wang et al., 2017b; Wang et al., 2017c).

In this study, we used the ground-based MAX-DOAS instrument installed in the Huairou District (suburban area) of Beijing to evaluate the effects of the sources and depositions of HCHO and their relations with emission control measures and meteorological conditions during the period from October 26, 2014 to November 20, 2014. Two pollution episodes and their relationships with meteorological conditions were analyzed during APEC to evaluate the effects of regional transport and local emissions. Afterwards, three episodes, defined as "pre-APEC," the period of APEC and "post-APEC," were used to evaluate the influences of emission control measures on the changes in HCHO VCD during APEC. The correlations between HCHO VCDs with $NO_2$ VCDs and $O_3$ were used to determine the main HCHO sources and evaluate the dominant error sources of HCHO simulations of Copernicus Atmosphere Monitoring Service (CAMS) model. Finally, the HCHO VCDs retrieved from the MAX-DOAS measurements were then compared with the results from the CAMS model simulations from October 1, 2014 to December 31, 2014. Consistencies and discrepancies between the model and measurements are discussed. This study can be used as a reference for evaluating the effectiveness of photochemical pollution control measures adopted during the APEC summit. Moreover, this study could practically assist in the development of control strategies in the future

and also provides support for verifying the model simulations.

## 2 Experiment and Methodology

### 2.1 Monitoring locations and instrument

To evaluate the emission control measures, a supersite was established in the Yanqi Lake campus of the University of Chinese Academy of Sciences (UCAS) in Huairou District, northeast suburban area of Beijing. The APEC conference was also held in the Yanqi Lake region near the campus of UCAS (**Fig. 1**). The field campaign was performed for nearly four months from October 1, 2014 to January 20, 2015. However, this study only discusses the period from October 2014 to December 2014.

The Yanshan Mountains are in the west of the site, and the Yanqi Lake lies at the edge of the mountains to southwest of the site. Beijing urban areas and the industrial cities of Tangshan, Baoding, Shijiazhuang and Tianjin are also located in the south. Relevant pollution sources in Tangshan, Baoding, Shijiazhuang, and Tianjin are primary pollution hotspots. In Beijing, vehicles are the predominant pollution source, especially in the urban areas (Lin et al., 2009; Lin et al., 2012; Shao et al., 2006; Tang et al., 2015; Wei et al., 2016). Thus, air flow coming from the south may bring anthropogenic emissions. The site is mainly influenced by emissions from vehicles on China National Highway 111 that runs from the north and south as well as some stationary sources from the rural settlements across the highway (Zhang et al., 2017).

The MAX-DOAS instrument was deployed on the balcony (without a roof) of a classroom on the $4^{th}$ floor in the laboratory building in the campus of UCAS (116.67 °E, 40.4 °N). The UCAS supersite is on the top floor of the laboratory building, which is about 10 m away from the MAX-DOAS instrument. Nitrogen oxide (NO, $NO_2$, and NOx) was measured by chemiluminescence (Thermo Scientific, Model 42i), and ozone ($O_3$) was measured by UV photometry (Thermo Scientific, Model 49i). These gas analyzers had precision values of 0.5 ppb and 0.4 ppb, respectively.

**Fig. 2** shows a structural representation of the MAX-DOAS system. This system comprises a telescope, stepper motor, spectrometer, and computer. Sunlight is focused by the telescope, which is installed outdoors and reaches the spectrometer through an optical fiber. The spectrometer was placed in a temperature-controlled box at 20 °C to ensure that the spectrograph could work at a stable temperature under the changing ambient temperature from -15 °C to 30 °C in China. The

spectrometer was produced by Ocean Optics and was named Maya (https://oceanoptics.com/product/maya2000-pro-custom/).
The spectrometer covers the range of 290 nm to 420 nm, and its instrumental function is approximated as a Gaussian
function with a full width at half maximum (FWHM) of 0.5 nm. MAX-DOAS was routinely operated for 24 h. Due to the
intensity of the sunlight, only the daytime measurements were used for analysis. The nighttime measurements could be used
5    to correct the dark current and offset. The azimuth angle view of the telescope was fixed at 0 °(North) during the entire
observation period. A full MAX-DOAS scan comprises six elevation angles (EAs) (3 °, 5 °, 10 °etc) and lasts for
approximately 10 min (see **Fig. 3**). Each measurement had an average of 100 scans, and the integration time was adjusted
automatically based on the light intensity. **Table 1** lists the detailed setup of the MAX-DOAS instrument.
Meteorological parameters, including wind speed (WS), wind direction (WD), temperature (T), and relative humidity (RH),
10   were continuously measured by a MetPak automatic weather station (Gill Instruments Ltd, Lymington, UK) at the UCAS
superstation from October 28, 2014 to December 31, 2014 (**Fig. 4a**). All of the measured meteorological parameters were
recorded at 1-min time intervals. During the campaign, there were clear diurnal variations in maximum temperature at noon
and minimum temperature at night. The temperature was within a range of -10.6 ℃ to 20.7 ℃, with a sudden drop to below
℃ on December 1, 2014. The wind rose indicated that the prevailing wind direction was from the northwest (**Fig. 4b**).
Halfacre et al (2014) defines the relatively calm conditions with wind speeds of less than 3.5 m s$^{-1}$ as the static weather
situation. The static weather situation frequently occurred during the observations, while wind speeds of more than 3.5 m s$^{-1}$
usually appeared under northwest and west winds.

**2.2 DOAS spectral retrieval and determination of tropospheric VCD**

MAX-DOAS, which is an optical remote-sensing technology that records the spectra of scattered sunlight at different
elevation angles, can be used to quantitatively measure trace gases based on the Beer-Lambert Law (H önninger and Platt,
2002; Bobrowski et al., 2013; Roozendael et al., 2003; Trebs et al., 2004; H önninger et al., 2004; Wagner et al., 2004). The
spectra obtained from the MAX-DOAS observations were analyzed using WINDOAS software (Hermans et al., 2003). The
fitting range was from 335 nm to 360 nm. The gas cross-sections of HCHO at 293K (Meller and Moortgat, 2000), BrO at
223 K (Fleischmann and Hartmann, 2004), NO$_2$ at 294K (Vandaele et al., 1996), O$_3$ at 223 K and 243K (Serdyuchenko et al.,

2013), and $O_4$ at 293 K (Thalman and Volkamer, 2013) were included in the fit. A spectrum at the 90 °EA recorded at 12:09 local time (LT) on November 6, 2014 was used as the Fraunhofer reference spectrum (FRS) for all the retrievals to determine slant column densities (SCD). This day was a very clear day with very low pollution and was also during the period where strict pollution control measures were in place. The Ring structure (Fish and Jones, 2013), which is used to account for rotational Raman scattering effects, was calculated using DOASIS software (Kraus, 2006) based on the FRS and was included in the fit. **Table 2** lists the parameter settings used for the HCHO analysis. We excluded data for solar zenith angle (SZAs) larger than 75 °because of the stronger absorptions of stratospheric species and a low signal-to-noise ratio. Data with a large root mean square (RMS) of the residuals ($>10^{-2}$) and large relative intensity offset were also excluded.

**Fig. 5** shows an example of the DOAS spectrum analysis in evaluating the HCHO SCD at 12:30 on November 19, 2014. The red and blue curves indicate the fitted absorption structures and the derived absorption structures from the measured spectra, respectively. HCHO SCD was $7.21 \times 10^{16}$ molecules cm$^{-2}$ with an error of $8.14 \times 10^{15}$ molecules cm$^{-2}$. The root mean square of the optical depth of the residual spectral structures was $1.08 \times 10^{-3}$.

The geometric approximation was used to convert the dSCD to the tropospheric VCD. In the first step, the differential slant column densities (dSCDs) were derived from the DOAS spectral analysis with a so-called FRS (and measured in a small sun zenith angle at 90 °elevation around noon) (Hermans et al., 2003; Hönninger and Platt, 2002; Kraus, 2006). The SCD includes two parts of the absorption signal of the troposphere and stratosphere. To remove the interference of stratosphere absorption and variation in instrumental properties, dSCD at off-zenith elevation angles were subtracted by the dSCD at 90 ° elevation angle in the same elevation sequence to derive ΔSCD following the equation below:

$$\Delta SCD_\alpha = dSCD_{\alpha \neq 90°} - dSCD_{\alpha = 90°}. \tag{5}$$

VCD is defined as the integrated concentration of trace gas concentration through the atmosphere along a vertical path and is calculated from dSCD by the use of air mass factor (AMF) as follows:

$$VCD = \frac{dSCD_{\alpha \neq 90°} - dSCD_{\alpha = 90°}}{AMF_{\alpha \neq 90°} - AMF_{\alpha = 90°}} = \frac{\Delta SCD}{\Delta AMF}. \tag{6}$$

AMF is often used to describe the absorption path of a gas in the atmosphere. Brinksma et al (2008) proposed the geometric approximation method to calculate AMF:

$$\text{AMF}(\alpha) = 1/\sin\alpha .$$ (7)

Then the tropospheric VCD can be obtained from the following equation:

$$\text{VCD} = \frac{\Delta\text{SCD}}{\dfrac{1}{\sin(\alpha)} - 1} .$$ (8)

Numerous studies have compared the error between geometrical VCD and VCD from profile inversion (Brinksma et al.,

2008; Hendrick et al., 2014; Hönninger et al., 2004; Wang et al., 2017b). Wang et al (2017b) showed that the geometric

approximations are usually underestimated by 10% in comparison to the profile inversions for HCHO VCDs at 20° elevation,

but the error is larger for larger elevation angles and larger relative azimuth angle (RAA). This study used the geometric

approximation method to determine HCHO VCDs at an elevation angle of 15°. The geometric light paths at 15° and 30° are

good approximations in the boundary layer. However lower systematic errors were achieved at 15° than at 30° by using the

geometrical approximation (discussed in Section 2.3 below). Additionally, the geometric approximation method is more

stable and less influenced by clouds than the profile inversion method (Hönninger et al., 2004; Clémer et al., 2010; Wagner

et al., 2009; Wagner et al., 2011; Erle et al., 2013).

**2.3 Error budgets**

The following error sources were considered as the error estimates for the MAX-DOAS results:

a. The systematic error of the HCHO VCDs calculated by the geometric approximation depends on the layer height of the

trace gases and aerosols. To evaluate the systematic error of the geometric approximation, we calculated more exact

tropospheric HCHO $\text{VCD}_{\text{AMF}}$ using the PriAM inversion algorithm (Wang et al, 2017b). HCHO VCDgeo at elevation angles

at 15° and 30° are obtained from the geometric approximation. The relative differences (Diff) between $\text{VCD}_{\text{AMF}}$ and

VCDgeo for HCHO were calculated by Eq. (9):

$$Diff = \frac{VCD_{geo} - VCD_{AMF}}{VCD_{AMF}}$$ (9)

In Fig. 6, the average relative differences for elevation angles of 15° and 30° are shown as a function of the effective cloud

fractions (eCF), as 0<eCF ≤1, 0<eCF≤0.3, 0.3<eCF ≤0.7, and 0.7<eCF≤1.0. The cloud fractions (eCF) are downloaded

from the ECMWF CAMS model. It can be seen that the biases caused by the use of the geometric approximation are

generally much smaller at EA=15 °than at EA=30 °, with the Diff being mostly smaller than 6% for the 15 °elevation angle

of and smaller than 16% for the 30 °elevation angle in all periods. The bias for Diff caused by using the geometric

approximation is about 2%.

b. The fitting error of the DOAS fit is derived from the dSCD fitting error to VCD error by using geometric approximation,

as

$$VCD_{fittingerror} = \frac{VCD_{error}}{VCD} = \frac{\sqrt{2(dSCD_{\alpha \neq 90° \, fittingerror}^2 + dSCD_{\alpha = 90° \, fittingerror}^2)}}{2(\frac{1}{\sin \alpha} - 1) \times VCD}, \tag{10}$$

and the hourly average of the HCHO VCD fitting error was from 4% to 27% for the entire period with the absolute fit error

is ~$1.2 \times 10^{15}$.

c. Cross section error also constitutes one of the error sources. Some previous research reported that cross section errors of

$O_4$ (aerosols) and HCHO are 5% and 9%, respectively (Bogumil et al., 2003; Meller and Moortgat, 2000;Thalman and

Volkamer, 2013; Vandaele et al. 1998 ). Wang et al (2017b) estimated the errors related to the temperature dependence of

the cross sections, and the corresponding systematic error of HCHO was estimated to up to 6%.

Since the three errors are mainly independent, the total error can be calculated by combining all the above error sources,

adding up to about 7% - 28% with 17% on average.

## 2.4 ECMWF CAMS model

The European Centre for Medium-Range Weather Forecasting (ECMWF) is at the forefront of research for numerical

weather prediction including probabilistic forecasting. Copernicus Atmosphere Monitoring Service (CAMS), which is

managed by ECMWF, publicly provides generally reliable atmospheric information. The CAMS model was established

utilizing the wealth of Earth observation data from satellite and ground-based systems. The CAMS model produces real-time

analyses and forecasts of atmospheric composition for the global view for each day (Persson and Grazzini, 2001). CAMS

real-time products can be freely downloaded via a platform (http://apps.ecmwf.int/datasets/data/cams-nrealtime/levtype=sfc/) (Anders, 2015). The operational CAMS uses fully integrated chemistry in the Composition Integrated Forecasting System (C-IFS). C-IFS is a new global chemistry model for the forecasting and assimilation of atmospheric composition. In the simulation of HCHO, chemistry originating from the Transport Model 5 (TM5) had been fully integrated into the C-IFS, in which only gas phase reactions of HCHO are included. The actual emission totals in the T255 simulation for 2008 from anthropogenic, biogenic sources, and biomass burning were used in C-IFS (Flemming et al., 2014). All of the analyzed parameters acquired from the CAMS model are at 00:00, 06:00, 12:00, and 18:00 UTC. We used the CAMS model data with grid resolutions of 0.125 $°\times$ 0.125 $°$, 0.25 $°\times$ 0.25 $°$, and 0. 5 $°\times$ 0. 5 $°$ at 8:00 LT (00:00 UTC) and 14:00 LT (06:00 UTC).

## 3 Results and discussion

We evaluate the impact of emission control policy on air quality during APEC based on the MAX-DOAS measurements from the period of October 26, 2014 to November 20, 2014. As the measurement station is at a rural site situated about 50 km away from the Beijing downtown area, wind fields, which are transporters of pollutants, were considered in the evaluation. Additionally, MAX-DOAS and the CAMS model were compared to verify the model data in the period of October 1, 2014 to December 31, 2014. Since the daytime period is relatively short in autumn, the available MAX-DOAS measurement time was from 06:30 to 18:30 LT. The effects of different cloud coefficients on MAX-DOAS inversion VCDs and the HCHO VCDs from MAX-DOAS and the CAMS model under different cloud coefficients were compared. The results show that the cloud coefficient had a negligible influence on the retrieval of HCHO VCDs by MAX-DOAS. Additionally, sunny and cloudless weather generally occurred during the entire APEC period. Thus, all of the data obtained in the different cloud coefficients were used.

### 3.1 Effects of pollutant transport

**Fig. 7** shows the time series of HCHO VCDs measured by MAX-DOAS and meteorological parameters in the period of 3 to 8 November, 2014. Two peak values (November 4, 2014 and November 7, 2014) were observed during APEC (**Fig. 7a and 7e**). Relative humidity, solar radiation intensity, and ambient temperature are important photo-oxidation factors (Starn et al.,

1998; Solberg et al., 2001; Zhang et al., 2009). The daily averaged intensity of solar radiation and temperature on November 4 and 7 were compared with data from the two periods of November 4 to 7 and October 1 to December 31, 2014 (**Fig. 8**). The differences in averaged solar radiation and temperature on November 4 and 7 compared to the period from October 1 to December 31, 2014, were 2.2%, -23.5% etc, respectively. Solar radiation and temperature did not change significantly

during the two peaks periods of November 4, 2014 and 7, 2014; thus, the enhanced HCHO values were not be caused by an increased photo-oxidation rate. The increase in HCHO was probably related to a change in wind direction to the south. **Fig. 7** indicates the increases in wind speed, and a dominant southerly wind flow with a speed of more than 2.0 m s$^{-1}$, during the two days was detected. South air flow was dominant during the two days of peak HCHO values during the APEC summit. At noon on November 3, 4, and 7, the wind direction changed to south and the wind speed was relatively strong (approximately

4.0 m s$^{-1}$), thereby leading to the rapid accumulation of HCHO in a few hours. Thereafter, the wind direction changed to northwest and the wind speed dropped below 2.0 m s$^{-1}$ during the nighttime along with a gradual dissipation in pollution, which indicated that the contamination of the supersite was affected by the pollution transported from the southwest areas. The value on November 6, 2014 was probably caused by the good dispersion conditions under the northwest winds with speeds of more than 3.5 m s$^{-1}$, with the air mass mainly originating from the clean northwest area.

To further demonstrate the effect of regional transport, we analyzed 24-h backward trajectories of air mass using the National Oceanic and Atmospheric Administration Hybrid Single Particle Lagrangian Integrated Trajectory (http://ready.arl.noaa.gov/HYSPLIT.php) on November 4 to 7, 2014 (**Fig. 9**). We used 8:00 LT (UTC 0:00) as the start time of the backward trajectories. As shown in **Fig. 9**, pollutants over the polluted region, including Shijiazhuang and Baoding, in the southwest of Beijing, were transported to the observed area on November 4, 2014. The observed area was mainly

affected by the pollution in Tangshan and Langfang located in the south and southeast of Beijing on November 7, 2014. Under the dominant northwest wind (November 5 to 6, 2014), the concentrations of HCHO were significantly lower than that on November 4 and 7, 2014, and reached the minimum under a high wind speed of up to 7.0 m s$^{-1}$. Changes in dominant wind fields play important roles in the changes in HCHO at UCAS. Regional transport from the south had a significant impact on the increase in HCHO observed at the site.

We further analyzed the relationship between HCHO VCDs and wind direction and speed from October 28, 2014 to

November 20, 2014 (**Fig. 10**). Both local emission and transport from remote sources impacted the observed HCHO VCDs. The main wind directions of the UCAS site were east, south, and northwest with approximate percentages of 17%, 16%, and 12%, respectively. The amount of HCHO VCDs was strongly dependent on the wind speed and direction **(Fig. 10).** HCHO VCDs considerably depend on wind directions, and the average HCHO VCDs were $7.6 \times 10^{15}$ molecules cm$^{-2}$ under the southerly wind (including southwest and southeast). This finding was attributed to the fact that Tangshan, Baoding, Shijiazhuang and Tianjin, and other heavily polluted cities are located in the south of UCAS, including the city center of Beijing (Lin et al., 2009; Lin et al., 2012; Shao et al., 2006; Tang et al., 2015; Wei et al., 2016). In contrast, the northeast and north directions correspond to a minimum in the average HCHO VCDs ($6.6 \times 10^{15}$ molec cm$^{-2}$). The northern cities are clean with low VOC emissions, and the nature sources of VOC in the north should be much lower than the anthropogenic sources in the winter season. Thus few precursors of HCHO were transported to the measurement station in the north wind. The lower HCHO VCDs under such conditions are mainly due to fewer VOC precursors of HCHO. In summary, the wind from this area prominently contributes to the dispersion of the pollutants. In terms of the dependence of HCHO on wind speed, the HCHO VCDs decrease along with the increasing wind speed under the northerly fast and clean wind, which results in the rapid dissipation of the pollution. Under the southerly wind, the HCHO VCDs increase with increasing wind speed. Thus, transport from the south polluted air to the observation site occurs more easily under southerly winds with relatively high wind speeds. In conclusion, when winds are from the south, the site was considerably affected by the transport of pollutants from the polluted urban areas, including urban superimposed emissions from the Beijing city center, Baoding, Shijiazhuang, Tianjin, and Langfang.

## 3.2 Evaluation of HCHO during APEC

The MAX-DOAS results in the period of October 26, 2014 to November 20, 2014 were used to evaluate the HCHO values during APEC. The period was split into three episodes. The first episode was defined as the period of APEC (from November 1–12, 2014), wherein strict air quality policies were implemented at a regional scale. The second and third episodes were defined as the "pre-APEC" period from October 26–31, 2014 and the "post-APEC" period from November 13, 2014 to November 20, 2014.

**Fig. 11** presents the varying series of the daily mean values of HCHO VCDs during APEC in the three episodes. The result shows a "fluctuating effect" with the HCHO VCDs increasing abruptly over several days and dropping sharply for a few days during the APEC summit. This phenomenon was also observed in $NO_2$ VCD observations in Beijing urban areas (Liu et al., 2016). The average HCHO VCDs were $9.7 \times 10^{15}$, $6.0 \times 10^{15}$, and $8.6 \times 10^{15}$ molec $cm^{-2}$ before, during, and after APEC, with fitting errors of 9.4%, 10.1%, and 9.7%, respectively. A noticeable decrease of ~38% $\pm$ 20%, and ~30% $\pm$ 24% during APEC was found compared with before and after APEC, which was calculated at 95% confidence limit. This reduction could be attributed to the control measurements implemented during APEC. However, the systematic difference in wind fields between the three episodes could also play a role considering the effects of the transport of pollutants discussed in Sect. 3.1. The wind rose for wind speed at the three episodes is shown in **Fig. 12**. The prevailing wind direction of the three episodes was from the northwest. The frequency of northwest winds in the APEC period was more than in the pre-APEC and post-APEC periods. Furthermore, the wind speed in the APEC period was higher. Under the prevailing northwest wind, the transport of pollutants from the polluted south area to the observed area was much less than that under the southerly winds. Therefore, the prevailing northwest wind fields may also contribute to the low HCHO during APEC.

As the measurement station is located in the northern suburban area of Beijing, the effects of the control measures, which were mainly implemented in the urban areas, on HCHO were only observed at the station when dominant southerly winds occurred (Fan et al., 2016, Li et al., 2015). We thus plotted the dependence of HCHO VCDs on the wind speed and directions in **Fig. 12d–f** for the pre-APEC, APEC, and post-APEC periods. **Fig. 12d–f** indicate that the averaged HCHO VCDs under south winds during APEC were about $6.5 \times 10^{15}$ molec $cm^{-2}$, which was considerably lower than 10.3, or $9.2 \times 10^{15}$ molec $cm^{-2}$ in the pre-APEC and post-APEC periods. In addition the peak values due to transport from the south urban area on November 4, 2014 and November 7, 2014 during APEC shown in **Fig. 11** were 25% and 18% lower than the peak values under similar wind fields in the pre-APEC and post-APEC periods. In general, the HCHO values under the dominant southerly wind field were considerably lower during APEC than the pre-APEC and post-APEC periods. The phenomenon implies that the control measures had a certain effect on reducing the concentration of HCHO. This suggests that the implementation of control measures during the APEC summit reduced the concentrations of $NO_2$ and aerosols (Liu et al., 2016; Zhang et al., 2017).

### 3.3 Sources of HCHO

The hourly averaged HCHO VCDs in the three episodes were analyzed to characterize the diurnal variation (**Fig. 13**). The hourly averaged HCHO VCDs exhibited evident daily variation. High values appeared in the early afternoon and low values appeared in the morning. Atmospheric photochemical reactions are related to the intensity of solar radiation as indicated in Equation 1. The atmospheric photochemistry reaction is generally active when the intensity of the solar radiation is strong. Therefore, HCHO productivity of secondary sources is high (Anderson et al., 1996). Peaks in diurnal variation generally emerged at 14:00, which is likely related to the diurnal variations in photochemical reaction rates. Since most light is available in the early afternoon and local direct emissions are relatively smaller compared to secondary production, the secondary production of formaldehyde primarily caused the peak at 14:00. The diurnal variation in VOC emissions could also play a role in the diurnal variation of HCHO. However, the typical life time of VOCs can reach several days. The diurnal variations in VOC emission are unlikely to change the abundance of atmospheric VOCs. Therefore, diurnal variation in photo reaction rate could be the dominant driving factor. Other smaller peaks appeared in the evening during other period of busy traffic (16:00–18:00 LT), which might be caused by primary pollution sources, e.g., exhaust fumes from vehicles. Thus, the diurnal variations in HCHO during all three episodes were similar to the typical patterns of secondary sources as reported in Anderson, et al (1996), Lee, et al (2015), Pinardi, et al (2013).The absolute HCHO values during the APEC period were statistically lower than those in the pre-APEC and post-APEC periods.

Determining pollution sources is crucial to controlling air pollution. Three time intervals were used for determining the main HCHO sources. The first interval was defined as noontime from 11:00–14:00 and is associated with strong photochemical reactions. The second and third intervals were defined as the morning rush hour from 7:00–9:00 and the evening rush hour from 16:00–18:00. To further determine whether the pollution sources of HCHO at UCAS were primary or secondary formations from other VOCs, the correlations of HCHO with the primary pollutant $NO_2$ or secondary pollutant $O_3$ were analyzed (Anderson et al, 1996; Possanzini et al., 2002). Surface $O_3$ data were obtained from *in situ* measurements in the UCAS supersite, and troposphere $NO_2$ VCD data were retrieved from the same MAX-DOAS measurements using geometric approximation. The linear correlations of noontime average HCHO VCD with $NO_2$ VCD and $O_3$ from 11:00–14:00 and rush

hour average HCHO VCD with $NO_2$ VCD and $O_3$ from 7:00–9:00 and 16:00–18:00 are shown in **Fig. 14**. Direct analysis of the data indicates that noontime average HCHO had a higher correlation coefficient with $NO_2$ VCD and $O_3$ than rush hour. This implies that a small amount of HCHO comes from the traffic emissions during rush hour. A good correlation coefficient $R^2$ of 0.73 was found between HCHO VCD and $O_3$ during the noontime, which indicates that the main source of HCHO was from secondary photo-oxidation formation at noon. Here it needs to clarify that the $O_3$ data is from the surface measurements, but $NO_2$ and HCHO are the tropospheric VCD. Since $NO_2$ and HCHO are mostly in the boundary layer, the effect of the discrepancy of measured layers on the correlation analysis is not significant. In contrast, a correlation coefficient of 0.38 between HCHO VCD and $NO_2$ VCD during noontime was better than during rush hour ($R^2$=0.06), which may be due to the contribution of vehicle emissions to HCHO precursors. A longer $NO_2$ lifetime with less dispersion efficiency in winter and HCHO from continuously generated photo-oxidation contributed to the higher correlation between HCHO VCD and $NO_2$ VCD at noon higher than during rush hour. The transport of $NO_2$ and VOC may constitute one of the causes. The VOCs from transport generate HCHO due to strong photo-oxidation at noon. This result indicates that secondary photo-oxidation formation of HCHO from other VOCs could be the dominant source at UCAS.

### 3.4. Comparisons with the model data

HCHO VCDs retrieved from the MAX-DOAS measurements were compared with those from the CAMS model data from the period of October 1, 2014 to December 31, 2014 **(Fig. 15)**. The average value of the data from MAX-DOAS in the period of 7:30–8:30 LT was selected for comparison with the CAMS data at 0:00 UTC (8:00 LT) **(Fig.15a)**. The data from MAX-DOAS in the period 13:30–14:30 LT were averaged for comparisons with the CAMS data at 6:00 UTC (14:00 LT) (**Fig. 15c**). The CAMS model results with different grids are shown in **Fig. 15,** and their differences were found to be negligible. On average, the CAMS model underestimated HCHO VCDs by 1.6–2.0 $\times 10^{15}$ molec $cm^{-2}$ and 1.3–2.1 $\times 10^{15}$ molec $cm^{-2}$ compared to the MAX-DOAS measurements at 8:00 LT and 14:00 LT, respectively, due to different grid-sizes. The correlation coefficients and linear regressions between the MAX-DOAS data and the model results are shown in **Fig. 16**. The correlation coefficient $R^2$ was more than 0.68 and 0.79 at 8:00 LT and 14:00 LT, respectively. A comparison of the results in **Fig. 15 and 16** shows that the CAMS model and MAX-DOAS results are generally consistent, and the peak values

were particularly consistently captured by both datasets, but the low values were systematically underestimated by the CAMS model. As the peak values were mostly related to the transportation of pollutants from the southern area, the high consistency of the peak values between MAX-DOAS and the CAMS model implies that the CAMS model can adequately simulate the transport of pollutants. In addition, in the CAMS model, only gas-phase chemical reactions were considered for HCHO, as described in Section 2.5. Therefore, the high consistency of the HCHO values between the MAX-DOAS measurements and the CAMS simulations indicates that the heterogeneous reactions could not dominate the secondary formation of HCHO from other VOCs. The underestimation of the low HCHO values by the CAMS model compared to the MAX-DOAS measurements could be attributed to the lower constraint of local emissions in the model near the UCAS measurement station, and the lack of heterogeneous reactions in the model could also contribute to the underestimation of the low HCHO values. The China National Highway 111 is nearby and runs from north to south. The actual emission totals for the 2008 inventory included anthropogenic, biogenic and natural sources, and biomass burning, thus, highway emissions were considered in the 2008 inventory. However, due to the establishment of the UCAS from 2013 and the holding of the APEC meeting in 2014, the economy near the UCAS has grown rapidly, and the traffic flow had increased significantly in recent years. Thus, the use of the 2008 inventory could underestimate the highway emissions. As UCAS is in the suburban area of Beijing, vehicle emissions could largely contribute the HCHO amount under the weak transportation of pollutants from the polluted southern area. In general, the CAMS model can suitably capture distinct day-to-day variations in HCHO. In addition, day-to-day variations in HCHO could be attributed to variations in transport of pollutants, the secondary formation rate of HCHO, and local primary emissions. While a constant primary emission rate is assumed in the CAMS model simulations, the fluctuations in solar radiance and temperature, which impact the secondary formation rate of HCHO, are much smaller than the day-to-day variations in HCHO. Thus, the transportation of pollutants could be a dominant factor of the captured distinct day-to-day variations in HCHO.

The underestimation of HCHO by the CAMS model became statistically significant after December 1, 2014. This phenomenon could be related to the decrease in the temperature after December 1, 2014. Time series data of coincident hourly averaged meteorological parameters measured at the supersite from October 28 to December 31, 2014 at 8:00 and 14:00 LT are shown in **Fig. 15b and d**, respectively. Temperature is an important parameter impacting the production rate of

secondary HCHO. When the temperature decreases, the generation yield of the secondary photochemical reaction to produce HCHO decreases, resulting in low concentrations of HCHO. The temperature dramatically dropped after December 1, 2014, which could cause a low production rate of HCHO. The secondary sources of HCHO can be better simulated than the local primary sources in the model (Stavrakou et al., 2014; Anderson et al., 2015; Kwon et al., 2017). The CAMS model could thus underestimate the local primary emissions of HCHO, but the MAX-DOAS measurements could suitably obtain the HCHO from both local primary emissions and secondary generation. Thus, when the secondary source of HCHO is reduced, namely the "local" primary emissions of HCHO predominantly contribute to the HCHO VCDs, the difference between the MAX-DOAS observation and CAMS model becomes more pronounced.

R represents the ratio of HCHO VCDs in the morning (8:00 LT) and noon (14:00 LT). If $R_{Model}$ is close to $R_{MAX-DOAS}$, it indicates that the trend in diurnal variation of HCHO from the model simulation and MAX-DOAS observation is consistent, suggesting that the model can reasonably simulate the systematic diurnal variation in HCHO. The consistency of $R_{MAX-DOAS}$ and $R_{Model}$ can be used to characterize the ability of the model to simulate the diurnal variation of HCHO. The $R_{MAX-DOAS}$ and $R_{Model}$ are calculated as following Eq. (11) and Eq. (12):

$$R_{MAX-DOAS} = \frac{HCHO\ VCD_{MAX-DOAS\ at\ 8:00\ LT}}{HCHO\ VCD_{MAX-DOAS\ at\ 14:00\ LT}} . \tag{11}$$

$$R_{model} = \frac{HCHO\ VCD_{model\ at\ 8:00\ LT}}{HCHO\ VCD_{model\ at\ 14:00\ LT}} . \tag{12}$$

The scatter plots and linear regressions of daily $R_{MAX-DOAS}$ and $R_{Model}$ are provided in **Fig. 17**. Although large scatters were found, most of the dots were around the 1:1 line. Thus, the model can reasonably simulate the systematic diurnal variation in HCHO. It needs to be noted that the diurnal variation in HCHO is the result of the combined influence of primary emissions, secondary formation, and meteorology. We found that $R_{MAX-DOAS}$ was generally larger than $R_{Model}$. However, it was impossible to determine the factor causing the deviation in $R_{MAX-DOAS}$ and $R_{Model}$. Therefore, the R comparisons generally only evaluate the quality of the model simulations on diurnal variations in HCHO.

Clouds can impact the MAX-DOAS measurements. First, clouds can affect atmospheric radiative transport and thus influence optical paths. Furthermore, the atmospheric absorber densities [by (photo-)chemistry or convective transport] are potentially altered due to the changes in optical paths (Grats, ea et al. 2016). Second, AMFs calculated by geometrical

approximation could be significantly biased from the reality under cloudy conditions (Brinksma, et al., 2008). The effects could impact comparisons between MAX-DOAS and the CAMS model. To test this, we compared MAX-DOAS and the CAMS model HCHO results under different effective cloud fractions (eCF) from the ECMWF (**Table S1 in the Supplement**) (**Fig. S1, S2, S3 and S4 in the Supplement**). The consistencies between the two datasets varied only slightly as the cloud fractions increased. This outcome indicates that clouds have little effect on the comparisons of MAX-DOAS and the CAMS model, which is consistent with the finding of Wang et al (2017b).

## 4 Conclusions

We studied the tropospheric HCHO VCDs at the UCAS site in Huairou District, Beijing around the APEC summit based on the MAX-DOAS measurements from October 1, 2014 to December 31, 2014.

The UCAS site was affected by the transportation of pollutants from the south. Two peak values on November 4 and November 7, 2014 during the APEC summit were caused by a change in wind direction to the south and an increase in wind speed of $> 2.0$ m s$^{-1}$ when the polluted air masses from the south were transported to the UCAS site. Marked wind speed and direction dependences of HCHO VCDs were identified. Wind direction dependencies indicated that the HCHO values in the area around UCAS were considerably affected by the transportation of pollutants from the south, including the southwest and southeast where heavily polluted cities are located, such as Tangshan, Shijiazhuang, and Tianjin. Conversely, winds from the north and northeast contributed to the dispersion of HCHO.

The impact of control measures on HCHO were evaluated using approximately one month of MAX-DOAS data from October 26 to November 20, 2014, which was defined in three episodes. The first episode was the period of APEC (from November 1–12); the second and third episodes were the "pre-APEC" period from October 26–31, 2014 and the "post-APEC" period from November 13–20, 2014. During the period of the APEC conference, the average HCHO was $6.0 \times 10^{15}$ molec cm$^{-2}$, which was ~38%±20%, and ~30%±24% lower than that during the pre-APEC and post-APEC periods calculated at 95% confidence limit, respectively. Prevailing northwest wind fields and strict control measures in combination led to the relatively low HCHO values during APEC.

The daily variation in HCHO VCDs at the UCAS site indicated that the values at noon and evening rush hour were higher

than those in the morning. Furthermore, peak values appeared around noon, and a good correlation coefficient $R^2$ of 0.73 between HCHO and $O_3$ was found around noontime. This finding indicates that the secondary sources of HCHO through photochemical reactions dominate in the area around UCAS.

The time series data of HCHO VCDs retrieved by MAX-DOAS and the CAMS model were consistent from October 1, 2014 to December 31, 2014. The CAMS model underestimated HCHO VCD by about 24% on average compared to the MAX-DOAS measurements. The CAMS model could adequately simulate the effects of the transport and the secondary sources of HCHO, but underestimated the local primary sources, which were more pronounced under low temperature conditions when the production rate of secondary HCHO was relatively low. Generally consistent ratios of HCHO VCDs at around 8:00 LT and 14:00 LT were found for the CAMS model simulations and MAX-DOAS measurements. It indicates that the CAMS model can reasonably simulate the systematic diurnal variation in HCHO.

**Author contributions**

XT, PX and JX contributed to designed the research. JX and AL designed the installation location of the MAX-DOAS instrument and installed it in the UCAS. ZH and QZ downloaded and extracted HCHO VCDs data from ECMWF. XT performed the data analyses and wrote the manuscript. FW and CL provided suggestions for the manuscript. PX, JX and YW edited and developed the manuscript.

**Acknowledgment**

We thank the Belgian Institute for Space Aeronomy (BIRA–IASB) in Brussels, Belgium for the freely accessible Windoas software and the European Meteorology Center for providing free medium-range weather forecasts for CAMS real-time products of HCHO. We also thank the University of Chinese Academy of Sciences and Peking University for their support and assistance in the observation and supply of essential data. This study was supported by the National Natural Science Foundation of China (Grant Nos.: 41530644, 41405033, and 41605013).

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

**Tables**

| Spectrometer | | Azimuth | Elevation | Temperature | Location | | Measuring time |
|---|---|---|---|---|---|---|---|
| Name | Maya (Ocean Optics) | 0 ° | 3 °; 5 °; 10 °; 15 °; | 20 ℃ | Site | Yanxi Lake campus of UCAS | 6:30-18:30 |
| Spectral range | 290– 420 nm | | 30 °; 90 °; | | Longitude | 116.67 °E | |
| FWHM | 0.5 nm | | | | Latitude | 40.4 °N | |

10 **Table 1: Setting of MAX-DOAS.**

| Parameter | Data source | Fitting interval: nm<br>335–360 nm (HCHO)<br>338–360 nm (NO$_2$) |
|---|---|---|
| NO$_2$ | Vandaele et al., (1996), 294 K | x |
| O$_3$ | Serdyuchenko et al., (2013), 223 K, 243K | x |
| O$_4$ | Thalman and Volkamer, (2013), 293 K | x |
| BrO | Fleischmann and Hartmann, (2004),223 K | x |
| HCHO | Meller and Moortgat, (2000),   293 K | x |
| Ring | Chance and Spurr, 1997 | x |

| Polynomial degree | | | 5 |
| --- | --- | --- | --- |

**Table 2: Parameter settings used for spectral analysis using WINDOAS, where "x" indicates the cross section used in the retrieval.**

**Figures**

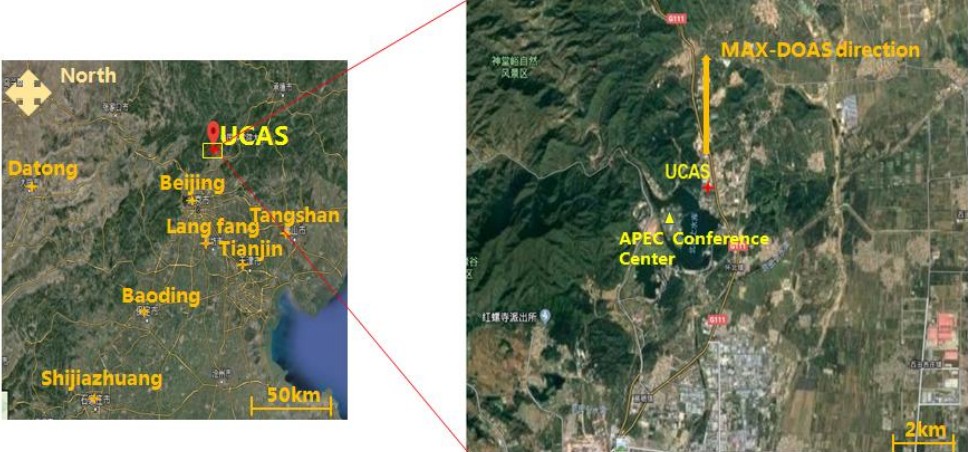

**Figure 1: Supersite on the Yanqi Lake campus of UCAS.**

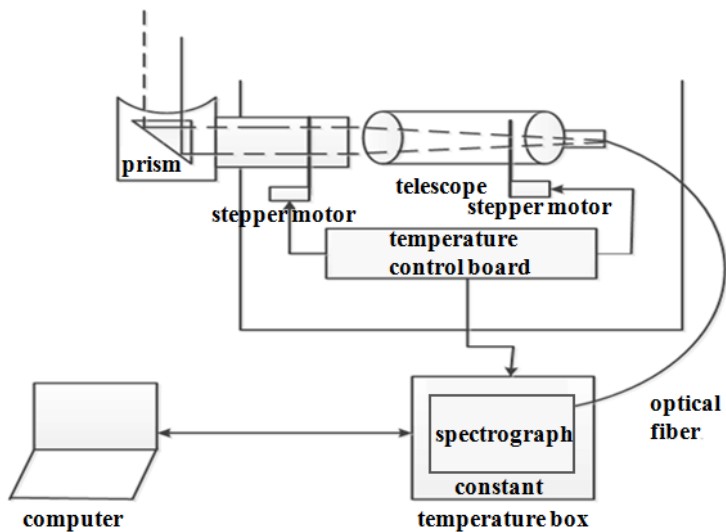

**Figure 2: Experimental setup of MAX-DOAS.**

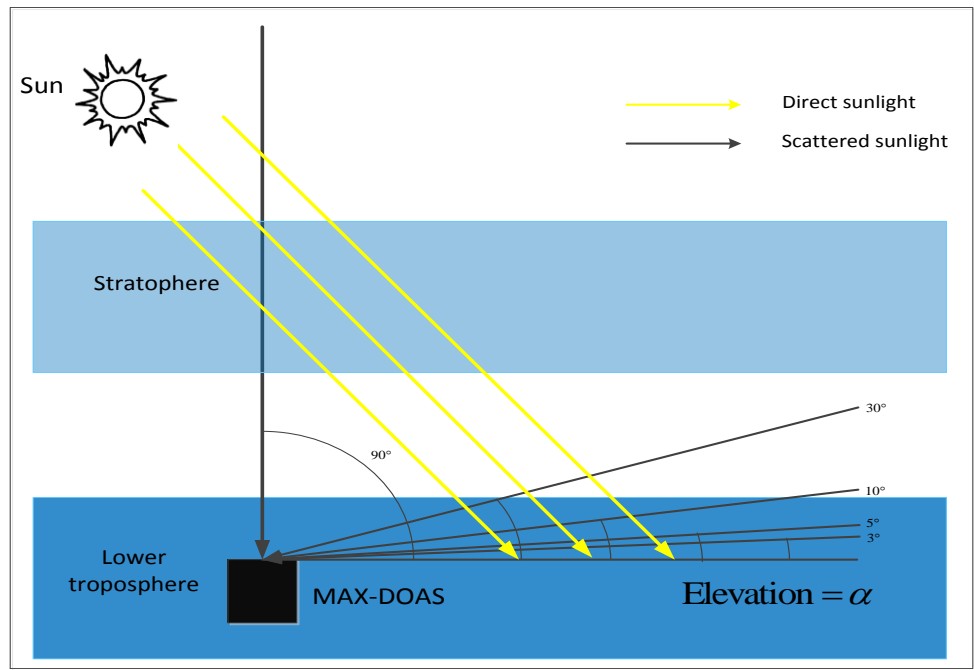

Figure 3: Observation geometry of MAX-DOAS.

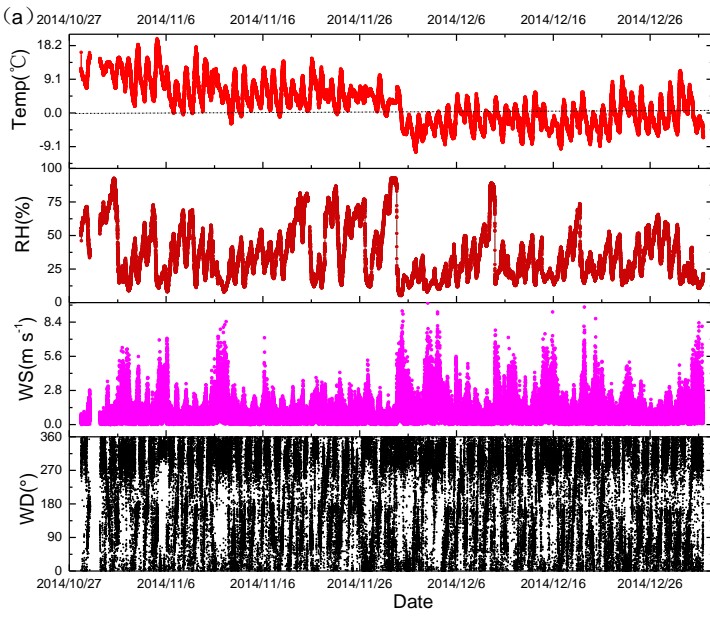

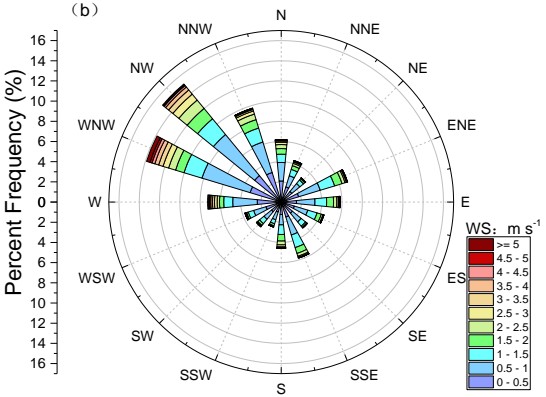

**Figure 4 (a) Time series of the meteorological parameters with a time resolution of 1 min containing ambient temperature, relative humidity (RH), wind speed (WS), and direction (WD). (b) Wind rose. Data obtained during the period of October 28, 2014 to December 31, 2014.**

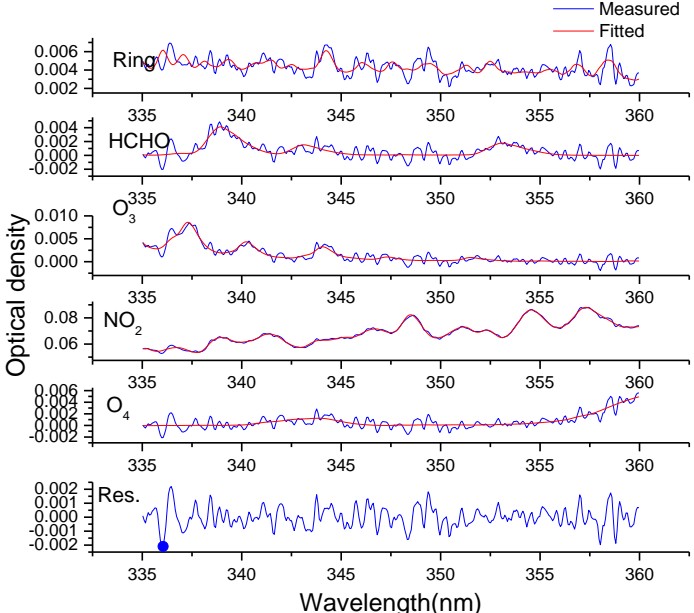

**Figure 5: Example of a DOAS fit of a spectrum to retrieve the slant column densities of HCHO; the red and blue curves indicate the fitted absorption structures and the derived absorption structures from the measured spectra, respectively.**

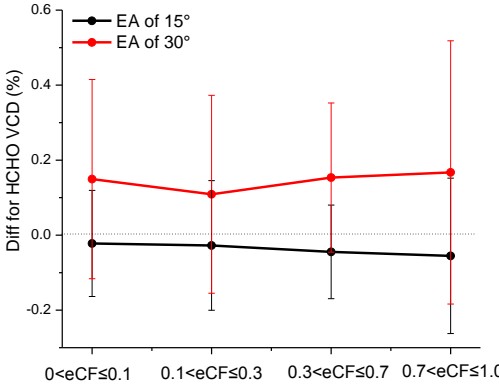

**Figure 6. Relative differences in the tropospheric HCHO VCDs derived by geometric approximation and from profile inversion as a function of the effective cloud fractions (eCF), as 0<eCF ≤1, 0<eCF≤0.3, 0.3<eCF ≤0.7, and 0.7<eCF≤1.0 for elevation angles of 15 °and 30 °. Error bars denote the standard deviations. Diff were calculated by Eq. (9) in the text.**

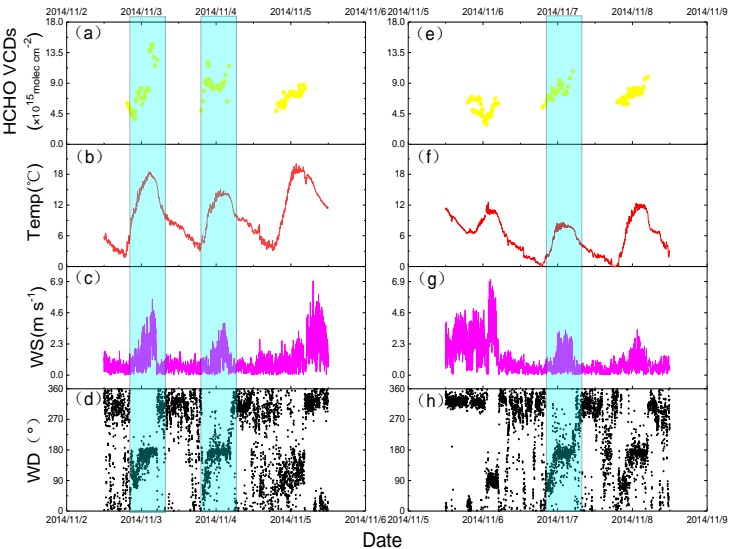

**Figure 7: Time series of HCHO VCDs (molec cm⁻²) and meteorological parameters [ambient temperature( ℃), relative humidity (%), wind direction ( ) and wind speed (m s⁻¹)] measured at the supersite in the two periods of November 3, 2014 to November 5, 2014 (left column) and November 6 to November 8, 2014 (right column).**

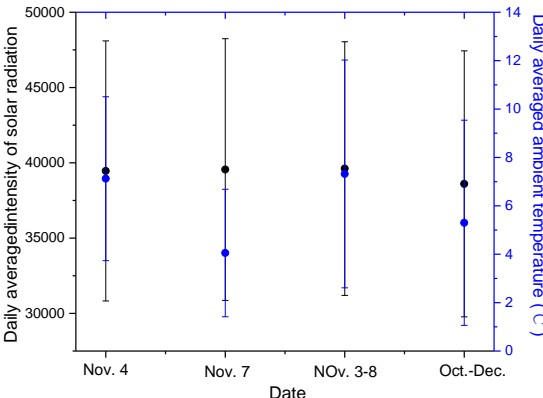

**Figure 8: Averaged intensity of solar radiation and temperature in the four periods of November 4, November 7, November 3 to 8, and October 1 to December 31, 2014. Error bars denote the standard deviations.**

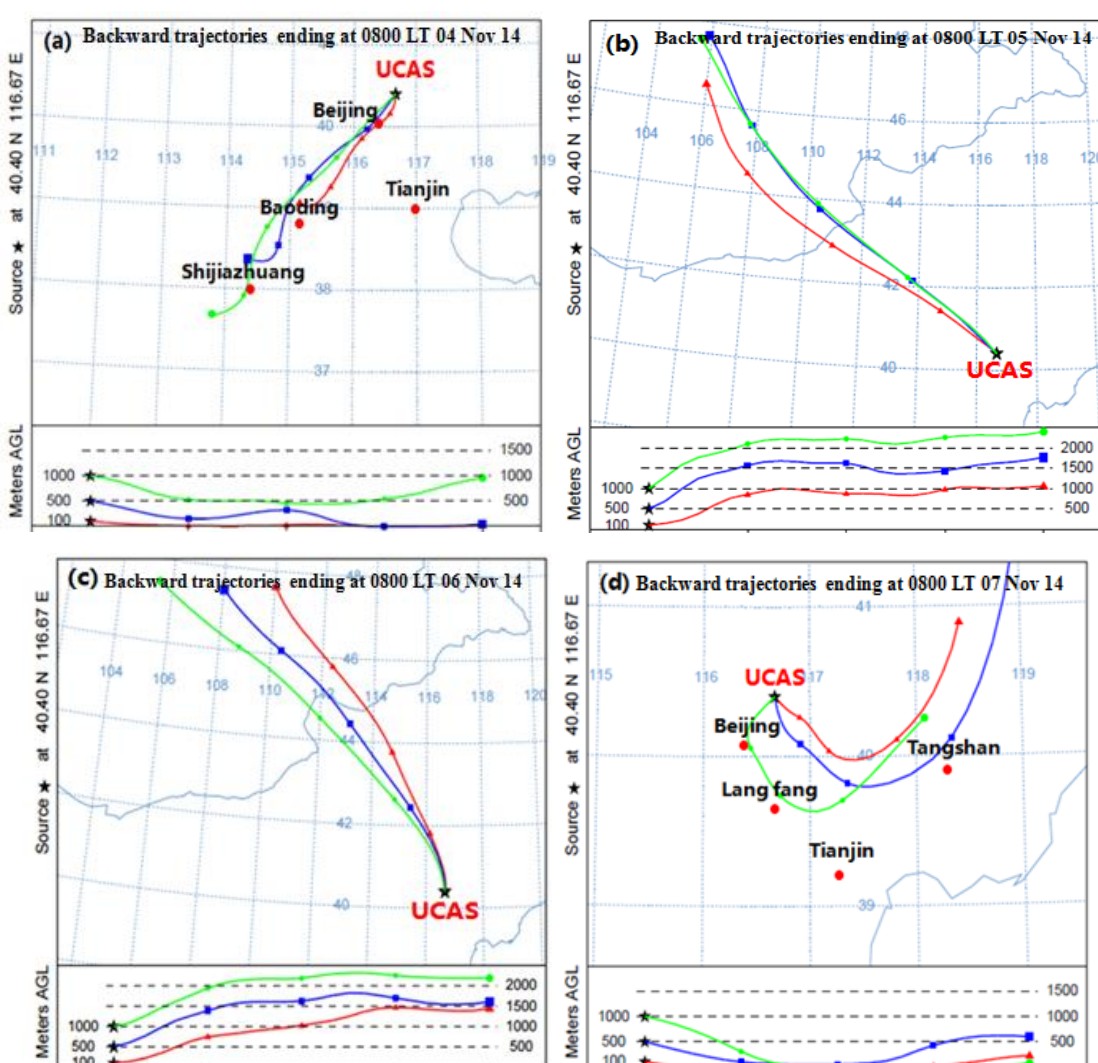

**Figure 9: Backward trajectories determined by the HYSPLIT model at UCAS on November (a) 4, (b) 5, (c) 6, and (d) 7, 2014.**

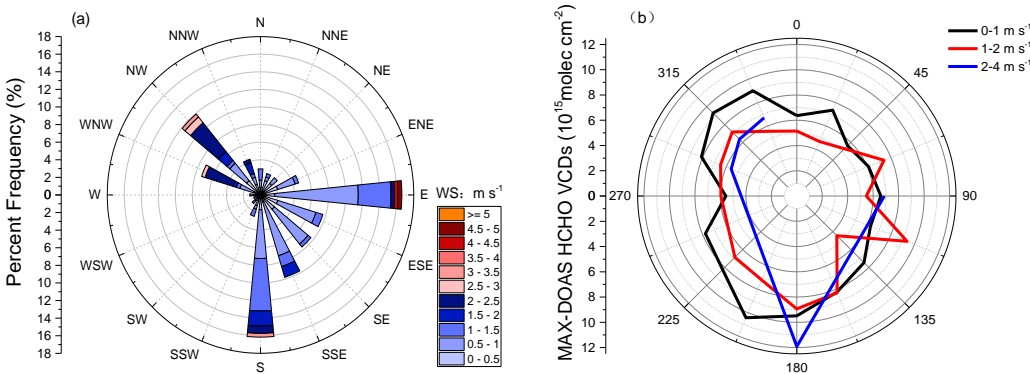

**Figure 10: (a) Wind rose; (b) dependence of HCHO VCD ($10^{15}$ molec cm$^{-2}$) on wind directions for different wind speeds.**

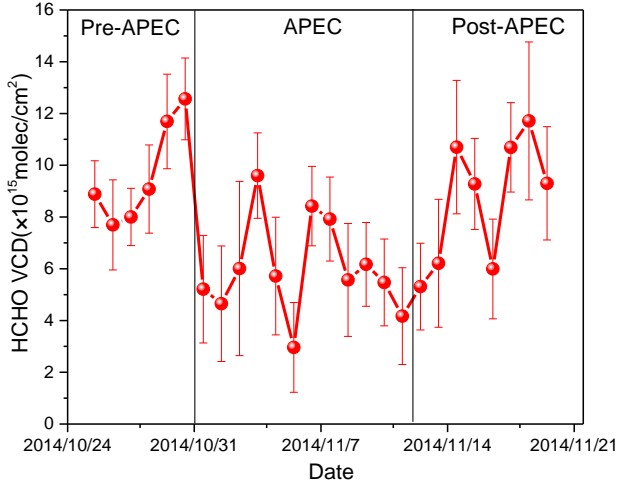

**Figure 11: Daily averaged values of HCHO VCDs from October 26, 2014 to November 20, 2014. Error bars denote standard**

deviations.

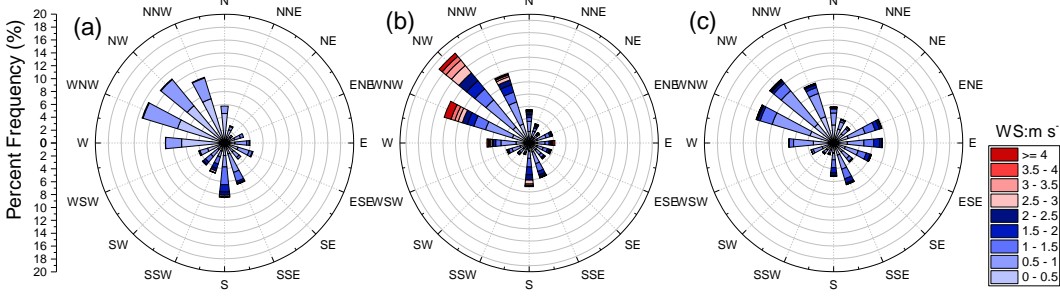

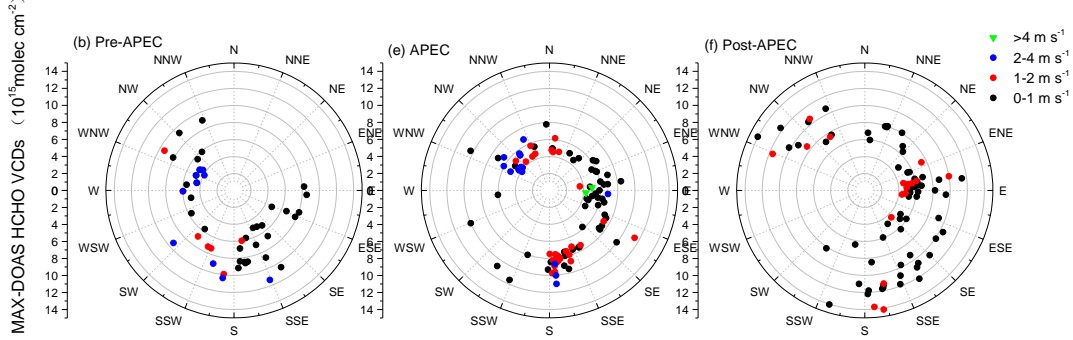

**Figure 12: Wind roses in (a) the "pre-APEC", (b) the APEC, and (c) the "post-APEC" periods. Dependence of HCHO VCDs (10$^{15}$ molec cm$^{-2}$) on wind directions for different wind speeds in the pre-APEC (d), during the APEC (e), and post-APEC (f) periods.**

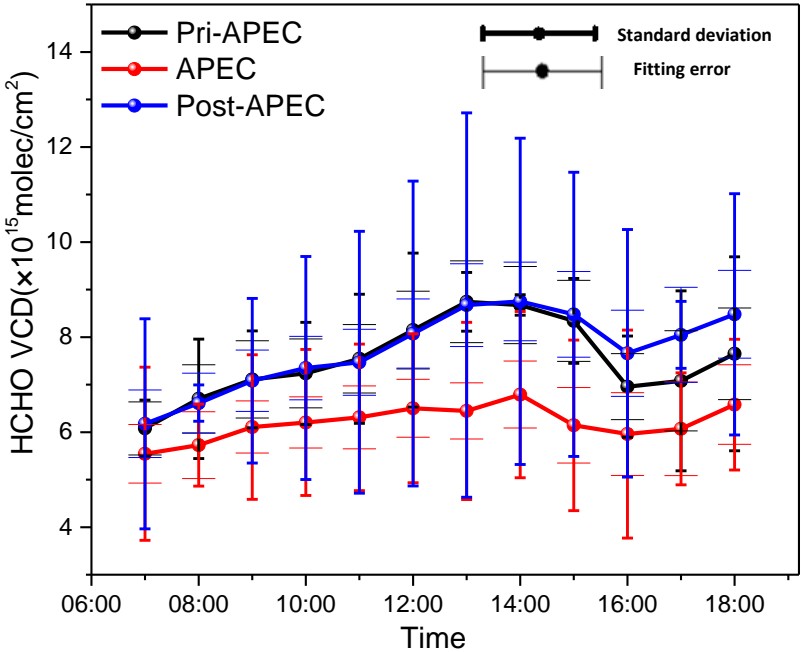

**Figure 13: Averaged diurnal variation in HCHO VCDs measured by MAX-DOAS in three episodes around APEC. The short cap width of the error bars denotes the one sigma standard deviations around the mean analysis values. The long cap width of the error bars denotes the fitting error.**

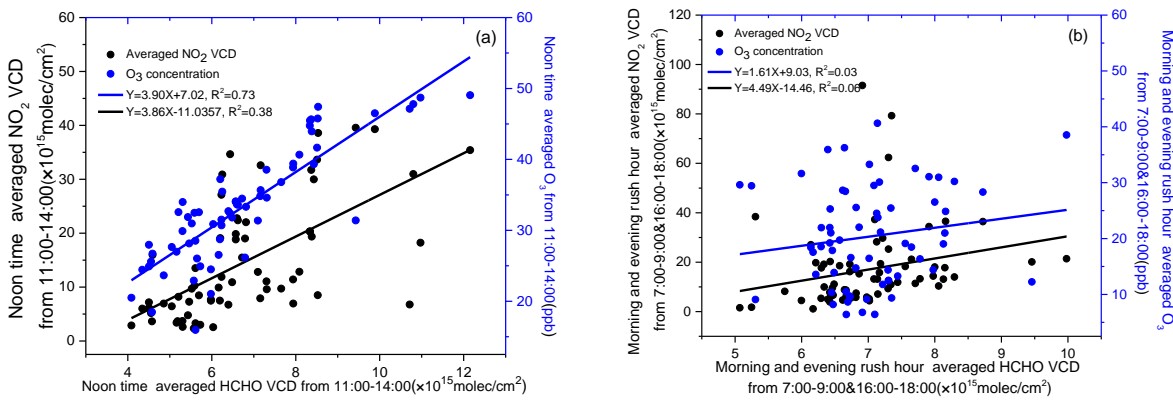

**Figure 14: Scatter plots and linear regressions (a) of noontime average HCHO VCD measured by MAX-DOAS against O₃ VMRs measured by a stationary ozone monitoring instrument, and (b) rush hour average HCHO VCD against NO₂ VCD measured by**

5     **MAX-DOAS from October to December 2014.**

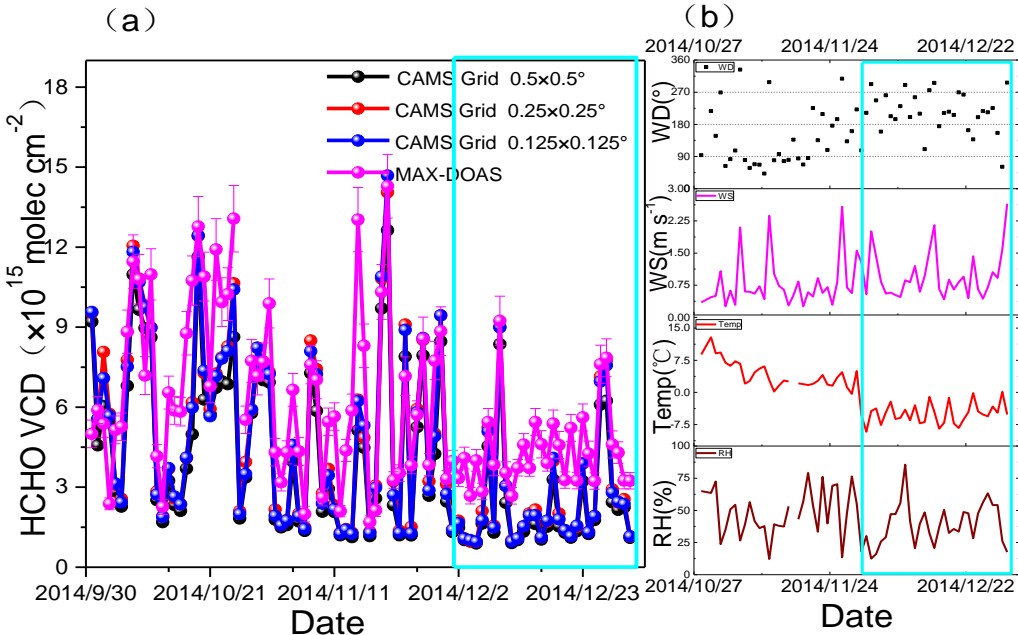

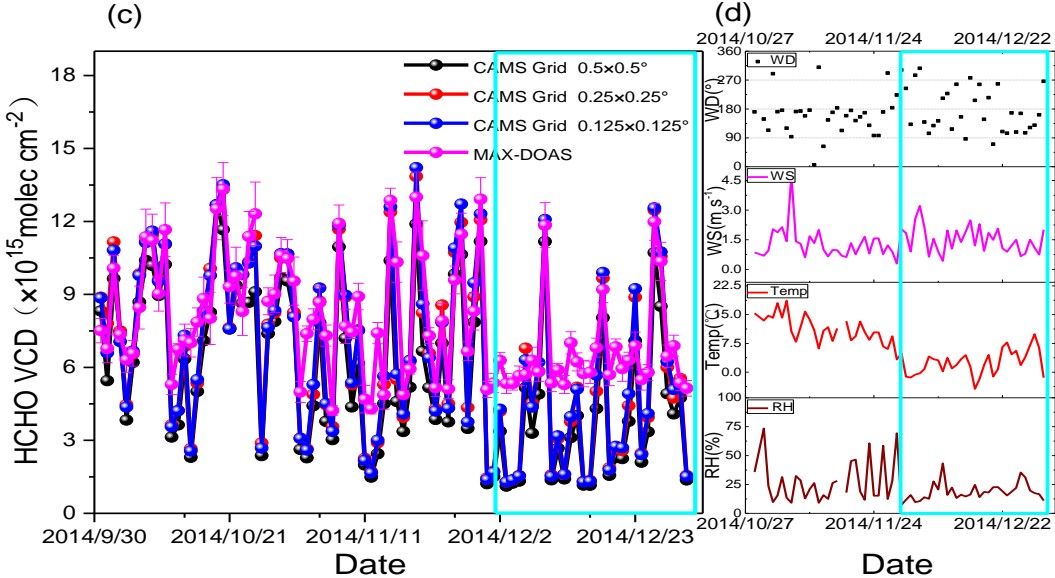

**Figure 15: Hourly averaged HCHO VCDs derived from the coincident CAMS model (grid of 0.125 °× 0.125 °, 0.25 °× 0.25 °, and 0.5 °× 0. 5 °) and MAX-DOAS observations at 8:00 (a) and 14:00 LT (c) from October 1 to December 31, 2014. Error bars denote retrieval error. Time series of coincident hourly averaged meteorological parameters measured at supersite at 8:00 (b) and 14:00 LT (d) from October 28 to December 31, 2014.**

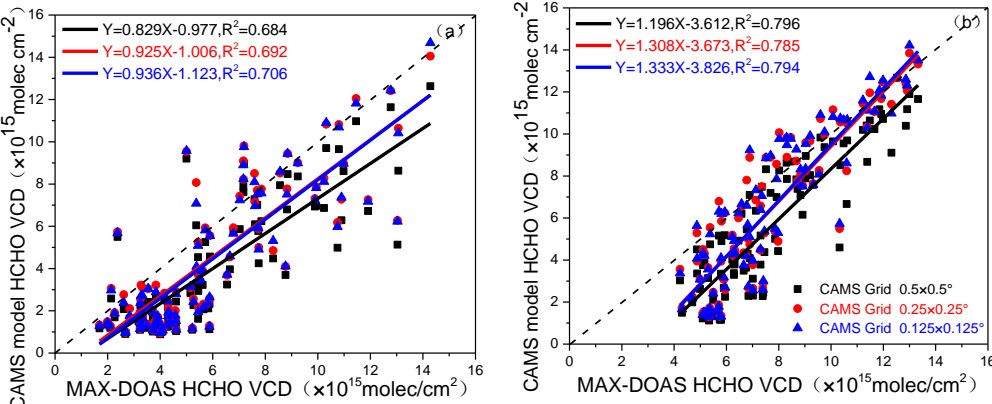

**Figure 16: Correlation between HCHO VCDs retrieved from the MAX-DOAS measurements and those obtained from the CAMS model at 8:00 LT (a) and 14:00 LT (b) from October to December 2014 in different grids.**

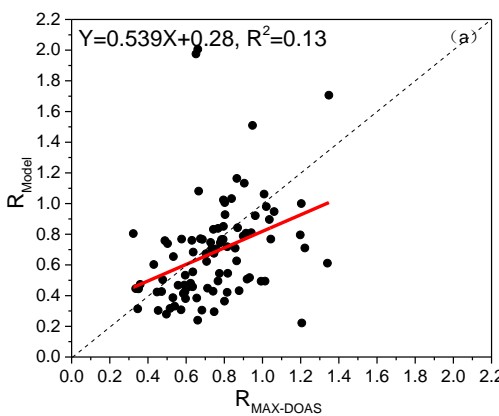

**Figure 17: Scatter plots and linear regression of** $R_{Model}$ **against** $R_{MAX-DOAS}$ **(refer to the text).**