# Peer review of "Ground-based MAX-DOAS observations of tropospheric formaldehyde VCDs and comparisons with the CAMS model at a rural site near Beijing during APEC 2014"

_Atmospheric Chemistry and Physics, 2018_

## Editor Comment (EC1) · Prof. Li (Editor) · 31 Jul 2018

We have exceptionally difficulties in finding reviewers for your article. Well over 10 potential reviewers have been contacted but so far none review has been received. Since they are all able to view your article, the lack of interest would imply either the study is not of broad interest, or insufficient original findings. Per my reading, I am afraid that that the study is concerned with some specific observation results from which not enough original scientific insights are gained. Without a further delay, I'd not pursue it any further so that the authors can choose another journal for publication.

[Figure]

2018.

---

## Referee Comment (RC1) · Anonymous Referee #1 · 20 Sep 2018

General Comments:

The paper does address relevant scientific questions within the scope of ACP. The authors give proper credit to related work and indicate their own new/original contribution. How their work is original could be more clearly emphasized. The abstract, number and quality of references, and the amount of supplementary material are all satisfactory. However, a problem that detracts from the entire paper is that the language is not fluent and precise. There are frequent spelling and grammatical errors (fragment sentences, unnecessary words, incorrect verb tense, missing articles, convoluted or run-on sentences). The authors are strongly suggested to engage the help

of an English editor. A few sections would benefit from re-structuring for increased logical flow and clarity. The scientific methods and assumptions are not sufficiently clearly outlined. The methodology section is disorganized and needs additional technical details. The paper does present some novel data and reaches substantial conclusions, but the results are sometimes not enough to support the interpretations and conclusions without further statistical analysis and/or expanded discussion. When discussing results, the authors must consider whether trends and differences in measured values they are interpreting are statistically significant given calculated or expected uncertainties. The authors must also try to place their conclusions within the context of previous literature (e.g., presented in the introduction).

Specific Comments:

For all regressions, the coefficient of determination ($R2$) statistic may be more appropriate since this value indicates the variance in the dependent variable that is predictable from the independent variable. Page 2 Line 3 What specific results had a good correlation coefficient? VCDs of HCHO? Other? Page 2 Line 23 Did the emissions decrease from 100% to these values or are these values the quantity of the decrease? It is not clear. Page 2 In general, since some of the pollutants were measured by multiple papers that you cite, consider sorting this paragraph by pollutant rather than by author. Otherwise, it becomes repetitive and confusing to have to keep referring to the values from the previous papers earlier in the paragraph. Page 3 Line 6 Three sources are listed despite elsewhere in the paper it is stated that there are two sources, which one is it? Page 3 This discussion may benefit from writing out some of the most important chemical equations for the reader equations for the reader. Page 3 Line 16 It is unclear whether the importance of quantifying HCHO is to track emissions of VOCs or NVOCs or the generation of OH, or all of these. Page 4 Line 5 What unit of HCHO? VCDs? Page 4 Line 10 The last paragraph of your introduction may benefit from explicitly stating your research objectives (perhaps as a list). What were all the components included in the spectral fitting for HCHO? Include what cross section reference spectra

used (including author), fitting window etc. Section 2.1 Please list the final equation used for calculating VCDs from fitted DSCDs given your geometric approximation. Reorganize section 2 so that sections 2.1, 2.3, and 2.4 are grouped together for a more logical sequence flow. What is included in the VCD error calculation? Are any of the MAX-DOAS VCDs removed from the dataset due to cloud fraction? If so, what was the cut off cloud fraction value? Page 5 Line 8 It is unclear what causes lower systematic errors. Page 6 Line 18 Why was the FRS from this day chosen to fit all retrievals? Since you are fitting all your measured spectra against one FRS, you must consider the effect of SCD(FRS) (the component of trace gas in the FRS used) and the SCD(Solar Zenith Angle), which is the difference in the stratospheric component of SCD observed due to the difference in SZA between the times of measurement and the FRS time. SCD(SZA) changes with time and change the apparent diurnal trends. Please justify why you did not account for the SCD(SZA) and SCD(FRS). For example, was your FRS was obtained during a very low pollution period and/or are the stratospheric HCHO levels are expected to be trivial. For more information see Wagner, T., Ibrahim, O., Shaiganfar, R. and Platt, U.: Mobile MAX-DOAS observations of tropospheric trace gases, ATMOSPHERIC Meas. Tech., 3(1), 129–140, 2010. Page 6 Line 23 What is the software reported SCD error in molec/cm2? Page 6 Line 24 Are you missing units on this number? Page 7 section 2.6 At what height were these meteorological parameters measured? At what time frequency before averaging? Page 7 Line 24 What does a "static" weather situation mean? Better organization and flow in section 2 may be achieved by describing MAX-DOAS methodology in this order: general description of the MAX-DOAS instrument, description DSCDs fitting, determination of VCDs from DSCDs, measurement sequence, and then viewing azimuth and location. Figure 6 – Are all the days measured or the ones that met some quality control criteria? Page 8 Line 13 Please quantify in some way the relative change in solar radiation and temperature compared to the days were peaks were not apparent. Page 8 Line 21 Please define (or find a better descriptor for) "good dispersion conditions". Page 9 Line 14 What type of relevant pollution sources do these cities have? Primary and/or

secondary? Are there many industry and/or vehicular sources? Page 9 Line 16 You may want to state explicitly that there are fewer, smaller or less polluted cities in the Northern region here. Are the lower VCDs due to just dispersion or is it also chemical aging, etc. Page 9 Line 17 and 18 Please explain the dependence of the VCDs on wind speed under different wind directions. Page 10 Line 6 What are the errors on each of the VCD values. Are they statistically different? Page 10 Line 17. This sentence is too vaguely written. Also, depending on whether the differences between the peak values are statistically significant, depending on the expected errors and the significance of the wind direction change, you may not have sufficient evidence to support this conclusion. Also consider that during APEC time the conditions were not only northerly winds but also higher wind speeds, which you state earlier in the paper tends to reduce the VCDs (which should be explained for clarity). Same comment for the sentence on lines 19 and 20. Page 10 Line 25 Some basic equations on HCHO chemistry in the introduction section would be very helpful for the reader by the time they get to this point in the paper. Page 11 Line 3 Are you suggesting that this peak in the diurnal variation is due primarily to secondary production of formaldehyde rather than direct emissions? Since the most light is available mid-afternoon and local direct emissions are relatively smaller compared to secondary production? Please make this clearer to the reader. Page 11 Are your conclusions that the diurnal variability is driven by variation in light levels rather than diurnal variations in emissions? If light measurements are available, you could try correlating the light intensity with the VCDs. Page 11 Line 5 Where can the reader see evidence of similar diurnal trends in the secondary sources? Page 11 Line 6 Many of the VCDs in the during, before and after APEC periods are equal within error. Are you referring to only the afternoon peaks HCHO values? The peak during APEC value appears to be equal within error with some of the highest post APEC values. Page 11 Line 7 What are the actual values with associated errors and are they statistically different? Page 11 Line 8 Please explain your reasoning. Page 11 Line 14 Where were the in-situ ozone measurements located relative to the MAX-DOAS measurements? Put this information in methodology. Section 3.3 would benefit from

a reorganization. Perhaps put information about primary versus secondary sources and the correlations first before making conclusions about diurnal trends. Section 3.4 How are VCDs calculated from the model output (i.e., what vertical height interval was integrated from the modeled vertical profile?) Section 3.4 Explain in more detail why the model poorly captures the local emissions. Could the lack of heterogeneous reaction in the model be contributing to the underestimation of the low HCHO values? Page 12 Line 14. Since the grid size seems to have little impact on the quality of the model output, is the "worse constraint" due to poor or outdated emission inventories local sources in this area? Are the highway emissions included in model calculations? How accurate is the emission inventory of the highway if it's included in the model? Page 13 Line 8 When you say "the primary HCHO is dominant" do you mean that the dominant contribution to the HCHO VCDs is the "local" primary emissions of HCHO? Edit for further clarity. Page 13 Line 14 Your conclusion is not necessarily sufficiently supported given the small R2 value and "reasonably" (too vague) would have to be defined before it is clear whether the data support this statement sufficiently. Page 14 Line 8 You may want to add that, in contrast, correlation with NO2 was lower and what that implies. If VCDs are calculated from the 10° and 30° spectra, how do the values compare to the 15° spectra VCDs? Given that the geometric approximation becomes worse under high aerosol conditions and these VCDs would be expected to diverge in that case, comparison with the 10o and 30o spectra may be a good measure of the validity of your use of the geometric approximation. Figure 11 Can you explain why the standard deviation of the pre-APEC time is so much smaller than the post-APEC period despite similar values? Figure 9 Since you show average values, how did the standard deviations of the averages compare to the retrieval errors? Are the larger of the two plotted as error bars? Figure 8 Why do moderate wind speeds appear to produce similar VCD values for all wind-directions. Also, why do southerly conditions appear to result in maximum VCDs occurred under the highest wind speeds given that you stated that high wind-speeds tend to reduce VCDs? Can you divide the VCD data into wind-speed and time of day and then see if there is a statistically significant reduction

of the VCDs under non-Southerly wind conditions during APEC compared to before and after? That may help to determine how much the emissions controls impacted the VCDs independent of wind-direction.

Technical Corrections:

Be careful always to include the correct type of units when quoting numbers, where applicable. Title: Consider adding VCDs after the word formaldehyde. Consider also including the APEC study to the title. General technical comment: when listing VCDs to route, please include the error values. Page 1 Line 22 Abstract: what are the units of HCHO and O3? VCDs? Mixing ratios? Page 2 Line 8 What were the specific dates of the conference? Page 2 Line 18 It is unclear what "traffic" and "regional" stations are? Page 2 Line 20 Missing subscript. Page 2 lines 20 and 21 This statement is too vague as to be informative Page 2 Line 21 What are the units of the measurements of these gases? Page 3 lines 10 and 11 The meaning of this sentence is unclear. Page 3 Line 24 I believe this should say tropospheric column densities, surface mixing ratios, and vertical profiles of aerosol extinction and trace gas mixing ratios. Page 4 Line 18 Consider changing to "derived from the DOAS spectral analysis [of the measured spectra]" Page 6 Line 22 In this sentence and figure 4 you use different terms for the blue and red lines: blue (measured, derived ) red (calculated, retrieved, fitted). Pick one term for each and ensure that the meaning of the term is clear. What about the contribution of the residual to the blue line? Page 10 Line 4 the sentence needs editing for greater clarity and to appropriately describe figure 9. Page 13 Line 24 HCHO were also studied before and after APEC, were they not? Table 1 There are small spacing and English errors. Clarity of Figure 9 may be improved by lines or boxes that indicate the afternoon period. Figure 6 If relative humidity is not discussed in the results or discussion, perhaps remove it from the figure to have more space to expand the more relevant data. Supplement: More helpful analysis may be achieved by dividing the regressions into bins that do not all include zero cloud fraction. For example, are different trends observed for eCF 0-0.3, 0.3-0.5, 0.5-0.7 etc.?

---

## Referee Comment (RC2) · Anonymous Referee #2 · 9 Oct 2018

Content-Summary

The paper entitled "Ground-based MAX-DOAS observations of tropospheric formaldehyde and comparisons with [the] CAMS model at a rural site near Beijing" focuses on the analysis of MAX-DOAS measurements of HCHO from the University of Chinese Academy of Science (UCAS) approximately 50km away from the Beijing city centre during the period of October 1 – December 31, 2014. The study includes the period 3-12 November 2014, during which the Asia-Pacific Economic Cooperation (APEC) summit was held. During APEC the Chinese government implemented strict control measures on emissions allowing this to be a rare "clean" period.

[Figure]

Not surprisingly the authors found decreased HCHO vertical column densities (VCDs) during the APEC period when compared to pre-APEC (-38%) and post-APEC periods (-31%). These reductions were thought to be from the control measures plus meteorological effects (predominant northwest wind fields bringing "cleaner" air to the site).

The authors also compared their HCHO VCDs with those determined using the Copernicus Atmosphere Monitoring Service (CAMS) model and found a good agreement (R=0.83). The trends of both data sets were similar, with the exception of the CAMS model's HCHO VCDs being significantly lower during periods of minima HCHO VCDs. The author's attribute this difference due to the CAMs model underestimating primary HCHO emissions.

Referee's Recommendation

This paper has good scientific significance and quality; however the presentation of the material and analysis of the results needs to be improved. In particular, although the level of English is passable several repeatable errors in grammar interrupt the general flow of the paper. I have no provided no comments on these errors. The paper's conclusions are fairly clear, but some of the detail of how the analysis was done is vague and needs to be more thoroughly explained. In particular, there is no mention of error analysis, while there are associated error bars on the figures, for example. At this time, I believe the paper is not ready for publication. However, if the comments outlined below are addressed I would be happy to consider a revised version for publication.

Scientific Comments

I have separated my comments into 2 categories (major and minor). Major comments require the author's attention. See page and line as required. {Q} = questions, {REFs} = reference, {C} = comments.

Major

(1) Use of the Geometric Approximation Only

P5, l8: Your comment, "...it has lower systematic errors because of the geometrical approximation" needs to be backed up. Do you have a reference, maybe a short explanation of your reasoning? You need to somehow prove to me that it is better to use a geometric approximation of the VCD vs. one of several RTM/inversions approaches. You did not do this here, but rather allude to some other studies. I am not convinced that the geometric approximation is the best? Prove me wrong?

Alternatively, you could provide a comparison of your VCDs with RTM-inversion derived HCHO VCDs. I would also suggest that you provide a comparison of your HCHO VCDs to those measured via satellite. This would give me more confidence in your conclusions.

(2) Emission Totals from 2008 for CAMS model P7, l12: You use emission totals from 2008. Your year of study is 2014, that is a difference of 6 years and a lot can change. Why didn't you use a more recent emission inventory? Is there one? If so, why didn't you use it?

(3) HCHO VCD error

P9, l11-120: It is likely that when the wind comes from the south it is more polluted than when the wind comes from the north. However, an average HCHO VCD of 7.57*1015 vs. 6.64*1015 is hardly conclusive. This is a 14% difference. WHAT IS THE ERROR OF YOUR VCD? I would estimate that is it a least 10%, likely over 20%. As such, your statistics here are weak. Please define the error of your VCDs and then re-word this section. For example, in Figure 9, you have error bars on your VCDs, but no mention of how you calculate them [they also look very low to me].

P10, l6: This comparison shows a distinct difference between pre-, APEC, and post-periods. However, again, there is no mention of errors? Please revise including using appropriate significant figures. Did each period have the same number of data points?

(4) Figure 12 – Correlation Analysis

I understand what you are trying to achieve here. However, I am not sure why you choose the period Oct 26 – Nov 20, 2014? This seems random? Why not use all your data?

The NO2 VCD is not described. Is it VCDgeo? Is it data from the same instrument and time? Did you also compare your O3 with the 7-9 & 16-18 periods? You don't have to show the plot but I would like to know the R of that? Hopefully it is very low to prove your point. Similarly, did you compare the NO2 VCDs with the HCHO VCDs from 11-14 period. You need a more complete assessment here to really prove your point.

What happens if the R value for O3 and 7-9&16-18 periods is also high? I believe you have something here but be careful about how you present it. I also need to know exactly where your O3 monitor is, is it at ground-level?

(5) Assumption that the HCHO VCD is the correct result

On P12, l3 you state that the CAMS model UNDERESTIMATES . . ..

How do you know this? How do you know the MAX-DOAS result is the correct result and better than the CAMS model? What other VALIDATION do you have? Did you compare it to the satellite data; ground-data extrapolated to a column {see comment 1}? You may be right, but you may also be wrong. I am not convinced, especially without any error analysis of your HCHO VCDs or CAMS model. I would say that your CAMS model could be really off since it uses emission totals from 2008. Maybe the emission estimates in the model for 2008 are simply much lower than the 2014 values? You allude to this on P12, l14-15, right?

P12, l3: What do these ranges mean? Is it due to different grid-sizes?

(6) RMAX-DOAS vs Rmodel

P13, l10: You R concept is interesting. Based on this I would think that R(DOAS) should be higher than R(model) for cases when the temp is cold (and secondary is HCHO is lower than predicted via the model), do you see this? Alternatively, if primary

HCHO emissions are under predicted in the model R(DOAS) again would be higher than R(model) right? So what does this R concept really tell us? A graph like Figure 15, does not tell me much? However, if you separate out case studies maybe you get some more information.

P14, 121-23: If the CAMS model underestimates primary sources of HCHO then R(DOAS) > R(model) but "under a situation with a low temperature when the production rate of secondary HCHO is relatively low" won't the CAMS model also underestimate the secondary HCHO production also causing R(DOAS) > R(model) as well? What is the assumed temp in the model, or does it use real-time met-data? How do we know what is the problem, is it a problem with the assumed temp, if so can you adjust that to check? OR is it a problem with the emissions inventory (perhaps a bigger issue).

Again, the above concept seems to have merit, but you need to develop this and explain it further, because I am somewhat confused. Also, despite your analysis I have no feeling as too how much HCHO is secondary and how much is primary (and isn't that what the R calculations are for?).

Minor

P2, l3: {Q} Is the correlation coefficient (R=0.83)? If so, say (R=0.83, not ~0.83)

P2, l14: {Q} How is "APEC blue" defined? Perhaps a brief statement of how the actual reduction strategies were defined and the defined APEC levels would be useful? Is there an APEC-red for example?

P2, l16-17: {Q} Do you or the authors of the Wang et al. make any conclusion as to why the O3 rose to 189%? Does this have to do with being in a NOx-limited or VOC-limited regime?

P2, l25-P3, l1: {Q} What were Zhang's conclusions (briefly)?

P3, l1-2: ADD {REFS} for the published studies here.

[Figure]

P3, l5: {REF} is not in your final reference list.

P3, l14-15: HO2 and OH are radicals not ions, please correct this.

P3, l22: fix {REF}, you mean Honninger et al., 2004 right?

P4, l6: I would call it the Beer-Lambert Law

P4, l9: {Q} Were clouds a factor? How often was it cloudy? Was the data pre-screened in any way?

Figure 1: Change the colour red on your figure, it is hard to read.

P5, l21: ...some point sources (e.g. XX and YY). Add some key examples, factories or power plants?

Figure 2: fix the text on your figure (e.g. spectrograph as one word)

P5, l24: {C} change stepping motor to stepper motor?

P6, l2: {Q} Why was the temp set to 20C?

P6, l10: replace scanning times with SCANS

Figure 3: replace a1, a2, etc. with a3 a30 a90 etc. {you don't need to number each one, simply add elevation angles to the alpha directly}

Table 1: fix text .. Longitude – one word, {Q} What is the MAYA? Is that Ocean Optics? If so, add that.

P6, l9: replace Doasis with DOASIS

P6, l22: replace [derived] with [measured] {as you did in your Figure4}

Figure 6 shows the period of 3-8 November. Why didn't you use the period of 3-12 November (the whole APEC period)?

P8, l11: You describe 2 peaks on Nov 4 and Nov 7, but what about Nov 3, as seen on

Figure 6 that actually has the HIGHEST HCHO VCDs?

Figure 7: perhaps replace UTC time with LT for consistency.

Figure 9: Error bars equal retrieval error. {Q} How is this calculated?

P10, l19-20: Is there any way to determine which is more important, the control measures of the meteorology? Perhaps a longer term study? Please comment.

P11, l7-8: Could this have to do with a change in NOx-limiting vs. VOC-limiting cases? Please advise.

P11, l14: Where was the surface O3 measurement location exactly? What type of NO2 VCD was it, geo-approximated, same instrument and location? Please describe.

P11, 118: Too many significant figures!

P12, l124: What is the assumed temp in the model for Dec 1, 2014 then?

P13, l16-20: Briefly state what associated errors clouds could pose. In l19 you say a slight variety (variation), give an error estimate please.

P14, l16: Where does this number come from and what dates? It is not the same as Figure 12 and it is not mentioned anywhere else in your paper. Is it a typo? Please advise.

P14,l20: Why the range? Grid sizes, I assume?

---

## Author Comment (AC1) · 13 Dec 2018

Dear editors and reviewers, Thank you very much for your constructive comments and advices on our manuscript. Your positive evaluation and comment encourage us and would be great helpful to our research. We have carefully considered every comment, and made corresponding revisions in the revised manuscript and marked every change in red.

Point to point response is following:

General Comments: 1ãĂĄHowever, a problem that detracts from the entire paper is

that the language is not fluent and precise. There are frequent spelling and grammatical errors (fragment sentences, unnecessary words, incorrect verb tense, missing articles, convoluted or run-on sentences). The authors are strongly suggested to engage the help of an English editor. Response: Thank you very much for your suggestions. We have considered your advice, and asked for help from an English language service.

2ãĂĄA few sections would benefit from re-structuring for increased logical flow and clarity. The scientific methods and assumptions are not sufficiently clearly outlined. The methodology section is disorganized and needs additional technical details. The paper does present some novel data and reaches substantial conclusions, but the results are sometimes not enough to support the interpretations and conclusions without further statistical analysis and/or expanded discussion. When discussing results, the authors must consider whether trends and differences in measured values they are interpreting are statistically significant given calculated or expected uncertainties. The authors must also try to place their conclusions within the context of previous literature (e.g., presented in the introduction). Response: Thank you for your advice. The error budgets are added in sec. 2.3. The section 2 has been reorganized. The introduction has also been rewritten.

Specific Comments: 1ãĂĄFor all regressions, the coefficient of determination (R2) statistic may be more appropriate since this value indicates the variance in the dependent variable that is predictable from the independent variable. Response: Thank you very much for your suggestion. We have considered your advice, and all coefficient of determination R are changed to R2 in this paper, including in the figures.

2ãĂĄPage 2 Line 3 What specific results had a good correlation coefficient? VCDs of HCHO? Other? Response: Thank you for pointing out. It's the VCDs of HCHO. It is changed to make it clear. Changes in manuscript: The HCHO VCDs of the CAMS model and MAX-DOAS were generally consistent with a correlation coefficient R2 greater than 0.69.

3ãĂĄPage 2 Line 23 Did the emissions decrease from 100% to these values or are these values the quantity of the decrease? It is not clear Response: Thank you for your advice. These values indicate the proportion of emissions from the corresponding source. It is changed to make it clear. Changes in manuscript: The analytical results of the Chemical Mass Balance (CMB) model showed that the contributions of coal-fired boilers, dust, and motor vehicles to PM2.5 in Beijing were around 2%, 7%, and 30%, respectively, during the APEC summit (Cheng et al., 2016).

4ãĂĄPage 2 In general, since some of the pollutants were measured by multiple papers that you cite, consider sorting this paragraph by pollutant rather than by author. Otherwise, it becomes repetitive and confusing to have to keep referring to the values from the previous papers earlier in the paragraph. Response: Thank you for your advice. We have considered your advice, and it is changed in the paper. Changes in manuscript: Presently, many studies have analyzed the effects of emission reduction measures during the APEC summit. Ground-based observations were taken to investigate the air quality changes associated with a series of stringent emission-reduction measures (Fan et al., 2016; Li et al., 2016; Liu et al., 2016; Tang et al., 2015; Chen et al., 2015; Wang et al., 2016a; Wang et al., 2016b; Wang et al., 2017a). Wang et al (2016a) selected five representative in situ stations in different locations in Beijing and found that average concentrations of SO2, NO2, PM10, and PM2.5 decreased by 61.5%, 40.8%, 36.4%, and 47.1%, respectively, whereas the average concentration of O3 increased by 101.8%, compared with the same period over the last five years (PM2.5 since 2013). O3 in urban and suburban areas of Beijing is mostly in the control area of volatile organic carbons (VOCs). The possible reason for the increase in O3 is that the emission control measures of NOx are greater than the emission control measures of VOCs, which leads to the weakening of the inhibition of O3 formation by NOx, resulting in significant increases in O3 concentration. Although the traffic and urban stations produce a lot of pollution due to motor vehicle emissions, the NO2 concentrations of suburban and regional stations significantly dropped compared with the traffic and urban stations as a result of the control measures. The NO2 emitted by motor vehicles in the Beijing urban area remained high even under the measures taken to limit the number of vehicles (Wang et al., 2016a). Space observations were also used to evaluate the effect of emission control measures on the changes in NO2 tropospheric vertical column densities (VCD) and aerosol optical depth (AOD) in Beijing and surrounds based on Ozone Monitoring Instrument (OMI) and Moderate Resolution Imaging Spectroradiometer (MODIS) retrieval. The results showed that NO2 VCD and AOD were mostly reduced by 47% and 34% in Beijing, respectively (Huang et al., 2015; Wei et al., 2016; Meng et al., 2015). The analytical results of the Chemical Mass Balance (CMB) model showed that the contributions of coal-fired boilers, dust, and motor vehicles to PM2.5 in Beijing were around 2%, 7%, and 30%, respectively, during the APEC summit (Cheng et al., 2016). Zhang et al (2017) analyzed the characteristics of aerosol size distribution and the vertical backscattering coefficient profile during the 2014 APEC summit using lidar observation. Particles with larger sizes were better controlled during the APEC period, with the number concentration of accumulation mode and coarse mode particles experiencing more significant decreases of 47% and 68% (Zhang et al., 2017). Published studies have focused mainly on the effects of commonly measured gas pollutants, particulate matter, and aerosols, but not HCHO (Cheng et al., 2016; Fan et al., 2016; Huang et al., 2015; Li et al., 2016; Liu et al., 2016; Meng et al., 2015; Tang et al., 2015; Chen et al., 2015; Wang et al., 2016a; Wang et al., 2016b; Wang et al., 2017a; Wei et al., 2016).

5ãĂĄPage 3 Line 6 Three sources are listed despite elsewhere in the paper it is stated that there are two sources, which one is it? Response: Thank you very much for your reminding. There are mainly two sources for troposphere formaldehyde, besides, only a small fraction of HCHO is from direct emissions of biogenic sources (e.g., vegetation). Changes in manuscript: Troposphere formaldehyde mainly originates from two sources.

6ãĂĄPage 3 This discussion may benefit from writing out some of the most important chemical equations for the reader equations for the reader. Response: Thank

you for your suggestion. Accordingly, equations are added in the paper. Changes in manuscript: All of the photolysis equations of HCHO to form OH radical at wavelengths below 370 nm are listed as follows: $HCHO + h\nu \rightarrow H+ HCO(\lambda \leq 370nm) \rightarrow H2 +CO$ (1) $H +O2 \rightarrow HO2$ (2) $HCO + O2 \rightarrow HO2 + CO$ (3) $HO2 +NO \rightarrow OH + NO2$ (4)

7ãĂĄPage 3 Line 16 It is unclear whether the importance of quantifying HCHO is to track emissions of VOCs or NVOCs or the generation of OH, or all of these. Response: Thank you for your suggestion. It represents all of these. And it is added to make it clear. Changes in manuscript: Therefore, HCHO can reflect anthropogenic VOC emissions and VOC emissions through the fast production of short-lived NMVOCs. Identifying the major sources of HCHO is essential for quantifying the photolysis sources of OH and their contributions to aerosol formation and for effectively controlling photochemical pollution (Bauwens, et al., 2016; Chang, et al., 2016; Ling, et al., 2017; Ma, et al., 2016; Tanaka, et al., 2016).

8ãĂĄPage 4 Line 5 What unit of HCHO? VCDs? Response: Thank you for pointing it out. It is VCDs. And it is added in the paper. Changes in manuscript: In this study, we used the ground-based MAX-DOAS instrument installed in Huairou District (suburban area) of Beijing to evaluate the effects of the sources and depositions of HCHO VCDs and their relations with emission control measures and meteorological conditions during the period from October 26, 2014 to November 20, 2014.

9ãĂĄPage 4 Line 10 The last paragraph of your introduction may benefit from explicitly stating your research objectives (perhaps as a list). What were all the components included in the spectral fitting for HCHO? Include what cross section reference spectra used (including author), fitting window etc. Response 1: Thank you very much for your advice. It is changed in the paper. Changes in manuscript: In this study, we used the ground-based MAX-DOAS instrument installed in Huairou District (suburban area) of Beijing to evaluate the effects of the sources and depositions of HCHO VCDs and their relations with emission control measures and meteorological conditions during the period from October 26, 2014 to November 20, 2014. Two pollution episodes and

their relationships with meteorological conditions were analyzed during APEC to evaluate the effects of regional transport and local emissions. Afterwards, three episodes, defined as "pre-APEC," the period of APEC and "post-APEC," were used to evaluate the influences of emission control measures on the changes in HCHO VCD during APEC. The correlations between HCHO VCDs with NO2 VCDs and O3 were used to determine the main HCHO sources and evaluate the dominant error sources of HONO simulations of Copernicus Atmosphere Monitoring Service (CAMS) model.

Response 2: Thank you very much for your suggestion. The spectral fitting for HCHO including cross section, reference spectra and fitting window is introduced in detail in section 2.1 (MAX-DOAS Methodology). And the information should belong to technique detail, so I think it is more appropriate to be written in section 2 than in the introduction. Changes in manuscript: Fig. 2 shows a structural representation of the MAX-DOAS system. This system comprises a telescope, stepper motor, spectrometer, and computer. Sunlight is focused by the telescope, which is installed outdoors and reaches the spectrometer through an optical fiber. The spectrometer was placed in a temperature-controlled box at 20°C to ensure that the spectrograph could work at a stable temperature under the changing ambient temperature from -15°C to 30°C in China. The spectrometer was produced by Ocean Optics and was named Maya (https://oceanoptics.com/product/maya2000-pro-custom/). The spectrometer covers the range of 290 nm to 420 nm, and its instrumental function is approximated as a Gaussian function with a full width at half maximum (FWHM) of 0.5 nm. MAX-DOAS was routinely operated for 24 h. Due to the intensity of the sunlight, only the daytime measurements were used for analysis. The nighttime measurements could be used to correct the dark current and offset. The azimuth angle view of the telescope was fixed at 0° (North) during the entire observation period. A full MAX-DOAS scan comprises six elevation angles (EA) (3, 5, 10, 15, 30, and 90°) and lasts for approximately 10 min (see Fig. 3). Each measurement had an average of 100 SCANS, and the integration time was adjusted automatically based on the light intensity. Table 1 lists the detailed setup of the MAX-DOAS instrument.

10ãĂĄSection 2.1 Please list the final equation used for calculating VCDs from fitted DSCDs given your geometric approximation. Response: Thank you for your advice. It is added in the paper. Changes in manuscript: Then the tropospheric VCD can be obtained from the following equation:

11ãĂĄReorganize section 2 so that sections 2.1, 2.3, and 2.4 are grouped together for a more logical sequence flow. What is included in the VCD error calculation? Response: Thank you for your advice. It is changed in the paper. The original sections 2.2 "Monitoring locations" and 2.3 "MAX-DOAS instrument and measurement" are grouped together as the new section 2.1 "Monitoring locations and instrument". The original sections 2.1 "MAX-DOAS Methodology" and 2.4 "DOAS analysis" are grouped together as the new section 2.2 "DOAS Spectral retrieval and determination of troposphericVCD". The error budgets is added in the text, see Sec. 2.3 "Error budgets".

12ãĂĄAre any of the MAX-DOAS VCDs removed from the dataset due to cloud fraction? If so, what was the cut off cloud fraction value? Response: Thank you for your question. In this paper, the effects of different cloud coefficients on MAX-DOAS inversion VCDs and the HCHO VCDs from MAX-DOAS and CAMS model under different cloud coefficients are both compared. It is found that the cloud coefficient has negligible influence on it. During the entire APEC period, it is basically sunny and cloudless weather. It is added in the paper to make it clear. The data is pre-screened. The spectrum with too small a light intensity and an excessive integration time are removed. The spectrum pre-screen is added in the section 2.3. Changes in manuscript: 1) The effects of different cloud coefficients on MAX-DOAS inversion VCDs and the HCHO VCDs from MAX-DOAS and the CAMS model under different cloud coefficients were compared. The results show that cloud coefficient had a negligible influence on the retrieval of HCHO VCDs by MAX-DOAS. Additionally, sunny and cloudless weather generally occurred during the entire APEC period. Thus, all of the data obtained in the different cloud coefficients were used.

2) We excluded data for solar zenith angle (SZAs) larger than 75° because of the

stronger absorptions of stratospheric species and a low signal-to-noise ratio. Data with a large root mean square (RMS) of the residuals and large relative intensity offset were also excluded.

13ãĂĄPage 5 Line 8 It is unclear what causes lower systematic errors. Response: Thank you for your question. This sentence is not very clear so that caused some misunderstandings for you. We make changes in the paper to make it clearer. 1) We mean that: According to Ma et al., 2013 and Wang et al., 2017c, they found the systematic error is larger for larger elevation angles and larger RAA. So this study uses the geometric approximation method to determine HCHO VCDs at an elevation angle of $15°$ to avoid surface obstacles on light paths along the line of sight, at the same time, it has lower systematic errors at $15°$ than $30°$. 2) And we add the discussion of the error budgets for geometric approximation in section 2.3. It also find the systematic errors at $15°$ is smaller than $30°$by using the geometrical approximation. Changes in manuscript: Lower systematic errors were achieved at $15°$ than at $30°$ by using the geometrical approximation (discussed in Section 2.3 below).

14ãĂĄPage 6 Line 18 Why was the FRS from this day chosen to fit all retrievals? Since you are fitting all your measured spectra against one FRS, you must consider the effect of SCD(FRS) (the component of trace gas in the FRS used) and the SCD(Solar Zenith Angle), which is the difference in the stratospheric component of SCD observed due to the difference in SZA between the times of measurement and the FRS time. SCD(SZA) changes with time and change the apparent diurnal trends. Please justify why you did not account for the SCD(SZA) and SCD(FRS). For example, was your FRS was obtained during a very low pollution period and/or are the stratospheric HCHO levels are expected to be trivial. For more information see Wagner, T., Ibrahim, O., Shaiganfar, R. and Platt, U.: Mobile MAX-DOAS observations of tropospheric trace gases, ATMO-SPHERIC Meas. Tech., 3(1), 129–140, 2010. Response: Thank you very much for your advice. Maybe my expression is not clear in the text, which makes you misunderstood. First, we use a FRS to retrieve all the spectra and obtain the dSCD. Then

I subtract the dSCD at 90° from the dSCD at off-zenith angles in the same elevation sequence to derive the delta SCD. The procedure can deduct the influence of stratospheric absorption and variation of instrumental properties. We add the description in the paper to make it clear. Changes in manuscript: The geometric approximation was used to convert the dSCD to the tropospheric VCD. In the first step, the differential slant column densities (dSCDs) were derived from the DOAS spectral analysis with a so-called FRS and measured in a small sun zenith angle at 90° elevation around noon (Hermans et al., 2003; Hönninger and Platt, 2002; Kraus, 2006). The SCD includes two parts of the absorption signal of the troposphere and stratosphere. To remove the interference of stratosphere absorption and variation in instrumental properties, dSCD at off-zenith elevation angles were subtracted by the dSCD at 90° elevation angle in the same elevation sequence to derive $\Delta$SCD following the equation below:

15ãĂĄPage 6 Line 23 What is the software reported SCD error in molec/cm2? Response: Thank you for your reminding. The SCD error is 8.14 × 1015 molec/cm2. Changes in manuscript: HCHO SCD was 7.21 × 1016 molecules cm-2 with an error of 8.14 × 1015 molecules cm-2.

16ãĂĄPage 6 Line 24 Are you missing units on this number? Response: Thank you for the asking. The root mean square of the residual error here is the residual spectral structures of optical depth, and there isn't unit for the optical depth. Changes in manuscript: The root mean square of the optical depth of the residual spectral structures was 1.08×10−3.

17ãĂĄPage 7 section 2.6 At what height were these meteorological parameters measured? At what time frequency before averaging? Response: The height of these meteorological parameters measured is about 15m. The weather station is about ten meters away from MAX-DOAS. All measured meteorological parameters are recorded of 1 min time intervals. Changes in manuscript: The MAX-DOAS instrument was deployed on the balcony (without a roof) of a classroom on the 4th floor in the laboratory building in the campus of UCAS (116.67°E, 40.4°N). The UCAS supersite is on the top

floor of the laboratory building, which is about 10 m away from the MAX-DOAS instrument. Meteorological parameters, including wind speed (WS), wind direction (WD), temperature (T), and relative humidity (RH), were continuously measured by a MetPak automatic weather station (Gill Instruments Ltd, Lymington, UK) at the UCAS superstation from October 28, 2014 to December 31, 2014 (Fig. 4a). All of the measured meteorological parameters were recorded at 1-min time intervals.

18ãĂĄPage 7 Line 24 What does a "static" weather situation mean? Response: The static weather means the wind speed less than 3.5 m s-1. Halfacre et al., 2014 defines that the wind speeds (mode of ∼3.5ms-1) is under relatively calm conditions (Halfacre, J.W., Knepp, T.N., Stephens, C.R., Pratt, K.A., Shepson, P., Simpson, W.R., et al.: Temporal and spatial characteristics of ozone depletion events, Atmos. Chem. Phys., 14: 4875–4894, doi:10.5194/acp-14-4875-2014, 2014.). It is defined in the paper to make it clear. Changes in manuscript: Halfacre et al (2014) defines the relatively calm conditions with wind speeds of less than 3.5 m s-1 as the static weather situation. The static weather situation frequently occurred during the observation with a wind speed of less than 3.5 m s-1, and wind speeds of more than 3.5 m s-1 usually appeared under northwest and west winds.

19ãĂĄBetter organization and flow in section2 may be achieved by describing MAX-DOAS methodology in this order: general description of the MAX-DOAS instrument, description DSCDs fitting, determination of VCDs from DSCDs, measurement sequence, and then viewing azimuth and location. Response: Thank you for your advice. It is changed in the paper. The original sections 2.2 Monitoring locations and 2.3 MAX-DOAS instrument and measurement are grouped together as the new section 2.1 Monitoring locations and instrument. The original sections 2.1 MAX-DOAS Methodology and 2.4 DOAS analysis are grouped together as the new section2.2 DOAS Spectral retrieval and determination of the troposphericVCD. Changes in manuscript: See section 2.

20ãĂĄFigure 6 . manuscript: oposphericVCDtion2.2 hen viewing azimuth and location.ion of Phys Response: All the days measured are quality controlled. The quality control criteria are added in the paper. Changes in manuscript: We excluded data for solar zenith angle (SZAs) larger than 75° because of the stronger absorptions of stratospheric species and a low signal-to-noise ratio. Data with a large root mean square (RMS) of the residuals and large relative intensity offset were also excluded.

21ãĂĄPage 8 Line 13 Please quantify in some way the relative change in solar radiation and temperature compared to the days were peaks were not apparent. Response: Thank you for your advice. It is added in the paper. Changes in manuscript: The daily averaged intensity of solar radiation and temperature on November 4 and 7 were compared with data from the two periods of November 4 to 7 and October 1 to December 31, 2014 (Fig. 8). The differences in averaged solar radiation and temperature on November 4 and 7 compared to the period from October 1 to December 31, 2014, were 2.22%, 2.47%, 34.4%, and -23.5%, respectively.

Figure 8: Averaged intensity of solar radiation and temperature in the four periods of November 4, November 7, November 3 to 8, and October 1 to December 31, 2014. Error bars denote the standard deviations.

22ãĂĄPage 8 Line 21 Please define (or find a better descriptor for) "good dispersion conditions". Response: Thank you for your advice. It is defined in the paper. Changes in manuscript: The value on November 6, 2014 was probably caused by the good dispersion conditions under the northwest winds with speeds of more than 3.5 m s-1, with the air mass mainly originating from the clean northwest area.

23ãĂĄPage 9 Line 14 What type of relevant pollution sources do these cities have? Primary and/orsecondary? Are there many industry and/or vehicular sources? Response: Thank you for your question. The type of relevant pollution source Tangshan, Baoding, Shijiazhuang and Tianjin is primary source. And there are a lot of industry in those cities. For Beijing, the main source is the vehicular source, especially in the urban area (Lin et al., 2009; Lin et al., 2012; Shao et al., 2006; Tang et al., 2015; Wei

et al., 2016). The reference is added in the paper (sec. 2.1). Changes in manuscript: Relevant pollution sources in Tangshan, Baoding, Shijiazhuang, and Tianjin is primary pollution sources. In Beijing, vehicles are the predominant pollution source, especially in the urban areas (Lin et al., 2009; Lin et al., 2012; Shao et al., 2006; Tang et al., 2015; Wei et al., 2016).

24ãĂĄPage 9 Line 16 You may want to state explicitly that there are fewer, smaller or less polluted cities in the Northern region here. Are the lower VCDs due to just dispersion or is it also chemical aging, et Response: Thank you very much for your advice. There are less polluted cities in the Northern region. The northern is clean with low VOC emissions, so there are few precursors of HCHO transported to the measurement station. The lower HCHO VCDs under such conditions are mainly due to less VOC precursors of HCHO. Changes in manuscript: The northern cities are clean with low VOC emissions, and thus few precursors of HCHO were transported to the measurement station in the north wind. The lower HCHO VCDs under such conditions are mainly due to fewer VOC precursors of HCHO. In summary, the wind from this area prominently contributes to the dispersion of the pollutants.

25ãĂĄPage 9 Line 17 and 18 Please explain the dependence of the VCDs on wind speed under different wind directions. Response: Thank you very much for your suggestion. It is added in the paper. Changes in manuscript: In terms of the dependence of HCHO on wind speed, the HCHO VCDs decrease along with the increasing wind speed under the northerly fast and clean wind, which results in the rapid dissipation of the pollution. Under the southerly wind, the HCHO VCDs increase with increasing wind speed. Thus, transport from the south polluted air to the observation site occurs more easily under southerly winds with relatively high wind speeds.

26ãĂĄPage 10 Line 6 What are the errors on each of the VCD values. Are they statistically different? Response: Thank you very much for your advice. According to the advice of review 2, we think it is more reasonable to use the standard deviation to represent the error bars. It is changed in the paper. And the figure 9 is changed to figure

11 due to some new figures were added. And we add the standard deviation on each of VCD values. It is added in the paper. Changes in manuscript: The average HCHO VCDs were 9.65×1015, 5.99×1015, and 8.65×1015 molec cm-2 before, during, and after APEC, with fitting errors of 9.39%, 10.12%, and 9.74%, respectively. 27ãĂĄ- Page 10 Line 17. This sentence is too vaguely written. Also, depending on whether the differences between the peak values are statistically significant, depending on the expected errors and the significance of the wind direction change, you may not have sufficient evidence to support this conclusion. Also consider that during APEC time the conditions were not only northerly winds but also higher wind speeds, which you state earlier in the paper tends to reduce the VCDs (which should be explained for clarity). Same comment for the sentence on lines 19 and 20. Response: Thank you very much for your advice. According to the question 43, we plot the VCD data against the wind speed and direction for the pre-APEC, during-APEC, and post-APEC periods, seperately. And we re-organized our discussion in the section 3.2. Please see the modified paragraph in the following . Changes in manuscript: As the measurement station is located in the northern suburban area of Beijing, the effects of the control measurements, which were mainly implemented in the urban areas, on HCHO were only observed at the station when dominant southerly winds occurred. We thus plotted the dependence of HCHO VCDs on the wind speed and directions in Fig. 12d–f for the pre-APEC, APEC, and post-APEC periods. Fig. 12d–f indicate that the averaged HCHO VCDs under south winds during APEC were about 6.46 × 1015 molec cm-2, which was considerably lower than 10.29, 6.46, and 9.20 × 1015 molec cm-2 in the pre-APEC and post-APEC periods. In addition the peak values due to transport from the south urban area on November 4, 2014 and November 7, 2014 during APEC shown in Fig. 11 were 25.75% and 18.3% lower than the peak values under similar wind fields in the pre-APEC and post-APEC periods. In general, the HCHO values under the dominant southerly wind field were considerably lower during APEC than the pre-APEC and post-APEC periods. The phenomenon implies that the control measures had a certain effect on reducing the concentration of HCHO. This suggests that the implementation

of control measures during the APEC summit reduced the concentrations of NO2 and aerosols (Liu et al., 2016; Zhang et al., 2017).

Figure 12: Wind roses in (a) the "pre-APEC", (b) the APEC, and (c) the "post-APEC" periods. Dependence of HCHO VCDs (1015 molec cm-2) on wind directions for different wind speeds in the pre-APEC (d), during the APEC (e), and post-APEC (f) periods.

28ãĂĄPage 10 Line 25 Some basic equations on HCHO chemistry in the introduction section would be very helpful for the reader by the time they get to this point in the paper. Response: Thank you very much for your suggestion. The equations are listed in the introduction, as the response to question 6. Changes in manuscript: Atmospheric photochemical reactions are related to the intensity of solar radiation as indicated in equation 1.

29ãĂĄPage 11 Line 3 Are you suggesting that this peak in the diurnal variation is due primarily to secondary production of formaldehyde rather than direct emissions? Since the most light is available mid-afternoon and local direct emissions are relatively smaller compared to secondary production? Please make this clearer to the reader. Response: Thank you for your question. Yes, we conclude this peak in the diurnal variation is due primarily to secondary production of formaldehyde rather than direct emissions. It is described in the paper. Changes in manuscript: Since most light is available in the early afternoon and local direct emissions are relatively smaller compared to secondary production, the secondary production of formaldehyde primarily caused the peak at 14:00.

30ãĂĄPage 11 Are your conclusions that the diurnal variability is driven by variation in light levels rather than diurnal variations in emissions? If light measurements are available, you could try correlating the light intensity with the VCDs. Response: Thank you for your advice . Anderson, et al., 1996; Lee, et al., 2015; Pinardi, et al., 2013 report that the increased HCHO at early afternoon implies that photo-oxidation of VOCs was very rapid due to the peak solar irradiance at this point in the day. We also compare the

correlating of the light intensity with the VCDs, but the result shows poor correlation. The reason should be that the lifetime of HCHO and the photochemical reaction rate of VOC to generate HCHO contribute to non linear dependence of HCHO VCDs on light intensity. In order to clarify this point, we modified the manuscript accordingly. Please see the modifications below.

Changes in manuscript: Since most light is available in the early afternoon and local direct emissions are relatively smaller compared to secondary production, the secondary production of formaldehyde primarily caused the peak at 14:00. The diurnal variation in VOC emissions could also play a role in the diurnal variation of HCHO. However, the typical life time of VOCs can reach several days. The diurnal variations in VOC emission are unlikely to change the abundance of atmospheric VOCs. Therefore, diurnal variation in photo reaction rate could be a dominant driving factor. Other smaller peaks appeared in the evening during another period of busy traffic (16:00–18:00 LT), which might be caused by primary pollution sources, e.g., exhaust fumes from vehicles. Thus, the diurnal variations in HCHO during all three episodes were similar to the typical patterns of secondary sources as reported in Anderson, et al (1996), Lee, et al (2015), Pinardi, et al (2013).

31ãĂĄPage 11 Line 5 Where can the reader see evidence of similar diurnal trends in the secondary sources? Response: Thanks for your question. Anderson, et al., 1996; Pinardi, et al., 2013 show typical HCHO diurnal variation of secondary sources, and my results are similar to what they reported. Changes in manuscript: Thus, the diurnal variations in HCHO during all three episodes were similar to the typical patterns of secondary sources as reported in Anderson, et al (1996), Lee, et al (2015), Pinardi, et al (2013).

32ãĂĄPage 11 Line 6 Many of the VCDs in the during, before and after APEC periods are equal within error. Are you referring to only the afternoon peaks HCHO values? The peak during APEC value appears to be equal within error with some of the highest post APEC values. Response: Thank you very much for your question. Many of the

VCDs in the before and after APEC periods are equal, but those all are higher than the value during the APEC. Please notes that the error bars in Fig. 11(Fig. 13 now) denote the standard deviation of HCHO VCDs, but not errors.

33ãĂĄPage 11 Line 7 What are the actual values with associated errors and are they statistically different? Response: Thank you very much for your question. The averaged VCD fitting errors of evening rush hours after and before APEC due to DOAS fit error here are 9.64% and 9.80%. Systematic error of the HCHO VCDs calculated by the geometric approximation is mostly smaller than 6% for the 15° elevation angle.

34ãĂĄPage 11 Line 8 Please explain your reasoning. Response: Thank you very much for your suggestion. After consideration, we decided to delete the conclusion that " The absolute HCHO values during the APEC period are obviously lower than those in the pre-APEC and post-APEC periods. The averaged HCHO during evening rush hours after APEC was higher than that before APEC. This finding is an interesting phenomenon, which may be related to some measures taken before the APEC.". Because this difference is small within the uncertainty range, the explanation is not reasonable.

35ãĂĄPage 11 Line 14 Where were the in-situ ozone measurements located relative to the MAX-DOAS measurements? Put this information in methodology. Response: Thank you for your advice. The MAX-DOAS instrument was deployed on the balcony (without a roof) of a classroom on the 4th floor in the laboratory building in the campus of UCAS (116.67°E, 40.4°N). And the UCAS supersite is on the top floor of the laboratory building, which is about ten meters away from MAX-DOAS. Ozone (O3) was measured by UV photometry (model 49i; Thermo Scientific), which is in the UCAS supersite. And I added the corresponding content in the article. Changes in manuscript: Sec.2.1: The MAX-DOAS instrument was deployed on the balcony (without a roof) of a classroom on the 4th floor in the laboratory building in the campus of UCAS (116.67°E, 40.4°N). The UCAS supersite is on the top floor of the laboratory building, which is about 10 m away from the MAX-DOAS instrument. Nitrogen oxide (NO, NO2, and NOx)

[Figure]

was measured by chemiluminescence (model 42i; Thermo Scientific), and ozone (O3) was measured by UV photometry (model 49i; Thermo Scientific). These gas analyzers had precision values of 0.5 ppb and 0.4 ppb, respectively.

36ãÅĄ Section 3.3 would benefit from a reorganization. Perhaps put information about primary versus secondary sources and the correlations first before making conclusions about diurnal trends. Response: We carefully think about the suggestion, but we think our current typesetting is more logical. Firstly, the source of HCHO was implied by the observation of diurnal variation of HCHO, and then the correlation analysis was used to further support the speculation.

37ãÅĄSection 3.4 How are VCDs calculated from the model output (i.e., what vertical height interval was integrated from the modeled vertical profile?) Section 3.4 Explain in more detail why the model poorly captures the local emissions. Could the lack of heterogeneous reaction in the model be contributing to the underestimation of the low HCHO values? Page 12 Line 14. Since the grid size seems to have little impact on the quality of the model output, is the "worse constraint" due to poor or outdated emission inventories local sources in this area? Are the highway emissions included in model calculations? How accurate is the emission inventory of the highway if it's included in the model? Response: The vertical discretisation uses 60 levels up to the model top at 0.1 hPa (65 km) in a hybrid sigma-pressure coordinate. The vertical extent of the lowest level is about 17 m; it is 100m at about 300m above ground, 400–600m in the middle troposphere and about 800m at about 10 km in height (Flemming, J., Huijnen, V., Arteta, J., Bechtold, P., Beljaars, A., Blechschmidt, A.-M., Diamantakis, M., Engelen, R. J., Gaudel, A., Inness, A., Jones, L., Josse, B., Katragkou, E., Marecal, V., Peuch, V.-H., Richter, A., Schultz, M. G., Stein, O., and Tsikerdekis, A.: Tropospheric chemistry in the Integrated Forecasting System of ECMWF, Geosci. Model Dev., 8, 975–1003, doi:10.5194/gmd-8-975-2015, 2015.). HCHO VCD is calculated by an integration of the modeled vertical profiles of HCHO. The model doesn't consider the heterogeneous reaction. So the lack of heterogeneous reaction in the model could contribute to the

underestimation of the low HCHO values. The actual emission totals for 2008 inventory including anthropogenic, biogenic and natural sources and biomass burning. So highway emissions are considered in the 2008 inventory. Due to the establishment of the UCAS from 2013 and the holding of APEC meeting in 2014, the economy near the UCAS has grown rapidly, and the traffic flow has increased significantly in recent years. Thus, it could underestimate the highway emissions by using the 2008 inventory. Changes in manuscript: The underestimation of the low HCHO values by the CAMS model compared to the MAX-DOAS measurements could be attributed to the lower constraint of local emissions in the model near the UCAS measurement station, and the lack of heterogeneous reactions in the model could also contribute to the underestimation of the low HCHO values. The China National Highway 111 is nearby and runs from north to south. The actual emission totals for the 2008 inventory included anthropogenic, biogenic and natural sources, and biomass burning, thus, highway emissions were considered in the 2008 inventory. However, due to the establishment of the UCAS from 2013 and the holding of the APEC meeting in 2014, the economy near the UCAS had grown rapidly, and the traffic flow had increased significantly in recent years. Thus, the use of the 2008 inventory could underestimate the highway emissions.

38ãĄPage 13 Line 8 When you say "the primary HCHO is dominant" do you mean that the dominant contribution to the HCHO VCDs is the "local" primary emissions of HCHO? Edit for further clarity. Response: Thank you very much for your advice. Yes, "the primary HCHO is dominant" mean that the dominant contribution to the HCHO VCDs is the "local" primary emissions of HCHO. It is changed in the paper to make it clear. Changes in manuscript: Thus, when the secondary source of HCHO is reduced, namely the "local" primary emissions of HCHO predominantly contribute to the HCHO VCDs, the difference between the MAX-DOAS observation and CAMS model is obvious.

39ãĄPage 13 Line 14 Your conclusion is not necessarily sufficiently supported given the small R2 value and "reasonably" (too vague) would have to be defined before it is

clear whether the data support this statement sufficiently. Response: Thank you for your advice. R represents the ratio of HCHO VCDs in the morning (8:00LT)and noon (14:00LT). If R from the model is close to obtained from the MAX-DOAS, it indicates that the trend of diurnal variation of HCHO from the model simulation and MAX-DOAS observation is consistent. In other word, therefore the model can reasonably simulate the systematic diurnal variation of HCHO. Changes in manuscript: R represents the ratio of HCHO VCDs in the morning (8:00 LT) and noon (14:00 LT). If RModel is close to RMAX-DOAS, it indicates that the trend in diurnal variation of HCHO from the model simulation and MAX-DOAS observation is consistent, suggesting that the model can reasonably simulate the systematic diurnal variation in HCHO.

40ãĂĄPage 14 Line 8 You may want to add that, in contrast, correlation with NO2 was lower and what that implies. If VCDs are calculated from the 10° and 30°spectra, how do the values compare to the 15° spectra VCDs? Given that the geometric approximation becomes worse under high aerosol conditions and these VCDs would be expected to diverge in that case, comparison with the 10° and 30°spectra may be a good measure of the validity of your use of the geometric approximation. Response: Thank you very much for your advice. In order to constrain the systematic error of the grometric approximation, we compare HCHO VCDs calculated with the geometric approximation with those retrieved using PriAM profile inversion algorithm. The discussion is added in the manuscript as following. Changes in manuscript: a. The systematic error of the HCHO VCDs calculated by the geometric approximation depends on the layer height of the TGs and aerosols. To evaluate the systematic error of the geometric approximation, we calculated more exact tropospheric HCHO VCDAMF using the PriAM inversion algorithm (Wang et al, 2017b). HCHO VCDgeo at elevation angles at 15° and 30° are usually obtained from the geometric approximation. The relative differences (Diff) between VCDAMF and VCDgeo for HCHO were calculated by Eq. (9):

In Fig. 6, the average relative differences for elevation angles of 15°and 30° are shown as a function of the effective cloud fractions (eCF), as 0<eCF $\leq$1, 0<eCF$\leq$0.3, 0.3<eCF

≤0.7, and 0.7<eCF≤1.0. The cloud fractions (eCF) are downloaded from the ECMWF CAMS model. It can be seen that the biases caused by the use of the geometric approximation are generally much smaller at EA=15° than at EA=30°, with the Diff being mostly smaller than 6% for the 15° elevation angle of and smaller than 16% for the 30° elevation angle in all periods. The bias for Diff caused by using the geometric approximation is about 2% (Ma et al., 2013; Wang et al., 2017c).

41ãĂĄFigure 11 Can you explain why the standard deviation of the pre-APEC time is so much smaller than the post-APEC period despite similar values? Response: Thank you very much for your question. As we can see in figure 11, there are two obvious pollution process after APEC. The HCHO concentration shows a significant lifting process, which makes the standard deviation large.

42ãĂĄFigure 9 Since you show average values, how did the standard deviations of the averages compare to the retrieval errors? Are the larger of the two plotted as error bars? Response: Thank you for your question. The standard deviations of the averages is larger than the retrieval errors. The fitting error is also added in the figure. Changes in manuscript:

Figure 13: Averaged diurnal variation in HCHO VCDs measured by MAX-DOAS in three episodes around APEC. The short cap width of the error bars denotes the one sigma standard deviations around the mean analysis values. The long cap width of the error bars denotes the fitting error.

43ãĂĄFigure 8 Why do moderate wind speeds appear to produce similar VCD values for all wind-directions. Also, why do southerly conditions appear to result in maximum VCDs occurred under the highest wind speeds given that you stated that high wind-speeds tend to reduce VCDs? Response: Thank you for your question. I mean that transport from the south polluted air to observation site is easier under southerly winds with high wind speed. Changes in manuscript: Under the southerly wind, the HCHO VCDs increase with increasing wind speed. Thus, transport from the south polluted air

to the observation site occurs more easily under southerly winds with relatively high wind speeds.

Can you divide the VCD data into wind-speed and time of day and then see if there is a statistically significant reduction of the VCDs under non-Southerly wind conditions during APEC compared to before and after? That may help to determine how much the emissions controls impacted the VCDs independent of wind-direction. Response: Thank you for your advice. We made the new plots following your suggestion (see attached figure) as Fig. 12 in the revised manuscript. We do not see the significant reduction of HCHO VCDs under non-southerly winds during APEC compared to before and after. However it is understandable. Because our station is in the north suburban area of Beijing city. The control measurements were mainly operated in the Beijing urban area. Therefore in order to evaluate the effects of control measures in the city, we need to compare HCHO observed at the suburban station under the southerly winds between different APEC periods, because pollutants in the city center can be transported to the measurement site under the southerly winds. Accordingly we modified the paragraph in Section 3.2 which has been given in the response to your point 27.

Figure 12: Wind roses in (a) the "pre-APEC" , (b) the APEC and (c) the "post-parade" periods. Dependence of HCHO VCDs (1015 molec cm-2) on wind directions for different wind speeds in the pre-APEC (d), during the APEC (e) and post-APEC (f).

Technical Corrections: 1ãĂĄTitle: Consider adding VCDs after the word formaldehyde. Consider also including the APEC study to the title. Response: Thank you very much for your suggestion. We have considered your advice, and it is added in the title. Changes in manuscript: Ground-based MAX-DOAS observations of tropospheric formaldehyde VCDs and comparisons with the CAMS model at a rural site near Beijing during APEC 2014

2ãĂĄGeneral technical comment: when listing VCDs to route, please include the error values. Response: I don't understand your meaning. Do you mean adding the error

after the VCDs in the text?

3ãĂĄPage 1 Line 22 Abstract: what are the units of HCHO and O3? VCDs? Mixing ratios? Response: Thank you very much for your question. The units of HCHO are VCDs, and the units of O3 are Volume mixing ratio. It is added in the paper. Changes in manuscript: Peak values of HCHO vertical column densities (VCDs) around noon and a good correlation coefficient R2 of 0.73 between HCHO VCDs and surface O3 concentration during noontime indicated that the secondary sources of HCHO through photochemical reactions of volatile organic compounds (VOCs) dominated the HCHO values in the area around UCAS.

4ãĂĄPage 2 Line 8 What were the specific dates of the conference? Response: Thank you very much for your suggestion. The specific dates of the conference are from November 5 to November 11, 2014. High emissions in Beijing and surround area were required to stop or limit their production during 3–12 November 2014. Changes in manuscript: The 2014 Asia-Pacific Economic Cooperation (APEC) conference was held in the Huairou District of Beijing from November 5–11, 2014.

5ãĂĄPage 2 Line 18 It is unclear what "traffic" and "regional" stations are? Response: Thank you very much for your question. Wang et al., (2016a) selected five representative in-situ stations in different locations of Beijing, which represents different emission types and backgrounds. The traffic station is located in the Xizhimen Station of city center in Beijing with heavy traffic and traffic flow. The regional station is located in the suburbs of Beijing to reflect the impact of urban development on the suburban environment.

6ãĂĄPage 3 lines 10 and 11 The meaning of this sentence is unclear. Response: Thank you very much for your remind. It is changed to make it clear in the paper. Changes in manuscript: Being a short lifetime oxidation product, long-living VOCs, such as methane (CH4), contribute to the background levels of HCHO (Pinardi et al., 2013; Stavrakou et al., 2009; Vrekoussis et al., 2010).

7ãĂĄPage 3 Line 24 I believe this should say tropospheric column densities, surface mixing ratios, and vertical profiles of aerosol extinction and trace gas mixing ratios. Response: Thank you very much for your remind. It is changed in the paper. Changes in manuscript: The information obtained from MAX-DOAS measurements includes tropospheric column densities, surface mixing ratios, and vertical profiles of aerosol extinction and trace gas mixing ratios.

8ãĂĄPage 4 Line 18 Consider changing to "derived from the DOAS spectral analysis [of the measured spectra]" Response: Thank you very much for your advice. It is changed in the paper. Changes in manuscript: In the first step, the differential slant column densities (dSCDs) were derived from the DOAS spectral analysis with a so-called FRS and measured in a small sun zenith angle at 90° elevation around noon (Hermans et al., 2003; Hönninger and Platt, 2002; Kraus, 2006).

9ãĂĄPage 6 Line 22 In this sentence and figure 4 you use different terms for the blue and red lines: blue (measured, derived ) red (calculated, retrieved, fitted). Response: Thank you very much for your suggestion. It is changed in the paper. Now figure 4 is changed to figure 5. Changes in manuscript: Figure 5: Example of a DOAS fit of a spectrum to retrieve the slant column densities of HCHO; the red and blue curves indicate the fitted absorption structures and the derived absorption structures from the measured spectra, respectively.

Pick one term for each and ensure that the meaning of the term is clear. What about the contribution of the residual to the blue line? Response: Thank you very much for your remind. The residual represents the remaining structure after the measured spectrum (blue line) minus the fitted absorption structures (red line). The smaller the residual, the better the spectral fit

10ãĂĄPage 10 Line 4 the sentence needs editing for greater clarity and to appropriately describe figure 9. Response: Thank you very much for your suggestion. It is changed in the paper. Changes in manuscript: The result shows a "fluctuating effect"

with the HCHO VCDs increasing abruptly over several days and dropping sharply for a few days during the APEC summit.

11ãĂĄPage 13 Line 24 HCHO were also studied before and after APEC, were they not? Response: Thank you very much for your suggestion. It is changed in the paper. Changes in manuscript: We studied the tropospheric HCHO VCDs at the UCAS site in Huairou District, Beijing around the APEC summit based on the MAX-DOAS measurements from October 1, 2014 to December 31, 2014.

12ãĂĄTable 1 There are small spacing and English errors. Response: Thank you very much for your remind. It is changed in the Table 1. Changes in manuscript:

13ãĂĄClarity of Figure 9 may be improved by lines or boxes that indicate the afternoon period. Response: I think maybe you give the wrong number of the figure.

14ãĂĄFigure 6 If relative humidity is not discussed in the results or discussion, perhaps remove it from the figure to have more space to expand the more relevant data. Response: Thank you very much for your suggestion. We have considered your advice, and it is removed in the Figure 6. Changes in manuscript:

15ãĂĄSupplement: More helpful analysis may be achieved by dividing the regressions into bins that do not all include zero cloud fraction. For example, are different trends observed for eCF 0-0.3, 0.3-0.5, 0.5-0.7 etc.? Response: The figure followed shows the result of dividing the regressions into bins of eCF. And the figure also supports the conclusions in our text. The figure is added in the supplement. Changes in manuscript:

Figure S3: Correlation between HCHO VCDs retrieved from the MAX-DOAS measurements and those obtained from the CAMS model data for $0<eCF \leq1$ (a), $0<eCF\leq0.3$(b), $0.3<eCF \leq0.7$ (c), and $0.7<eCF\leq1.0$ (d) at 8:00 LT from October to December 2014.

Figure S4: Correlation between HCHO VCDs retrieved from the MAX-DOAS measurements and those obtained from the CAMS model for $0<eCF \leq1$ (a), $0<eCF\leq0.3$(b),

0.3<eCF $\leq$0.7 (c), and 0.7<eCF$\leq$1.0 (d) at 14:00 LT from October to December 2014.

Thank you for taking care of our manuscript.

Kind regards, Xin Tian E-mail: xtian@aiofm.ac.cn

Corresponding author : Pinhua Xie, Jin Xu, Yang Wang E-mail address: phxie@aiofm.ac.cn; jxu@aiofm.ac.cn; y.wang@mpic.de

Please also note the supplement to this comment:
https://www.atmos-chem-phys-discuss.net/acp-2018-440/acp-2018-440-AC1-supplement.pdf
* * *
[Figure]

Response to comment 21:

[Figure]

**Figure 8: Averaged intensity of solar radiation and temperature in the four periods of November 4, November 7, November 3 to 8, and October 1 to December 31, 2014. Error bars denote the standard deviations.**

Response to comment 27:

[Figure]

**Figure 12: Wind roses in (a) the "pre-APEC", (b) the APEC, and (c) the "post-APEC" periods. Dependence of HCHO VCDs ($10^{15}$ molec cm$^{-2}$) on wind directions for different wind speeds in the pre-APEC (d), during the APEC (e), and post-APEC (f) periods.**

**Fig. 1.**

**Supplement:**

**Supplement**

| effective cloud fractions (eCF) x | Number of days and date |
|---|---|
| 0＜x≤0.1 | Number of days: 29

(October 3, 5, 11, 12, 13, 15, 16, 23;

November 2, 3, 12, 13, 14;

December 1, 3, 4, 5, 7, 9, 10, 11, 14, 15, 16, 21, 24, 25, 29, 30) |
| 0.1＜x≤0.3 | Number of days: 10

(October 6, 17, 21, 22, 27;

November 22, 27;

December 1, 20, 31) |
| 0.3＜x≤0.7 | Number of days: 23

(October 2, 7, 14, 18, 28;

November 1, 4, 5, 8, 9, 10, 18, 23, 25, 30;

December 2, 6, 12, 17, 19, 22, 23, 28) |
| 0.7＜x≤1 | Number of days: 30

(October 1, 4, 8, 9, 10, 19, 20, 24, 25, 26, 29, 30, 31;

November 6, 7, 11, 15, 16, 17 19, 20, 21, 24, 26, 29;

December 8, 13, 18, 26, 27) |

**Table S1: Number of days and dates corresponding to different effective cloud fractions from October 1, 2014 to December 31, 2014.**

[Figure]

**Figure S1: Correlation between HCHO VCDs retrieved from the MAX-DOAS measurements and those obtained from the CAMS model data for 0<eCF ≤1 (a), 0<eCF≤0.1(b), 0<eCF ≤0.3 (c), and 0<eCF≤0.7 (d) at 8:00 LT from October to December 2014.**

[Figure]

**Figure S2: Correlation between HCHO VCDs retrieved from the MAX-DOAS measurements and those obtained from the CAMS model for 0<eCF ≤1 (a), 0<eCF≤0.1(b), 0<eCF ≤0.3 (c), and 0<eCF≤0.7 (d) at 14:00 LT from October to December 2014.**

[Figure]

**Figure S3: Correlation between HCHO VCDs retrieved from the MAX-DOAS measurements and those obtained from the CAMS model data for 0<eCF ≤1 (a), 0<eCF≤0.3(b), 0.3<eCF ≤0.7 (c), and 0.7<eCF≤1.0 (d) at 8:00 LT from October to December 2014.**

[Figure]

5    **Figure S4: Correlation between HCHO VCDs retrieved from the MAX-DOAS measurements and those obtained from the CAMS**

**model for 0<eCF ≤1 (a), 0<eCF≤0.3(b), 0.3<eCF ≤0.7 (c), and 0.7<eCF≤1.0 (d) at 14:00 LT from October to December 2014.**

---

## Author Comment (AC2) · 13 Dec 2018

Dear editors and reviewers, Thank you very much for your constructive comments and advices on our manuscript. Your positive evaluation and comment encourage us and would be great helpful to our research. We have carefully considered every comment, and made corresponding revisions in the revised manuscript and marked every change in red.

Point to point response is following:

Major (1) Use of the Geometric Approximation Only P5, 8: Your comment, ": : :it has

lower systematic errors because of the geometrical approximation" needs to be backed up. Do you have a reference, maybe a short explanation of your reasoning? You need to somehow prove to me that it is better to use a geometric approximation of the VCD vs. one of several RTM/inversions approaches. You did not do this here, but rather allude to some other studies. I am not convinced that the geometric approximation is the best? Prove me wrong? Response: Thank you for your question. This sentence is not very clear so that caused some misunderstandings for you. We make changes in the paper to make it clearer. 1) we mean that: According to Ma et al., 2013 and Wang et al., 2017c, they found the systematic error is larger for larger elevation angles and larger RAA. So this study uses the geometric approximation method to determine HCHO VCDs at an elevation angle of 15° to avoid surface obstacles on light paths along the line of sight, at the same time, it has lower systematic errors at 15° than at 30°. 2) And we add the discussion of the error budgets for geometric approximation in section 2.3. It also shows the systematic errors at 15° is smaller than at 30°by using the geometrical approximation. Changes in manuscript: Lower systematic errors were achieved at 15° than at 30° by using the geometrical approximation (discussed in Section 2.3 below).

Alternatively, you could provide a comparison of your VCDs with RTM-inversion derived HCHO VCDs. I would also suggest that you provide a comparison of your HCHO VCDs to those measured via satellite. This would give me more confidence in your conclusions Response: Thank you very much for your suggestion. We have considered your advice, and I add the comparison of geometrical approximation with inversions approaches at section 2.3. Changes in manuscript: 2.3 Error budgets The following error sources were considered as the error estimates for the MAX-DOAS results: a. The systematic error of the HCHO VCDs calculated by the geometric approximation depends on the layer height of the TGs and aerosols. To evaluate the systematic error of the geometric approximation, we calculated more exact tropospheric HCHO VCDAMF using the PriAM inversion algorithm (Wang et al, 2017b). HCHO VCDgeo at elevation angles at 15° and 30° are usually obtained from the geometric approximation. The

relative differences (Diff) between VCDAMF and VCDgeo for HCHO were calculated by Eq. (9):

In Fig. 6, the average relative differences for elevation angles of 15°and 30° are shown as a function of the effective cloud fractions (eCF), as 0<eCF ≤1, 0<eCF≤0.3, 0.3<eCF ≤0.7, and 0.7<eCF≤1.0. The cloud fractions (eCF) are downloaded from the ECMWF CAMS model. It can be seen that the biases caused by the use of the geometric approximation are generally much smaller at EA=15° than at EA=30°, with the Diff being mostly smaller than 6% for the 15° elevation angle of and smaller than 16% for the 30° elevation angle in all periods. The bias for Diff caused by using the geometric approximation is about 2% (Ma et al., 2013; Wang et al., 2017c). b. The fitting error of the DOAS fit is derived from the dSCD fitting error to VCD error by using geometric approximation, as

and the hourly average of the HCHO VCD fitting error was from 3.61% to 27.19% for the entire period. c. Cross section error also constitutes one of the error sources. Some previous research reported that cross section errors of O4 (aerosols) and HCHO are 5% and 9%, respectively (Bogumil et al., 2003; Meller and Moortgat, 2000;Thalman and Volkamer, 2013; Vandaele et al. 1998 ). Wang et al (2017b) estimated the errors related to the temperature dependence of the cross sections, and the corresponding systematic error of HCHO was estimated to up to 6%. Since the three errors are mainly independent, the total error can be calculated by combining all the above error sources, adding up to about 12% on average.

(2) Emission Totals from 2008 for CAMS model P7, l12: You use emission totals from 2008. Your year of study is 2014, that is a difference of 6 years and a lot can change. Why didn't you use a more recent emission inventory? Is there one? If so, why didn't you use it? Response: Thank you for your question. The simulation work of the model is made by the ECMWF, and we download the data from CAMS real-time products in ECMWF (http://apps.ecmwf.int/datasets/data/cams-nrealtime/levtype=sfc/). Besides, the MAX-DOAS data can verify the model. We also make the conclusion that inventory needs to be updated according to our comparative study. Annual emissions from anthropogenic, biogenic and natural sources and biomass burning for 2008 in Tg for a composition Integrated Forecasting System (C-IFS) (CB05) run at T255 resolution (Flemming et al., 2014). The 2008 global emissions is used as a total amount of emissions to assimilate data in the C-IFS model. Then, for their near real-time data, they will be added to the latest satellite observation data for assimilation.

(3) HCHO VCD error P9, l11-l20: It is likely that when the wind comes from the south it is more polluted than when the wind comes from the north. However, an average HCHO VCD of 7.57*1015 vs. 6.64*1015 is hardly conclusive. This is a 14% difference. WHAT IS THE ERROR OF YOUR VCD? I would estimate that is it a least 10%, likely over 20%. As such, your statistics here are weak. Please define the error of your VCDs and then re-word this section. For example, in Figure 9, you have error bars on your VCDs, but no mention of how you calculate them [they also look very low to me]. Response: Thank you for your advice. Although the uncertainty of HCHO VCD is about 6% for the 15° elevation angle, the uncertainty is comparable to the systematic difference of HCHO under different wind fields. However, uncertainty effects on systematic bias can be averaged as zero for a long-term measurements, therefore the systematic differences of HCHO VCDs still considerably indicate that more pollutants are transported from the southern region.

(4) Figure 12 – Correlation Analysis I understand what you are trying to achieve here. However, I am not sure why you choose the period Oct 26 – Nov 20, 2014? This seems random? Why not use all your data? Response: Thank you for your question. Because in the previous study around APEC, the period used was from October 26 to November 20, 2014, so the analysis here we used the corresponding period. However we have considered your advice, and I use all the data for the correlation analysis. The change is made in the paper. The new correlation analysis indicates that the correlation coefficients between HCHO VCD and NO2 VCD at rush hour and between HCHO VCD and O3 during the noon time are slightly reduced. However, the results

still show high correlation between HCHO VCD and O3 during the noon time and low correlation between HCHO VCD and NO2 VCD at rush hour.

The NO2 VCD is not described. Is it VCDgeo? Is it data from the same instrument and time? Did you also compare your O3 with the 7-9 & 16-18 periods? You don't have to show the plot but I would like to know the R of that? Hopefully it is very low to prove your point. Similarly, did you compare the NO2 VCDs with the HCHO VCDs from 11-14 period. You need a more complete assessment here to really prove your point. Response: Thank you for your advice. The NO2 VCD is VCDgeo and the data is from the same MAX-DOAS instrument and time. All the suggested comparisons are added in the paper and please see the following changes. Changes in manuscript: Determining pollution sources is crucial to controlling air pollution. Three time intervals were used for determining the main HCHO sources. The first interval was defined as noontime from 11:00–14:00 and is associated with strong photochemical reactions. The second and third intervals were defined as the morning rush hour from 7:00–9:00 and the evening rush hour from 16:00–18:00. To further determine whether the pollution sources of HCHO at UCAS were primary or secondary formations from other VOCs, the correlations of HCHO with the primary pollutant NO2 or secondary pollutant O3 were analyzed (Anderson et al, 1996; Possanzini et al., 2002). Surface O3 data were obtained from in situ measurements in the UCAS supersite, and troposphere NO2 VCD data were retrieved from the same MAX-DOAS measurements using geometric approximation. The linear correlations of noontime average HCHO VCD with NO2 VCD and O3 from 11:00–14:00 and rush hour average HCHO VCD with NO2 VCD and O3 from 7:00–9:00 and 16:00–18:00 are shown in Fig. 14. Direct analysis of the data indicates that noontime average HCHO had a higher correlation coefficient with NO2 VCD and O3 than rush hour. This implies that a small amount of HCHO comes from the traffic emissions during rush hour. A good correlation coefficient R2 of 0.73 was found between HCHO VCD and O3 during the noontime, which indicates that the main source of HCHO was from secondary photo-oxidation formation at noon. In contrast, a correlation coefficient of 0.38 between HCHO VCD and NO2 VCD during noontime was

better than during rush hour (R2=0.06), which may be due to the contribution of vehicle emissions to HCHO precursors. A longer NO2 lifetime with less dispersion efficiency in winter and HCHO from continuously generated photo-oxidation contributed to the higher correlation between HCHO VCD and NO2 VCD at noon higher than during rush hour. The transport of NO2 and VOC may constitute one of the causes. The VOCs from transport generate HCHO due to strong photo-oxidation at noon. This result indicates that secondary photo-oxidation formation of HCHO from other VOCs should be the dominant source at UCAS.

Figure 14: Scatter plots and linear regressions (a) of noontime average HCHO VCD measured by MAX-DOAS against O3 VMRs measured by a stationary ozone monitoring instrument, and (b) rush hour average HCHO VCD against NO2 VCD measured by MAX-DOAS from October to December 2014.

What happens if the R value for O3 and 7-9&16-18 periods is also high? I believe you have something here but be careful about how you present it. I also need to know exactly where your O3 monitor is, is it at ground-level? Response: Thank you for your question. The R2 value for O3 and HCHO at 7-9&16-18 periods is 0.03. The MAX-DOAS instrument was deployed on the balcony (without a roof) of a classroom on the 4th floor in the laboratory building in the campus of UCAS (116.67°E, 40.4°N). And the UCAS supersite is on the top floor of the laboratory building, which is about ten meters away from MAX-DOAS. Ozone (O3) was measured by UV photometry (model 49i; Thermo Scientific), which is in the UCAS supersite. And we add the corresponding content in the revised manuscript.

(5) Assumption that the HCHO VCD is the correct result On P12, l3 you state that the CAMS model UNDERESTIMATES : : :. How do you know this? How do you know the MAX-DOAS result is the correct result and better than the CAMS model? What other VALIDATION do you have? Did you compare it to the satellite data; ground-data extrapolated to a column {see comment 1}? You may be right, but you may also be wrong. I am not convinced, especially without any error analysis of your HCHO VCDs or CAMS

model. I would say that your CAMS model could be really off since it uses emission totals from 2008. Maybe the emission estimates in the model for 2008 are simply much lower than the 2014 values? You allude to this on P12, l14-15, right? Response: Thank you for your question. According to your advice, we evaluate the systematic error of the geometric approximation by comparing the VCD calculated using the geometric approximation and those retrieved using a PriAM profile inversion algorithm.. The new discussion is added in Section 2.3. The result shows that the systematic error is less than 6% for the elevation angle of 15 degrees. Besides, satellite retrievals of HCHO have more problem than MAX-DOAS measurements. MAX-DOAS is an usual technique to validate the HCHO satellite data (cite: De Smedt, I., Stavrakou, T., Hendrick, F., Danckaert, T., Vlemmix, T., Pinardi, G., Theys, N., Lerot, C., Gielen, C., Vigouroux, C., Hermans, C., Fayt, C., Veefkind, P., Müller, J.-F., and Van Roozendael, M.: Diurnal, seasonal and long-term variations of global formaldehyde columns inferred from combined OMI and GOME-2 observations, Atmos. Chem. Phys., 15, 12519-12545, doi:10.5194/acp-15-12519-2015, 2015.). In addition MAX-DOAS retrievals of HCHO have been well proved and evaluated in the previous study. Wang et al., 2017b retrieved tropospheric HCHO VCDs and vertical profile in Wuxi from 2011 to 2014, and the DOAS fit setting derived from the formaldehyde slant column measurements during CINDI: intercomparison and analysis improvement. Therefore MAX-DOAS results of HCHO are valuable and sufficiently confident to be used for validation of model simulations. For the old emission inventory, the inventory is used by the operational CAMS model. We agree it could be lower than the current emission. The conclusion is also our finding by comparing MAX-DOAS measurements with the model data.

P12, l3: What do these ranges mean? Is it due to different grid-sizes? Response: Thank you for your question. These ranges are due to the different grid-sizes. We do some change in the paper to make it clear. Changes in manuscript: On average, the CAMS model underestimated HCHO VCDs by 1.56–2.02 × 1015 molec cm-2 and 1.27–2.12 × 1015 molec cm-2 compared to the MAX-DOAS measurements at 8:00 LT and 14:00 LT, respectively, due to different grid-sizes.

(6) RMAX-DOAS vs Rmodel P13, l10: You R concept is interesting. Based on this I would think that R(DOAS) should be higher than R(model) for cases when the temp is cold (and secondary is HCHO is lower than predicted via the model), do you see this? Alternatively, if primary HCHO emissions are under predicted in the model R(DOAS) again would be higher than R(model) right? So what does this R concept really tell us? A graph like Figure 15, does not tell me much? However, if you separate out case studies maybe you get some more information. P14, l21-23: If the CAMS model underestimates primary sources of HCHO then R(DOAS) > R(model) but "under a situation with a low temperature when the production rate of secondary HCHO is relatively low" won't the CAMS model also underestimate the secondary HCHO production also causing R(DOAS) > R(model) as well? What is the assumed temp in the model, or does it use real-time met-data? How do we know what is the problem, is it a problem with the assumed temp, if so can you adjust that to check? OR is it a problem with the emissions inventory (perhaps a bigger issue). Again, the above concept seems to have merit, but you need to develop this and explain it further, because I am somewhat confused. Also, despite your analysis I have no feeling as too how much HCHO is secondary and how much is primary (and isn't that what the R calculations are for?). Response: Thank you for your suggestions. Here are some explanations for your questions. We agree on your conclusion, if there is big bias in the model simulations of the secondary production of HCHO, it can also cause deviations of R(DOAS) and R(model). Following your suggestion: 1) we separate the plots in the periods of October to November and for December (see below). But there is no significant difference between the two periods.

Scatter plots and linear regression of RModel against RMAX-DOAS (see the text) from October to December 2014 (a) and from October to November and for December due to the changing of temperature (b). There are not significant differences between the two periods.

2) we check the source of meteorological data in the model. The CAMS global real-time

production system uses all the meteorological observations from the ECMWF numerical weather prediction system, which is extracted from satellite real-time meteorological data. We also compared the temperature in the model with in-situ measurements. The results are shown in the response according to your point "P12, l124" in the minor comment. Generally good agreement can be seen. Therefore the model simulations could predict the secondary formation of HCHO well, but it can't be confirmed.

Based on the two further analysis, we noticed that the diurnal variation of HCHO is a mixed effect of primary emission, secondary formation, and probably also meteorology. It is impossible to gain the conclusion that which is the factor which causing the deviation of R(DOAS) and R(model). Therefore the R comparisons only generally evaluate the quality of model simulations on diurnal variations of HCHO. As you asked, both underestimation of primary emission and overestimation of secondary emission by model simulations can cause the similar fact that R(DOAS)>R(model). We can not firmly conclude which is the reason. And the method can't give quantified conclusion of HCHO source. Therefore we add a clarification in the revised manuscript. Changes in manuscript: It needs to be noted that the diurnal variation in HCHO is the result of the combined influence of primary emissions, secondary formation, and meteorology. We found that RMAX-DOAS was generally larger than RModel. However, it was impossible to determine the factor causing the deviation in RMAX-DOAS and RModel. Therefore, the R comparisons generally only evaluate the quality of the model simulations on diurnal variations in HCHO.

Minor P2, l3: {Q} Is the correlation coefficient (R=0.83)? If so, say (R=0.83, not ∼0.83) Response: Thank you for your suggestions. Correlation between HCHO VCDs retrieved from the MAX-DOAS measurements and those obtained from the CAMS model at 8:00 LT and 14:00 LT from October to December 2014 in different grids were compared. The correlation coefficient R is more than 0.83, So we use ∼0.83. And it is changed in the paper. Changes in manuscript: The HCHO VCDs of the CAMS model and MAX-DOAS were generally consistent with a correlation coefficient R2 greater

than 0.69..

P2, l14: {Q} How is "APEC blue" defined? Perhaps a brief statement of how the actual reduction strategies were defined and the defined APEC levels would be useful? Is there an APEC-red for example? Response: Thank you for your suggestions. For the sake of guaranteeing the smooth convening of the APEC meeting, China took a series of effective measures which played a prominent role in improving the air condition in Beijing and surrounding regions. As a result, a better quality environment emerged, which we called "APEC-Blue". This is reported in the Chinese website (https://baike.so.com/doc/7519682-7792600.html). The actual reduction strategies were added in the paper to make it clear. Changes in manuscript: Since November 1, 2014, parts of the Jing-Jin-Ji region and surrounding areas had begun to implement an emission reduction plan according to the APEC conference air quality assurance policy. Formal emission reduction measures were implemented in the Jing-Jin-Ji region and surrounding areas from November 3 and included limiting the production of factories, shutting down construction sites, implementing traffic restrictions based on even- and odd- numbered license plates, and improving road cleaning (Wang et al., 2016). In response to the possible adverse weather conditions from November 8– 10, the "enhanced emission reduction measures" were implemented in the Jing-Jin-Ji region and surrounding areas from November 6. These various efforts coupled with relatively favorable weather conditions than previous years resulted in the emission reduction measures having significant effects. Based on estimations, all types of main pollutants were reduced by over 40% in Beijing and by over 30% in other provinces, through these measures (Wang et al., 2016).

P2, l16-17: {Q} Do you or the authors of the Wang et al. make any conclusion as to why the O3 rose to 189%? Does this have to do with being in a NOx-limited or VOC-limited regime? Response: Thank you for your question. Wang et al.,2016a gave the reason that the O3 in urban and suburban areas of Beijing is mostly in the control area of VOCs. The possible reason for the increase of O3 is that the emission

control measures of NOx are greater than the emission control measures of VOCs, which leads to the weakening of the inhibition of O3 formation by NOx, resulting in significant increasing of O3 concentration. And it is added in the paper to make it clear.(Besides, the introduction is reorganization to make it more logical. ) Changes in manuscript: Wang et al (2016a) selected five representative in situ stations in different locations in Beijing and found that average concentrations of SO2, NO2, PM10, and PM2.5 decreased by 61.5%, 40.8%, 36.4%, and 47.1%, respectively, whereas the average concentration of O3 increased by 101.8%, compared with the same period over the last five years (PM2.5 since 2013). O3 in urban and suburban areas of Beijing is mostly in the control area of volatile organic carbons (VOCs). The possible reason for the increase in O3 is that the emission control measures of NOx are greater than the emission control measures of VOCs, which leads to the weakening of the inhibition of O3 formation by NOx, resulting in significant increases in O3 concentration.

P2, l25-P3, l1: {Q} What were Zhang's conclusions (briefly)? Response: Thank you for your advices. We have considered your advice, and we add the Zhang's conclusions to make it clear. During the APEC conference period, the average concentration of PM2.5 was 37.7 $\pm$ 35.4 mg/m3, which was 48% and 54% lower than that of BAPEC and AAPEC period, respectively. Compared with ultrafine particles (<100 nm), the number concentration of accumulation mode and coarse mode particles experienced more significant decreases by 47% and 68%, indicating that particles with larger sizes were better controlled during the APEC period. Changes in manuscript: Zhang et al (2017) analyzed the characteristics of aerosol size distribution and the vertical backscattering coefficient profile during the 2014 APEC summit using lidar observation. Particles with larger sizes were better controlled during the APEC period, with the number concentration of accumulation mode and coarse mode particles experiencing more significant decreases of 47% and 68% (Zhang et al., 2017).

P3, l1-2: ADD {REFS} for the published studies here. Response: Thank you for your advice. The REFS were added in the paper. Changes in manuscript: Published studies

have focused mainly on the effects of commonly measured gas pollutants, particulate matter, and aerosols, but not HCHO (Cheng et al., 2016; Fan et al., 2016; Huang et al., 2015; Li et al., 2016; Liu et al., 2016; Meng et al., 2015; Tang et al., 2015; Chen et al., 2015; Wang et al., 2016a; Wang et al., 2016b; Wang et al., 2017a; Wei et al., 2016).

P3, l5: {REF} is not in your final reference list. Response: Thank you for your remind. We are so sorry for making this mistake. The REF is added in the final reference list.. Changes in manuscript: Fried A, Cantrell C, Olson J, Crawford J H. Detailed comparisons of airborne formaldehyde measurements with box models during the 2006 INTEX-B and MILAGRO campaigns: potential evidence for significant impacts of unmeasured and multi-generation volatile organic carbon compounds, Atmos. Chem. Phys., 11, 9887-9957, 2011.

P3, l14-15: HO2 and OH are radicals not ions, please correct this. Response: Thank you for your remind. We are so sorry for making this mistake. It is corrected. Changes in manuscript: As an active gas, HCHO can be photolyzed to generate HO2 free radicals. HO2 rapidly and radically reacts with NO to generate OH, which can influence the oxidation ability of the atmosphere.

P3, l22: fix {REF}, you mean Honninger et al., 2004 right? Response: Thank you for your remind. We are so sorry for making this mistake. Yes, it is corrected in the paper. Changes in manuscript: A type of passive differential optical absorption spectroscopy system, called Multi-axis Differential Optical Absorption Spectroscopy (MAX-DOAS), has been used over the past decade to measure tropospheric trace gases (Honninger et al., 2004; Wagner et al., 2004; Sinreich et al., 2005; Wagner et al., 2007; Vigouroux wt al., 2009).

P4, 16: I would call it the Beer-Lambert Law Response: Thank you for your advice. It is corrected in the paper. Changes in manuscript: MAX-DOAS, which is an optical remote-sensing technology that records the spectra of scattered sunlight at different elevation angles, can be used to quantitatively measure trace gases based on Beer-

Lambert Law (Hönninger and Platt, 2002; Bobrowski et al., 2013; Roozendael et al., 2003; Trebs et al., 2004; Hönninger et al., 2004; Wagner et al., 2004).

P4, l9: {Q} Were clouds a factor? How often was it cloudy? Was the data pre-screened in any way? Response: Thank you for your question. In this paper, the effects of different cloud coefficients on MAX-DOAS inversion VCDs and the HCHO VCDs from MAX-DOAS and CAMS model under different cloud coefficients are both compared. It is found that the cloud coefficient has negligible influence on it. During the entire APEC period, it is basically sunny and cloudless weather. The data is pre-screened. The spectrum with too small a light intensity and an excessive integration time are removed.

Figure 1: Change the colour red on your figure, it is hard to read. Response: Thank you for your advice. It is corrected in the paper. Changes in manuscript:

P5, l21: : : :some point sources (e.g. XX and YY). Add some key examples, factories or power plants? Response: Thank you for your advice. Some point sources here mean stationary sources from the rural settlement. They are not factories and power plants. And it is added in the paper. Changes in manuscript: The site is mainly influenced by emissions from vehicles on China National Highway 111 that runs from the north and south as well as some stationary sources from the rural settlements across the highway (Zhang et al., 2017).

Figure 2: fix the text on your figure (e.g. spectrograph as one word) Response: Thank you for your suggestion. It is changed in the paper. Changes in manuscript:

P5, l24: {C} change stepping motor to stepper motor? Response: Thank you for your advice. It is changed in the paper. Changes in manuscript: This system comprises a telescope, stepper motor, spectrometer, and computer.

P6, l2: {Q} Why was the temp set to 20C? Response: Thank you for your question. The changing ambient temperature in China is from -15 °C to 35 °C in a year. And the

weather in spring and autumn is a little longer in Beijing with the temperature around 20 °C. So we set the temp as 20 °C to make sure a stable temperature in all seasons. Changes in manuscript: The spectrometer was placed in a temperature-controlled box at 20°C to ensure that the spectrograph could work at a stable temperature under the changing ambient temperature from -15°C to 30°C in China.

P6, l10: replace scanning times with SCANS Response: Thank you for your advice. It is changed in the paper. Changes in manuscript: Each measurement had an average of 100 SCANS, and the integration time was adjusted automatically based on the light intensity.

Figure 3: replace a1, a2, etc. with a3 a30 a90 etc. {you don't need to number each one, simply add elevation angles to the alpha directly} Response: Thank you for your advice. It is changed in the figure 3. Changes in manuscript:

Table 1: fix text .. Longitude – one word, {Q} What is the MAYA? Is that Ocean Optics? If so, add that. Response: Thank you for your advice. It is changed in the table. And Maya is a Ocean Optics spectrometer (https://oceanoptics.com/product/maya2000-pro-custom/). The briefly introduction of Maya is added in the paper. Changes in manuscript:

The spectrometer was produced by Ocean Optics and was named Maya (https://oceanoptics.com/product/maya2000-pro-custom/). The spectrometer covers the range of 290 nm to 420 nm, and its instrumental function is approximated as a Gaussian function with a full width at half maximum (FWHM) of 0.5 nm.

P6, 19: replace Doasis with DOASIS Response: Thank you for your advice. It is changed in the paper. Changes in manuscript: The ring structure (Fish and Jones, 2013), which is used to account for rotational Raman scattering effects, was calculated using DOASIS software (Kraus, 2006) based on the FRS and was included in the fit.

P6, l22: replace [derived] with [measured] {as you did in your Figure4} Response:

Thank you for your advice. I changed the description of figure 4 here. Now figure 4 is changed to figure 5 because some new figure is added. Changes in manuscript: Figure 5: Example of a DOAS fit of a spectrum to retrieve the slant column densities of HCHO; the red and blue curves indicate the fitted absorption structures and the derived absorption structures from the measured spectra, respectively.

Figure 6 shows the period of 3-8 November. Why didn't you use the period of 3-12 November (the whole APEC period)? Response: Thank you for your question. This data is used to support the analysis of the transport event. So the meteorological data for the time period corresponding to the transport event is displayed, which is the period of 3 to 8 November, 2014.

P8, l11: You describe 2 peaks on Nov 4 and Nov 7, but what about Nov 3, as seen on Figure 6 that actually has the HIGHEST HCHO VCDs? Response: Thank you for your advice. Two daily averaged HCHO VCD peaks were on Nov 4 and Nov 7, and the rise process of Nov 4 is from the evening of Nov 3.

Figure 7: perhaps replace UTC time with LT for consistency. Response: Thank you for your suggestion. I changed in the paper. Changes in manuscript:

Figure 9: Error bars equal retrieval error. {Q} How is this calculated? Response: Thank you for your question. We have considered your question, and we think it is more reasonable to use the standard deviation to represent the error bars. It is changed in the paper. And the figure 9 is changed to figure 11 due to some new figures were added. And about the retrieval error, we discuss at the section 2.3. Changes in manuscript:

Figure 11: Daily averaged values of HCHO VCDs from October 26, 2014 to November 20, 2014. Error bars denote standard deviations.

P10, l19-20: Is there any way to determine which is more important, the control measures of the meteorology? Perhaps a longer term study? Please comment. Response: Thank you for your suggestion. Comparisons of transports events between from the

polluted south area and clean north area indicate meteorology condition can vary HCHO amounts by about 50%. However meteorology condition is not under control. Reduction of HCHO emission in the south polluted area can be estimated by ~20% due to control measures of emissions. The significant effects of control measures are important for improving air qualities, especially under a meteorology condition which obstructs depositions of pollutants.

P11, l7-8: Could this have to do with a change in NOx-limiting vs. VOC-limiting cases? Please advise. Response: Thank you for your advice. We carefully think about your advice and do some research on previous literature. Wang et al., 2009 found that ozone formation is mainly controlled by VOCs in the near-suburbs of Beijing City and its high-value ozone areas in the downwind direction. In suburban counties and rural areas, the sensitivity of ozone generation to NOx becomes important. And the UCAS is located in the outer suburbs of Beijing, in other word, the UCAS belongs to the NOx-limiting area. During APEC, the NOx concentration gradually decreases due to the control measurement. As a results, the HCHO decreases. After APEC, control measurements are abolished, HCHO concentration is increased with the increasing of NOx. There should be not a change in NOx-limiting vs. VOC-limiting cases. So we can't draw the exact conclusion. On the other hands, according to the recommendation of reviewer 1, we seriously discussed it. The SNR in evening is low that makes the data not very credible. So I decided to remove this part from the text.

P11, l14: Where was the surface O3 measurement location exactly? What type of NO2 VCD was it, geo-approximated, same instrument and location? Please describe. Response: Thank you for your advice. The MAX-DOAS instrument was deployed on the balcony (without a roof) of a classroom on the 4th floor in the laboratory building in the campus of UCAS (116.67°E, 40.4°N). And the UCAS supersite is on the top floor of the laboratory building, which is about ten meters away from MAX-DOAS. The O3 was measured by UV photometry in the UCAS supersite. The NO2 VCD was obtained from the MAX-DOAS observation by using geo-approximated. The description was added

in the paper. Changes in manuscript: The MAX-DOAS instrument was deployed on the balcony (without a roof) of a classroom on the 4th floor in the laboratory building in the campus of UCAS (116.67°E, 40.4°N). The UCAS supersite is on the top floor of the laboratory building, which is about 10 m away from the MAX-DOAS instrument. Nitrogen oxide (NO, NO2, and NOx) was measured by chemiluminescence (model 42i; Thermo Scientific), and ozone (O3) was measured by UV photometry (model 49i; Thermo Scientific). These gas analyzers had precision values of 0.5 ppb and 0.4 ppb, respectively. Sec. 3.3 : Surface O3 data were obtained from in situ measurements in the UCAS supersite, and troposphere NO2 VCD data were retrieved from the same MAX-DOAS measurements using geometric approximation.

P11, 118: Too many significant figures! Response: Thank you for your advice. It is changed in the paper. Changes in manuscript: Direct analysis of the data indicates that noontime average HCHO had a higher correlation coefficient with NO2 VCD and O3 than rush hour. This implies that a small amount of HCHO comes from the traffic emissions during rush hour. A good correlation coefficient R2 of 0.73 was found between HCHO VCD and O3 during the noontime, which indicates that the main source of HCHO was from secondary photo-oxidation formation at noon. In contrast, a correlation coefficient of 0.38 between HCHO VCD and NO2 VCD during noontime was better than during rush hour (R2=0.06), which may be due to the contribution of vehicle emissions to HCHO precursors. A longer NO2 lifetime with less dispersion efficiency in winter and HCHO from continuously generated photo-oxidation contributed to the higher correlation between HCHO VCD and NO2 VCD at noon higher than during rush hour. The transport of NO2 and VOC may constitute one of the causes. The VOCs from transport generate HCHO due to strong photo-oxidation at noon.

P12, l124: What is the assumed temp in the model for Dec 1, 2014 then? Response: Thank you for your asking. We download the temp data of model at 2 meter and compare with the temp from in-situ instrument. The results show that the temp in the model also plummeted in December 1, 2014, and fell below 0 °C.

Figure: Hourly averaged temperature in CAMS model (grid of 0.125°× 0.125°and 0.25°× 0.25°,)at 2 metre and in-situ observations at 8:00 (a) and 14:00 LT (c) from October 29 to December 31, 2014.

P13, l16-20: Briefly state what associated errors clouds could pose. In l19 you say a slight variety (variation), give an error estimate please. Response: Thank you for your suggestion. It is added in the paper. Changes in manuscript: First, clouds can affect atmospheric radiative transport and thus influence optical paths. Furthermore, the atmospheric absorber densities [by (photo-)chemistry or convective transport] are potentially altered due to the changes in optical paths (Grats, ea et al. 2016). Second, AMFs calculated by geometrical approximation could be significantly biased from the reality under cloudy conditions (Brinksma, et al., 2008). REF: Gratsea, M., Vrekoussis, M., Richter, A., Wittrock, F., Schonhardt, Anja., Burrows, J., Kazadzis, S., Mihalopoulos, N., Gerasopoulos, E.: Slant column MAX-DOAS measurements of nitrogen dioxide, formaldehyde, glyoxal and oxygen dimer in the urban environment of Athens, Atmos. Environ., 135,118-131,2016.

P14, l16: Where does this number come from and what dates? It is not the same as Figure 12 and it is not mentioned anywhere else in your paper. Is it a typo? Please advise. Response: Thank you for your remind. There is mistake. This number (0.87) is the correlation coefficient R2, and the 0.934 in figure 12 is the correlation coefficient R. We redraw the figure 12 by using all the data from October to December, 2014. And the figure 12 is changed to figure 14 due to some new figures were added. The new correlation coefficient R2 of average HCHO VCDs with O3 is 0.73. It is changed in the paper. Changes in manuscript: A good correlation coefficient R2 of 0.73 was found between HCHO VCD and O3 during the noontime, which indicates that the main source of HCHO was from secondary photo-oxidation formation at noon.

P14,l20: Why the range? Grid sizes, I assume? Response: Thank you for your question. The range is mainly due to two different time periods(8:00 and 14:00 LT) and three different grid points. In order to make it clear, I calculate the averaged value

and change it in the paper. Changes in manuscript: The CAMS model underestimated HCHO VCD by about $1.63 \times 10^{15}$ molec cm-2 on average compared to the MAX-DOAS measurements.

Thank you for taking care of our manuscript.

Kind regards, Xin Tian E-mail: xtian@aiofm.ac.cn

Corresponding author : Pinhua Xie, Jin Xu, Yang Wang E-mail address: phxie@aiofm.ac.cn; jxu@aiofm.ac.cn; y.wang@mpic.de

Please also note the supplement to this comment:
https://www.atmos-chem-phys-discuss.net/acp-2018-440/acp-2018-440-AC2-supplement.pdf

[Figure]

The NO2 VCD is not described. Is it VCDgeo? Is it data from the same instrument and time? Did you also compare your O3 with the 7-9 & 16-18 periods? You don't have to show the plot but I would like to know the R of that? Hopefully it is very low to prove your point. Similarly, did you compare the NO2 VCDs with the HCHO VCDs from 11-14 period. You need a more complete assessment here to really prove your point.

**Changes in manuscript:**
**figure:**

[Figure]

[Figure]

**Figure 14: Scatter plots and linear regressions (a) of noontime average HCHO VCD measured by MAX-DOAS against O₃ VMRs measured by a stationary ozone monitoring instrument, and (b) rush hour average HCHO VCD against NO₂ VCD measured by MAX-DOAS from October to December 2014.**

**Fig. 1.**

(6) RMAX-DOAS vs Rmodel

figure:

[Figure]

[Figure]

Scatter plots and linear regression of $R_{Model}$ against $R_{MAX-DOAS}$ (see the text) from October to December 2014 (a) and from October to November and for December due to the changing of temperature (b). There are not significant differences between the two periods.

**Fig. 2.**

Figure 1: Change the colour red on your figure, it is hard to read.

**Changes in manuscript:**

**figure**

[Figure]

[Figure]

**Fig. 3.**

Figure 2: fix the text on your figure (e.g. spectrograph as one word)

**Changes in manuscript:**

**figure:**

[Figure]

**Fig. 4.**

Figure 3: replace a1, a2, etc. with a3 a30 a90 etc. {you don't need to number each
one, simply add elevation angles to the alpha directly}

**Changes in manuscript:**
**figure:**

[Figure]

**Fig. 5.**

Table 1: fix text .. Longitude – one word, {Q} What is the MAYA? Is that Ocean Optics?
If so, add that.

**Changes in manuscript:**
**Table**

| Spectrometer | | Azimuth | Elevation | Temperature | Location | | Measuring time |
|---|---|---|---|---|---|---|---|
| Name | Maya (Ocean Optics) | 0 ° | 3 °, 5 °, 10 °, 15 °, | 20 ℃ | Site | Yanxi Lake campus of UCAS | 6:30-18:30 |
| Spectral range | 290–420 nm | | 30 °, 90 °, | | Longitude | 116.67 °E | |
| FWHM | 0.5 nm | | | | Latitude | 40.4 °N | |

**Fig. 6.**

Figure 7: perhaps replace UTC time with LT for consistency.

**Changes in manuscript:**

**figure:**

[Figure]

**Fig. 7.**

[Figure]

Figure 9: Error bars equal retrieval error. {Q} How is this calculated?

**Changes in manuscript:**

**figure:**

[Figure]

**Figure 11: Daily averaged values of HCHO VCDs from October 26, 2014 to November 20, 2014. Error bars denote standard deviations.**

**Fig. 8.**

P12, l124: What is the assumed temp in the model for Dec 1, 2014 then?

**Response**:

[Figure]

**Figure: Hourly averaged temperature in CAMS model (grid of 0.125 °×0.125 °and 0.25 °×0.25 °)at 2 metre and in-situ observations at 8:00 (a) and 14:00 LT (c) from October 29 to December 31, 2014.**

**Fig. 9.**

---

## Author Response (AR2)

Response to Report#1

Dear reviewer,

Thank you very much for you giving minor corrections on our manuscript. We have carefully considered every comment, and made corresponding revisions in the revised manuscript and marked every change in red.

Point to point response is following:

1. Some suggestions:

P1, L17 : using A Multi-AXis Differential ….

P2, L5: cut out the word "suitably"

P2, L6: This finding MAY indicate…

P2, L19: …road cleaning [procedures] {optional word to add}

P2, L22: …weather conditions [compared with] previous years ….

P2, L25: …add a REFERENCE to the end of this line please.

P3, L4: … Wang et al.

P3,   L5: {would reduce significant figures here, 62%, 41%, etc.}

L14: VCDs

L15: {replace "surrounds" with "its surroundings based on THE Ozone…" or with"its suburbs based.."

P4, L17-20: {check with the journal as to how radicals are represented correctly and make sure you are using the correct notation}

P5, L5: …called Multi-AXis Differential {see P1, L17}

L12: ..in THE Huairou District..

L19: {I think you have a typo here, replace HONO with HCHO, right?}

P6, L5: …in the Yanqi Lake [region] near …

L6: from October 1, 2014 TO January ….

L8: …the mountains TO the southwest…

L10: ..and Tianjin ARE the primary pollution sources {or …ARE primary pollution hotspots.}

L18-19: …(Thermo Scientific, Model 42i), and ozone …photometry (Thermo Scientific, Model 49i).

P7, L4: …six elevation angles (EAs) 3, 5, 10 etc {add the degree symbol to each angle please}

FIGURE 3 – {not sure where your scattered light is, please fix the figure}

L5 : {this is a typo from me, change SCANS to scans}

L11: … of -10.6C to 20.7C.. {keep degree signs though}

L14: cut out the words {with a wind speed of less than 3.5 m s-1} and replace with … during the observationS, while wind speeds of ….

L19: …based on THE Beer-Lambert Law…

L25: ..for all the retrievals to determine {retrieve} slant …

P8, L2: …The Ring structure …

L12-L13: add brackets (and measured in a small sun zenith angle 90 elevation around noon).

P9, L16: .. VCDs ..

L17: … are obtained from the ..

P11, L1: {is it T5 or T255?}

L2-L3: change to 00:00, 06:00, 12:00, 18:00 UTC.

L15: …show that THE cloud coefficient …

P12, L1: {I would reduce to 1 decimal, e.g. 2.2%, -23.5% etc.}

L4: …and a dominant southerly wind flow with a speed of more than 2.0 m/s, during the two days …

L24: {keep significant figures constant, 17%, 16%, 12% …}

P14, L10: … the effects of the control measures, …

L14: … considerably lower than 10.3 or 9.20 …

L16: significant figures please

P15, l2: Equation 1

L9: ..could be the dominant ..

L9:… during other periods …

L13: .. replace obviously with statistically {or significantly} depending on what you mean

P16, L7: ..other VOCs could be the …

L12: (Figure 15a)?

L14: (Figure 15c)?

P17, L7: … the economy near the UCAS has grown rapidly …

L16: {statistically significant?}

P18, L2: .. and the CAMS model becomes larger or …and the CAMS model becomes more pronounced.

P19, L12: …on HCHO were evaluated…

L16: significant figures

**Response to above:** Thank you very much for your suggestion. We have considered your advice, it is changed in the manuscript.

2. Questions and advices:

P2, L1 {suggest 38% and 31%, as I don't think 4 significant figures is appropriate}

**Response:** Thank you very much for your suggestion. We considered the errors attributed to them and the correct number of significant figures. It was changed as ~40% and ~30%.

**Changes in manuscript:** During the period of the APEC conference, the average HCHO VCDs were ~40% and ~30% lower than that during the pre-APEC and post-APEC periods, respectively.

P2, L15: …emission reduction measurements {is it measurements or strategies or maybe measurement strategies? I would use emission reduction strategies in accordance…}

**Response:** Thank you very much for your advice. We used emission reduction strategies here.

**Changes in manuscript:** Some provinces, including Beijing, Tianjin, Hebei, Shanxi, Inner Mongolia, and Shandong implemented different emission reduction strategies in accordance with the air quality assurance plan (Wang et al., 2016a).

P3, L4-L7: {So the O3 increased by 101%, you could also say that the O3 level approximately doubled over this period then, right?}

**Response:** Thank you for your question. The O3 increased by 101%, which means the O3 level approximately doubled over this period then the same period over the last five years. It was changed in the manuscript.

**Changes in manuscript:** Wang et al, (2016a) selected five representative *in situ* stations in different locations in Beijing and found that average concentrations of $SO_2$, $NO_2$, $PM_{10}$, and $PM_{2.5}$ decreased by 62%, 41%, etc, respectively, whereas the average $O_3$ level approximately doubled over this period than the same period over the last five years ($PM_{2.5}$ since 2013)

L9 : {do you mean that there is less O3 titration due to NOx, if so re-word this? Not sure if inhibition of O3 is the best wording here?}

**Response:** Thank you for your advice. We re-worded this to make it clear.

**Changes in manuscript:** $O_3$ production rate depends on the ratios of volatile organic carbon (VOC) and NOx. The urban and suburban areas of Beijing are controlled by the NOx saturated condition of $O_3$ production. Since emission control measures are mainly focus on NOx, but not VOCs, decrease of NOx can cause significant increases of $O_3$ concentration (Wang et al., 2016a).

L10-12: {sorry but I am not sure what the difference is between traffic, urban, suburban and regional stations, define it or at least reference it please}

**Response:** Thank you for your question. At the beginning we introduced five representative in suit station, which were City Background Station, Regional Station, Suburban Station, City Station, and Transport Station. We defined it to make it clear.

**Changes in manuscript:** Wang et al, (2016a) selected five representative *in situ* stations in different locations in Beijing, which were Miyun Reservoir Station (City Background Station), Yuzhan Station (Regional Station), Changping Station (Suburban Station), Olympic Sports Center Station (City Station) and Xizhimen North Street Station (Transport Station), and found that average concentrations of $SO_2$, $NO_2$, $PM_{10}$, and $PM_{2.5}$ decreased by 62%, 41%, etc, respectively, whereas the average $O_3$ level approximately doubled over this period than the same period over the last five years ($PM_{2.5}$ since 2013).

Although the traffic and urban stations produce a lot of pollution due to motor vehicle emissions, the $NO_2$ concentrations of suburban and regional stations significantly dropped (47%) compared with the traffic and urban stations (23%) as a result of the control measures.

L22-23: .. experiencing significant descreases of 47% and 68% versus what? {…versus the average 2008 values, state what you mean here}

**Response:** Thank you for your question. It is comparison between before, during and after APEC. It was added in the manuscript to make it clear.

**Changes in manuscript:** Particles with larger sizes were better controlled during the APEC period, with the number concentration of accumulation mode and coarse mode particles experiencing more significant decreases of 47% and 68% than before and after the APEC period (Zhang et al., 2017).

P8,  L5-L6: {define what large RMS values are, what was your threshold please?}

**Response:** Thank you very much for your advice. The threshold $>10^{-2}$ was added to make it clear..

**Changes in manuscript:** Data with a large root mean square (RMS) of the residuals ($>10^{-2}$) and large relative intensity offset were also excluded.

P9, L7: {do your lower EAs have obstacles in their paths, if so, what is the use of them? what about using the geometric approximation for the lower EAs, that doesn't work either right? You have a lot of

unused data in your data set (ie. 3, 5, 10 degrees). Perhaps later to get a VCD from RTM?}

**Response:** Thanks for the suggestion! We checked the data and found the lower EAs are not obstructed. However the sensitivity height at a lower EA is too close to the surface. In order to derive the HCHO VCD in the boundary layer, the geometric light path at 15 ° is a good approximation in the boundary layer. In a following study, we will try to retrieve HCHO profiles from the full sequence for the comparisons with model simulations.

**Changes in manuscript:** This study used the geometric approximation method to determine HCHO VCDs at an elevation angle of 15 °. The geometric light paths at 15 ° and 30 ° are good approximations in the boundary layer. However lower systematic errors were achieved at 15 ° than at 30 ° by using the geometrical approximation (discussed in Section 2.3 below).

L15: {what is TG?}

**Response:** Thank you very much for your question. The TGs here means trace gases. It was changed in the manuscript.

**Changes in manuscript:** The systematic error of the HCHO VCDs calculated by the geometric approximation depends on the layer height of the trace gases and aerosols.

P9-10, l23-l1 : {from your figure 6, it appears that the VCDgeo > VCDamf for 30 degrees, but VCDamf>VCDgeo at 15 degrees, is there any explanation for this?}

**Response:** Thanks for the asking! The effect shown in Fig. 6 is from RTM simulations and consistent with the previous researches based on RTM simulations (Wang et al., 2017c, Ma et al., 2013, and Shaiganfar et al., Atmos. Chem. Phys., 11, 10871–10887, 2011).

This phenomenon is mainly due to the layer height of HCHO and aerosol, and relative azimuth angles.

L6: {any explanation for the large range in the fitting error?}

**Response:** Thanks for the question. The absolute fit error is almost constant value of ($1.2*10^{15}$ molecules cm$^{-2}$) at different HCHO dSCDs. The relative error is given here. The relative error also depends on the HCHO dSCDs. The large variation range of the relative error is due to the variation of HCHO dSCD. In order to clarify this point, we add the absolute fit error in the manuscript.

**Changes in manuscript:** and the hourly average of the HCHO VCD fitting error was from 4% to 27% for the entire period with the absolute fit error is ~$1.2 \times 10^{15}$.

L12: {if the total error is 12% on average, what is the range?}.

**Response:** Thank you very much for your suggestion. The range of the total error is 3%-21%. It was added in the manuscript

**Changes in manuscript:** Since the three errors are mainly independent, the total error can be calculated by combining all the above error sources, adding up to about 3% - 21% with 12% on average.

L16: {you defined what CAMS stands for in the intro, but I would also repeat it here since it has been 9 pages since you introduced it}

**Response:** Thank you very much for your suggestion. It was added in the manuscript

**Changes in manuscript:** Copernicus Atmosphere Monitoring Service (CAMS), which is managed by

ECMWF, publicly provides generally reliable atmospheric information.

L23 : TM5 chemical transport model (CTM)?

**Response:** Thank you very much for your advice. It was added in the manuscript to make it clear.

**Changes in manuscript:** In the simulation of HCHO, chemistry originating from the Transport Model 5 (TM5) had been fully integrated into the C-IFS, in which only gas phase reactions of HCHO are included.

L21: {it looks like 3 peaks to me?}

**Response:** Thank you for your question. The increasing process of HCHO VCDs is in the periods of 3-5 and 6-9 November. And the peaks appeared on November 4, 2014 and November 7, 2014, in the two periods, respectively.

P13, L5: … correspond to a minimum in the average HCHO VCDs (6.64 * 1015 molec cm-2).

**Response:** Thank you very much for your advice. We considered the correct number of significant figures. It was changed as $6.6 \times 10^{15}$ molec cm$^{-2}$.

**Changes in manuscript:** In contrast, the northeast and north directions correspond to a minimum in the average HCHO VCDs ($6.6 \times 10^{15}$ molec cm$^{-2}$).

L6-7: {what about VOCs from natural sources?}

**Response:** Thank you for your advice. The nature sources of VOC should be much lower than the anthropogenic sources, especially in the winter season. We clarify the point in the manuscript as" the nature sources of VOC in the north should be much lower than the anthropogenic sources in the winter season. "

**Changes in manuscript:** The northern cities are clean with low VOC emissions, and the nature sources of VOC in the north should be much lower than the anthropogenic sources in the winter season. Thus few precursors of HCHO were transported to the measurement station in the north wind.

P14, L1: {I would say 38% and 31% here, be consistent in your significant figures please.}

**Response:** Thank you very much for your suggestion. According to P2.L1, we considered the errors attributed to them and the correct number of significant figures. It was changed as ~40% and ~30%.

**Changes in manuscript:** The average HCHO VCDs were $10 \times 10^{15}$, $6 \times 10^{15}$, and $9 \times 10^{15}$ molec cm$^{-2}$ before, during, and after APEC, with fitting errors of 9.4%, 10.1%, and 9.7%, respectively. A noticeable decrease of ~40%, and ~30% during APEC was found compared with before and after APEC.

L10-L12: {how do you know this, what is your proof? }

**Response:** Thank you very much for your suggestion. We considered your advice, and the references were added.

As the measurement station is located in the northern suburban area of Beijing, the effects of the control measures, which were mainly implemented in the urban areas, on HCHO were only observed at the station when dominant southerly winds occurred (Fan et al., 2016, Li et al., 2015).

Li, W.T., G, Q.X., Liu, J.R., Li, L., Gao, W.K., Su, B.D.: Comparative Analysis on the Improvement of Air Quality in Beijing During APEC, 36(12), 4340-4347, 2015.

L19-L20: the O3 is surface but the NO2 is VCD, is this a potential source of error? If so, mention it somewhere.

**Response:** Thank you very much for your advice. We clarified the point as the following:

Here it needs to clarify that the O3 data is from the surface measurements, but NO2 and HCHO are the tropospheric VCD. Since NO2 and HCHO are mostly in the boundary layer, the effect of the discrepancy of measured layers on the correlation analysis is not significant.

**Changes in manuscript:** Here it needs to clarify that the $O_3$ data is from the surface measurements, but $NO_2$ and HCHO are the tropospheric VCD. Since $NO_2$ and HCHO are mostly in the boundary layer, the effect of the discrepancy of measured layers on the correlation analysis is not significant

P19. L23: {could state the per cent error here instead of exact value, since you did that earlier, but it is your choice}

**Response:** Thank you very much for your suggestion. We have considered your advice, and the per cent error is instead of exact value. It is changed in the manuscript.

**Changes in manuscript:** The CAMS model underestimated HCHO VCD by about 24% on average compared to the MAX-DOAS measurements.

Thank you for your valuable comments!

Kind regards,
Xin Tian
E-mail: xtian@aiofm.ac.cn

Corresponding author : Pinhua Xie, Jin Xu, Yang Wang
E-mail address: phxie@aiofm.ac.cn; jxu@aiofm.ac.cn; y.wang@mpic.de

Response to Co-editor

Dear editors,

  Thank you very much for your decision "Publish subject to minor revisions" on our manuscript. We have carefully considered your comment and comments from the reviewer. And we made corresponding revisions (marked in red) in the revised manuscript and wrote a point-to-point response to all the comments.

Changing in the manuscript is following:

P1-2, L27-L1: During the period of the APEC conference, the average HCHO VCDs were ~40% and ~30% lower than that during the pre-APEC and post-APEC periods, respectively.

P13, L1: HCHO VCDs considerably depend on wind directions, and the average HCHO VCDs were $8 \times 10^{15}$ molecules $cm^{-2}$ under the southerly wind (including southwest and southeast).

P13, L4-5: In contrast, the northeast and north directions correspond to a minimum in the average HCHO VCDs ($6.6 \times 10^{15}$ molec $cm^{-2}$).

P14, L1-3: The average HCHO VCDs were $10 \times 10^{15}$, $6 \times 10^{15}$, and $9 \times 10^{15}$ molec $cm^{-2}$ before, during, and after APEC, with fitting errors of 9.4%, 10.1%, and 9.7%, respectively. A noticeable decrease of ~40%, and ~30% during APEC was found compared with before and after APEC.

P14, L14-18: **Fig. 12d–f** indicate that the averaged HCHO VCDs under south winds during APEC were about $6.5 \times 10^{15}$ molec $cm^{-2}$, which was considerably lower than 10.3, or $9.2 \times 10^{15}$ molec $cm^{-2}$ in the pre-APEC and post-APEC periods. In addition the peak values due to transport from the south urban area on November 4, 2014 and November 7, 2014 during APEC shown in **Fig. 11** were 25% and 18% lower than the peak values under similar wind fields in the pre-APEC and post-APEC periods.

P16, L16-17: On average, the CAMS model underestimated HCHO VCDs by $1.6–2.0 \times 10^{15}$ molec $cm^{-2}$ and $1.3–2.1 \times 10^{15}$ molec $cm^{-2}$ compared to the MAX-DOAS measurements at 8:00 LT and 14:00 LT, respectively, due to different grid-sizes.

P17, L16-17: During the period of the APEC conference, the average HCHO was $6 \times 10^{15}$ molec $cm^{-2}$, which was ~40% and ~30% lower than that during the pre-APEC and post-APEC periods, respectively.

Thank you for taking care of our manuscript.

Kind regards,
Xin Tian
E-mail: xtian@aiofm.ac.cn

Corresponding author : Pinhua Xie, Jin Xu, Yang Wang
E-mail address: phxie@aiofm.ac.cn; jxu@aiofm.ac.cn; y.wang@mpic.de

---

## Author Response (AR3)

Dear editors,

Thank you very much for your advices. We have carefully considered your comment, and made corresponding revisions in the revised manuscript and marked every change in red.

The changing in the manuscript follows the calculation results below:

Pre-APEC:

averaged HCHO VCD, marked as a=10 $\times 10^{15}$ molecules cm$^{-2}$

During APEC

averaged HCHO VCD, marked as b=6 $\times 10^{15}$ molecules cm$^{-2}$

Post- APEC

averaged HCHO VCD, marked as c=9 $\times 10^{15}$ molecules cm$^{-2}$

the averaged total relative error of VCD = 17%, which is derived from the discussion in Sect. 2.3.

Since the reduction of Pre-APEC and APEC is calculated as following:

X=U/V=(b-a)/b

The absolute error of reduction is

$$\sigma X = \sqrt{\left[(\frac{\sigma_u}{u})^2 + (\frac{\sigma_v}{v})^2\right]} = \sqrt{(\frac{\sigma_{a-b}}{a-b})^2 + (\frac{\sigma_b}{b})^2} = \sqrt{\frac{\sigma^2_a + \sigma^2_b}{(a-b)^2} + (\frac{\sigma_b}{b})^2}$$

$$= \sqrt{\frac{(0.17 \times 10 \times 10^{15})^2 + (0.17 \times 6 \times 10^{15})^2}{(10 \times 10^{15} - 6 \times 10^{15})^2} + \frac{(0.17 \times 6 \times 10^{15})^2}{(6 \times 10^{15})^2}}$$

$$= \sqrt{\frac{0.17^2 \times 136}{4^2} + 0.17^2} \approx 0.52$$

So the difference of Pre-APEC and APEC is ~40% $\pm$ 52%.

As the same method, the reduction of Post-APEC and APEC is 0.63

So the difference of Pre-APEC and APEC is ~33% $\pm$ 63%.

**Changes in manuscript:**

P2L1: During the period of the APEC conference, the average HCHO VCDs were ~40%$\pm$52%, and ~33%$\pm$63% lower than that during the pre-APEC and post-APEC periods, respectively.

P14L3-4: A noticeable decrease of ~40%$\pm$52%, and ~33%$\pm$63% during APEC was found compared with before and after APEC.

P19L19-20: During the period of the APEC conference, the average HCHO was 6 $\times 10^{15}$ molec cm$^{-2}$, which was ~40%$\pm$52%, and ~33%$\pm$63% lower than that during the pre-APEC and post-APEC periods, respectively.

Thank you for taking care of our manuscript.

Kind regards,
Xin Tian
E-mail: xtian@aiofm.ac.cn

Corresponding author : Pinhua Xie, Jin Xu, Yang Wang
E-mail address: phxie@aiofm.ac.cn; jxu@aiofm.ac.cn; y.wang@mpic.de

---

## Author Response (AR4)

Dear editors,

Thank you very much for your advices. We have carefully redo statistics and resubmit in the revised manuscript and marked every change in red.

1. The changing in the manuscript follows the calculation results below:

The standard error of mean is calculated by the following formula:

$$s(\delta\bar{x}) = \sqrt{\frac{1}{n(n-1)}\sum_{i=1}^{n}(x_i - \bar{x})^2}$$

The averaged HCHO VCDs at three periods were changed to retain a decimal.
Thus
Pre-APEC:
averaged HCHO VCD, marked as a = $9.7 \times 10^{15}$ molecules cm$^{-2}$ with standard error of mean $0.8 \times 10^{15}$ molecules cm$^{-2}$
During APEC
averaged HCHO VCD, marked as b = $6.0 \times 10^{15}$ molecules cm$^{-2}$ with standard error of mean $0.5 \times 10^{15}$ molecules cm$^{-2}$
Post- APEC
averaged HCHO VCD, marked as c = $8.6 \times 10^{15}$ molecules cm$^{-2}$ with standard error of mean $0.9 \times 10^{15}$ molecules cm$^{-2}$

Since the reduction of Pre-APEC and APEC is calculated as following:
U =(a-b)/a
Averaged reduction U = (9.7-6.0)/9.7≈38%
The relative uncertainty of reduction is

$$\frac{\sigma u}{u} = \sqrt{(\frac{\sigma_{a-b}}{a-b})^2 + (\frac{\sigma_a}{a})^2} = \sqrt{\frac{\sigma^2_a + \sigma^2_b}{(a-b)^2} + (\frac{\sigma_a}{a})^2}$$

$$= \sqrt{\frac{0.8^2 + 0.5^2}{(9.7-6.0)^2} + \frac{0.8^2}{9.7^2}} \approx 27\%$$

Then the uncertainty of reduction is

$$\sigma_u = u \times \frac{\sigma_u}{u} = 38\% \times 27\% \approx 10\%$$

So the difference of Pre-APEC and APEC is $U+2\sigma_u = 38\% + 20\%$ calculated at 95% confidence limit.

As the same method, the averaged reduction of Post-APEC is 30%, and the Uncertainty($\sigma_u$) is 12%,

So the difference of Post-APEC and APEC is ~30% $\pm$ 24% calculated at 95% confidence limit..

**Changes in manuscript:**

P2L1:  During the period of the APEC conference, the average HCHO VCDs were ~38% $\pm$ 20%, and ~30% $\pm$ 24% lower than that during the pre-APEC and post-APEC periods calculated at 95% confidence limit, respectively.

P14L3-4: The average HCHO VCDs were $9.7 \times 10^{15}$, $6.0 \times 10^{15}$, and $8.6 \times 10^{15}$ molec cm$^{-2}$ before, during, and after APEC, with fitting errors of 9.4%, 10.1%, and 9.7%, respectively. A noticeable decrease of ~38% $\pm$ 20%, and ~30% $\pm$ 24% during APEC was found compared with before and after APEC, which was calculated at 95% confidence limit.

P19L19-20: During the period of the APEC conference, the average HCHO was $6.0 \times 10^{15}$ molec cm$^{-2}$, which was ~38% $\pm$ 20%, and ~30% $\pm$ 24% lower than that during the pre-APEC and post-APEC periods calculated at 95% confidence limit, respectively.

2. And according to Figure 16(a), I changed the correlation coefficient $R^2$ in P2L5 to 0.68.
**Changes in manuscript:**
The HCHO VCDs of the CAMS model and MAX-DOAS were generally consistent with a correlation coefficient $R^2$ greater than 0.68.

 3. The Author contribution was added before the acknowledgements in the manuscript.
**Changes in manuscript:**
**Author contributions**

XT, PX and JX contributed to designed the research. JX and AL designed the installation location of

the MAX-DOAS instrument and installed it in the UCAS. ZH and QZ downloaded and extracted

HCHO VCDs data from ECMWF. XT performed the data analyses and wrote the manuscript. FW and

CL provided suggestions for the manuscript. PX, JX and YW edited and developed the manuscript.

Thank you for taking care of our manuscript.

Kind regards,
Xin Tian
E-mail: xtian@aiofm.ac.cn

Corresponding author : Pinhua Xie, Jin Xu, Yang Wang
E-mail address: phxie@aiofm.ac.cn; jxu@aiofm.ac.cn; y.wang@mpic.de

---

## Author Response (AR5)

Dear editors and reviewers,

Thank you very much for your constructive comments and advices on our manuscript. Your positive evaluation and comment encourage us and would be great helpful to our research. We have carefully considered every comment, and made corresponding revisions in the revised manuscript and marked every change in red.

Point to point response is following:

**Response to RC1**

General Comments:

1、However, a problem that detracts from the entire paper is that the language is not fluent and precise. There are frequent spelling and grammatical errors (fragment sentences, unnecessary words, incorrect verb tense, missing articles, convoluted or run-on sentences). The authors are strongly suggested to engage the help of an English editor.

**Response:** Thank you very much for your suggestions. We have considered your advice, and asked for help from an English language service.

2、A few sections would benefit from re-structuring for increased logical flow and clarity. The scientific methods and assumptions are not sufficiently clearly outlined. The methodology section is disorganized and needs additional technical details. The paper does present some novel data and reaches substantial conclusions, but the results are sometimes not enough to support the interpretations and conclusions without further statistical analysis and/or expanded discussion. When discussing results, the authors must consider whether trends and differences in measured values they are interpreting are statistically significant given calculated or expected uncertainties. The authors must also try to place their conclusions within the context of previous literature (e.g., presented in the introduction).

**Response:** Thank you for your advice. The error budgets are added in sec. 2.3. The section 2 has been reorganized. The introduction has also been rewritten.

Specific Comments:

1、For all regressions, the coefficient of determination (R2) statistic may be more appropriate since this value indicates the variance in the dependent variable that is predictable from the independent variable.

**Response:** Thank you very much for your suggestion. We have considered your advice, and all coefficient of determination R are changed to $R^2$ in this paper, including in the figures.

2、Page 2 Line 3 What specific results had a good correlation coefficient? VCDs of HCHO? Other?

**Response:** Thank you for pointing out. It's the VCDs of HCHO. It is changed to make it clear.

**Changes in manuscript:** The HCHO VCDs of the CAMS model and MAX-DOAS were generally consistent with a correlation coefficient $R^2$ greater than 0.68.

3、Page 2 Line 23 Did the emissions decrease from 100% to these values or are these values the quantity of the decrease? It is not clear

**Response:** Thank you for your advice. These values indicate the proportion of emissions from the corresponding source. It is changed to make it clear.

**Changes in manuscript:** Balance (CMB) model showed that the contributions of coal-fired boilers, dust, and motor vehicles to $PM_{2.5}$ in Beijing were around 2%, 7%, and 30%, respectively, during the APEC summit (Cheng et al., 2016).

4、Page 2 In general, since some of the pollutants were measured by multiple papers that you cite, consider sorting this paragraph by pollutant rather than by author. Otherwise, it becomes repetitive and confusing to have to keep referring to the values from the previous papers earlier in the paragraph.

**Response:** Thank you for your advice. We have considered your advice, and it is changed in the paper.

**Changes in manuscript:** Presently, many studies have analyzed the effects of emission reduction measures during the APEC summit. Ground-based observations were taken to investigate the air quality changes associated with a series of stringent emission-reduction measures (Fan et al., 2016; Li et al., 2016; Liu et al., 2016; Tang et al., 2015; Chen et al., 2015; Wang et al., 2016a; Wang et al., 2016b; Wang et al., 2017a). Wang et al, (2016a) selected five representative *in situ* stations in different locations in Beijing, which were Miyun Reservoir Station (City Background Station), Yuzhan Station (Regional Station), Changping Station (Suburban Station), Olympic Sports Center Station (City Station) and Xizhimen North Street Station (Transport Station), and found that average concentrations of $SO_2$, $NO_2$, $PM_{10}$, and $PM_{2.5}$ decreased by 62%, 41%, etc, respectively, whereas the average $O_3$ level approximately doubled over this period than the same period over the last five years ($PM_{2.5}$ since 2013). $O_3$ production rate depends on the ratios of volatile organic carbon (VOC) and NOx. The urban and suburban areas of Beijing are controlled by the NOx saturated condition of $O_3$ production. Since emission control measures are mainly focus on NOx, but not VOCs, decrease of NOx can cause significant increases of $O_3$ concentration (Wang et al., 2016a). Although the traffic and urban stations produce a lot of pollution due to motor vehicle emissions, the $NO_2$ concentrations of suburban and regional stations significantly dropped (47%) compared with the traffic and urban stations (23%) as a result of the control measures. The $NO_2$ emitted by motor vehicles in the Beijing urban area remained high even under the measures taken to limit the number of vehicles (Wang et al., 2016a). Space observations were also used to evaluate the effect of emission control measures on the changes in $NO_2$ tropospheric vertical column densities (VCDs) and aerosol optical depth (AOD) in Beijing and its surroundings based on the Ozone Monitoring Instrument (OMI) and Moderate Resolution Imaging Spectroradiometer (MODIS) retrieval. The results showed that $NO_2$ VCD and AOD were mostly reduced by 47% and 34% in Beijing, respectively (Huang et al., 2015; Wei et al., 2016; Meng et al., 2015). The analytical results of the Chemical Mass Balance (CMB) model showed that the contributions of coal-fired boilers, dust, and motor vehicles to $PM_{2.5}$ in Beijing were around 2%, 7%, and 30%, respectively, during the APEC summit (Cheng et al., 2016). Zhang et al (2017) analyzed the characteristics of aerosol size distribution and the vertical backscattering coefficient profile during the 2014 APEC summit using lidar observation. Particles with larger sizes were better controlled during the APEC period, with the number concentration of accumulation mode and coarse mode particles experiencing more significant decreases of 47% and 68% than before and after the APEC period (Zhang et al., 2017). Published studies have focused mainly on the effects of commonly measured gas pollutants, particulate matter, and aerosols, but not HCHO (Cheng et al., 2016; Fan et al., 2016; Huang et al., 2015; Li et al., 2016; Liu et al., 2016; Meng et al., 2015; Tang et al., 2015; Chen et al., 2015; Wang et al., 2016a; Wang et al., 2016b; Wang et al., 2017a; Wei et al., 2016).

5、Page 3 Line 6 Three sources are listed despite elsewhere in the paper it is stated that there are two sources, which one is it?

**Response:** Thank you very much for your reminding. There are mainly two sources for troposphere formaldehyde, besides, only a small fraction of HCHO is from direct emissions of biogenic sources (e.g., vegetation).

**Changes in manuscript:** Troposphere formaldehyde mainly originates from two sources.

6、Page 3 This discussion may benefit from writing out some of the most important chemical equations for the reader equations for the reader.

**Response:** Thank you for your suggestion. Accordingly, equations are added in the paper.

**Changes in manuscript:** All of the photolysis equations of HCHO to form OH radical at wavelengths below 370 nm are listed as follows:

$$HCHO + hv \rightarrow H + HCO(\lambda \leq 370nm) \rightarrow H_2 + CO \tag{1}$$

$$H + O_2 \rightarrow HO_2 \tag{2}$$

$$HCO + O_2 \rightarrow HO_2 + CO \tag{3}$$

$$HO_2 + NO \rightarrow OH + NO_2 \tag{4}$$

7、Page 3 Line 16 It is unclear whether the importance of quantifying HCHO is to track emissions of VOCs or NVOCs or the generation of OH, or all of these.

**Response:** Thank you for your suggestion. It represents all of these. And it is added to make it clear.

**Changes in manuscript:** Therefore, HCHO can reflect anthropogenic VOC emissions and VOC emissions through the fast production of short-lived NMVOCs. Identifying the major sources of HCHO is essential for quantifying the photolysis sources of OH and their contributions to aerosol formation and for effectively controlling photochemical pollution (Bauwens, et al., 2016; Chang, et al., 2016; Ling, et al., 2017; Ma, et al., 2016; Tanaka, et al., 2016).

8、Page 4 Line 5 What unit of HCHO? VCDs?

**Response:** Thank you for pointing it out. It is VCDs. And it is added in the paper.

**Changes in manuscript:** In this study, we used the ground-based MAX-DOAS instrument installed in Huairou District (suburban area) of Beijing to evaluate the effects of the sources and depositions of HCHO VCDs and their relations with emission control measures and meteorological conditions during the period from October 26, 2014 to November 20, 2014.

9、Page 4 Line 10 The last paragraph of your introduction may benefit from explicitly stating your research objectives (perhaps as a list). What were all the components included in the spectral fitting for HCHO? Include what cross section reference spectra used (including author), fitting window etc.

**Response 1:** Thank you very much for your advice. It is changed in the paper.

**Changes in manuscript:** In this study, we used the ground-based MAX-DOAS instrument installed in the Huairou District (suburban area) of Beijing to evaluate the effects of the sources and depositions of

HCHO and their relations with emission control measures and meteorological conditions during the period from October 26, 2014 to November 20, 2014. Two pollution episodes and their relationships with meteorological conditions were analyzed during APEC to evaluate the effects of regional transport and local emissions. Afterwards, three episodes, defined as "pre-APEC," the period of APEC and "post-APEC," were used to evaluate the influences of emission control measures on the changes in HCHO VCD during APEC. The correlations between HCHO VCDs with $NO_2$ VCDs and $O_3$ were used to determine the main HCHO sources and evaluate the dominant error sources of HCHO simulations of Copernicus Atmosphere Monitoring Service (CAMS) model.

**Response 2:** Thank you very much for your suggestion. The spectral fitting for HCHO including cross section, reference spectra and fitting window is introduced in detail in section 2.1 (MAX-DOAS Methodology). And the information should belong to technique detail, so I think it is more appropriate to be written in section 2 than in the introduction.

**Changes in manuscript: Fig. 2** shows a structural representation of the MAX-DOAS system. This system comprises a telescope, stepper motor, spectrometer, and computer. Sunlight is focused by the telescope, which is installed outdoors and reaches the spectrometer through an optical fiber. The spectrometer was placed in a temperature-controlled box at 20 ℃ to ensure that the spectrograph could work at a stable temperature under the changing ambient temperature from -15 ℃ to 30 ℃ in China. The spectrometer was produced by Ocean Optics and was named Maya (https://oceanoptics.com/product/maya2000-pro-custom/). The spectrometer covers the range of 290 nm to 420 nm, and its instrumental function is approximated as a Gaussian function with a full width at half maximum (FWHM) of 0.5 nm. MAX-DOAS was routinely operated for 24 h. Due to the intensity of the sunlight, only the daytime measurements were used for analysis. The nighttime measurements could be used to correct the dark current and offset. The azimuth angle view of the telescope was fixed at 0 °(North) during the entire observation period. A full MAX-DOAS scan comprises six elevation angles (EAs) (3 °, 5 °, 10 °etc) and lasts for approximately 10 min (see **Fig. 3**). Each measurement had an average of 100 scans, and the integration time was adjusted automatically based on the light intensity. **Table 1** lists the detailed setup of the MAX-DOAS instrument.

10、Section 2.1 Please list the final equation used for calculating VCDs from fitted DSCDs given your geometric approximation.
**Response:** Thank you for your advice. It is added in the paper.
**Changes in manuscript:**

Then the tropospheric VCD can be obtained from the following equation:

$$VCD = \frac{\triangle SCD}{\dfrac{1}{\sin(\alpha)} - 1}$$

11、Reorganize section 2 so that sections 2.1, 2.3, and 2.4 are grouped together for a more logical sequence flow. What is included in the VCD error calculation?

**Response:** Thank you for your advice. It is changed in the paper. The original sections 2.2 "**Monitoring locations**" and 2.3 **"MAX-DOAS instrument and measurement"** are grouped together as the new section 2.1 **"Monitoring locations and instrument".** The original sections 2.1 "**MAX-DOAS Methodology**" and 2.4 "**DOAS analysis**" are grouped together as the new section 2.2 "**DOAS Spectral retrieval and determination of troposphericVCD".** The error budgets is added in the text, see Sec. 2.3 "**Error budgets"**.

12、Are any of the MAX-DOAS VCDs removed from the dataset due to cloud fraction? If so, what was the cut off cloud fraction value?

**Response:** Thank you for your question. In this paper, the effects of different cloud coefficients on MAX-DOAS inversion VCDs and the HCHO VCDs from MAX-DOAS and CAMS model under different cloud coefficients are both compared. It is found that the cloud coefficient has negligible influence on it. During the entire APEC period, it is basically sunny and cloudless weather. It is added in the paper to make it clear. The data is pre-screened. The spectrum with too small a light intensity and an excessive integration time are removed. The spectrum pre-screen is added in the section 2.3.

**Changes in manuscript:**

P11L16-20, 1) The effects of different cloud coefficients on MAX-DOAS inversion VCDs and the HCHO VCDs from MAX-DOAS and the CAMS model under different cloud coefficients were compared. The results show that the cloud coefficient had a negligible influence on the retrieval of HCHO VCDs by MAX-DOAS. Additionally, sunny and cloudless weather generally occurred during the entire APEC period. Thus, all of the data obtained in the different cloud coefficients were used. P8L6-8, 2) We excluded data for solar zenith angle (SZAs) larger than 75 °because of the stronger absorptions of stratospheric species and a low signal-to-noise ratio. Data with a large root mean square (RMS) of the residuals ($>10^{-2}$) and large relative intensity offset were also excluded.

13、Page 5 Line 8 It is unclear what causes lower systematic errors.

**Response:** Thank you for your question. This sentence is not very clear so that caused some misunderstandings for you. We make changes in the paper to make it clearer. 1) We mean that: According to Ma et al., 2013 and Wang et al., 2017c, they found the systematic error is larger for larger elevation angles and larger RAA. So this study uses the geometric approximation method to determine HCHO VCDs at an elevation angle of 15 °to avoid surface obstacles on light paths along the line of sight, at the same time, it has lower systematic errors at 15 °than 30 °. 2) And we add the discussion of the error budgets for geometric approximation in section 2.3. It also find the systematic errors at 15 °is smaller than 30 °by using the geometrical approximation.

**Changes in manuscript:** However lower systematic errors were achieved at 15 °than at 30 °by using the geometrical approximation (discussed in Section 2.3 below). (P9L9-10)

14、Page 6 Line 18 Why was the FRS from this day chosen to fit all retrievals? Since you are fitting all your measured spectra against one FRS, you must consider the effect of SCD(FRS) (the component of trace gas in the FRS used) and the SCD(Solar Zenith Angle), which is the difference in the stratospheric component of SCD observed due to the difference in SZA between the times of measurement and the FRS time. SCD(SZA) changes with time and change the apparent diurnal trends. Please justify why you did not account for the SCD(SZA) and SCD(FRS). For example, was your FRS was obtained during a very low pollution period and/or are the stratospheric HCHO levels are expected to be trivial. For more information see Wagner, T., Ibrahim, O., Shaiganfar, R. and Platt, U.: Mobile MAX-DOAS observations of tropospheric trace gases, ATMOSPHERIC Meas. Tech., 3(1), 129–140, 2010.

**Response:** Thank you very much for your advice. Maybe my expression is not clear in the text, which makes you misunderstood. First, we use a FRS to retrieve all the spectra and obtain the dSCD. Then I subtract the dSCD at 90 °from the dSCD at off-zenith angles in the same elevation sequence to derive the delta SCD. The procedure can deduct the influence of stratospheric absorption and variation of instrumental properties. We add the description in the paper to make it clear.

**Changes in manuscript:** The geometric approximation was used to convert the dSCD to the tropospheric VCD. In the first step, the differential slant column densities (dSCDs) were derived from the DOAS spectral analysis with a so-called FRS (and measured in a small sun zenith angle at 90 ° elevation around noon) (Hermans et al., 2003; Hönninger and Platt, 2002; Kraus, 2006). The SCD includes two parts of the absorption signal of the troposphere and stratosphere. To remove the interference of stratosphere absorption and variation in instrumental properties, dSCD at off-zenith elevation angles were subtracted by the dSCD at 90 °elevation angle in the same elevation sequence to derive $\Delta$SCD following the equation below:

$$\Delta\mathrm{SCD}\alpha{=}dSCD\alpha \neq 90° - dSCD\alpha = 90°. \qquad\qquad (5)$$

(P8L13-19)

15、Page 6 Line 23 What is the software reported SCD error in molec/cm2?
**Response:** Thank you for your reminding. The SCD error is $8.14 \times 10^{15}$ molec/cm$^2$.
**Changes in manuscript:** HCHO SCD was $7.21 \times 10^{16}$ molecules cm$^{-2}$ with an error of $8.14 \times 10^{15}$ molecules cm$^{-2}$. (P8L11)

16、Page 6 Line 24 Are you missing units on this number?
**Response:** Thank you for the asking. The root mean square of the residual error here is the residual spectral structures of optical depth, and there isn't unit for the optical depth.

**Changes in manuscript:** The root mean square of the optical depth of the residual spectral structures was $1.08 \times 10^{-3}$. (P8L11-12)

17、Page 7 section 2.6 At what height were these meteorological parameters measured? At what time

frequency before averaging?

**Response:** The height of these meteorological parameters measured is about 15m. The weather station is about ten meters away from MAX-DOAS. All measured meteorological parameters are recorded of 1 min time intervals.

**Changes in manuscript:** The MAX-DOAS instrument was deployed on the balcony (without a roof) of a classroom on the 4$^{th}$ floor in the laboratory building in the campus of UCAS (116.67 °E, 40.4 °N). The UCAS supersite is on the top floor of the laboratory building, which is about 10 m away from the MAX-DOAS instrument.    (P6L17-19)

Meteorological parameters, including wind speed (WS), wind direction (WD), temperature (T), and relative humidity (RH), were continuously measured by a MetPak automatic weather station (Gill Instruments Ltd, Lymington, UK) at the UCAS superstation from October 28, 2014 to December 31, 2014 (**Fig. 4a**). All of the measured meteorological parameters were recorded at 1-min time intervals. (P7L9-12)

18、Page 7 Line 24 What does a "static" weather situation mean?

**Response:** The static weather means the wind speed less than 3.5 m s$^{-1}$. Halfacre et al., 2014 defines that the wind speeds (mode of ~3.5ms-1) is under relatively calm conditions (Halfacre, J.W., Knepp, T.N., Stephens, C.R., Pratt, K.A., Shepson, P., Simpson, W.R.,        et        al.: Temporal and spatial characteristics of ozone depletion events, Atmos. Chem. Phys., 14: 4875–4894, doi:10.5194/acp-14-4875-2014, 2014.**).** It is defined in the paper to make it clear.

**Changes in manuscript:** Halfacre et al (2014) defines the relatively calm conditions with wind speeds of less than 3.5 m s$^{-1}$ as the static weather situation. The static weather situation frequently occurred during the observations, while wind speeds of more than 3.5 m s$^{-1}$ usually appeared under northwest and west winds. (P7L15-17)

19、Better organization and flow in section2 may be achieved by describing MAX-DOAS methodology in this order: general description of the MAX-DOAS instrument, description DSCDs fitting, determination of VCDs from DSCDs, measurement sequence, and then viewing azimuth and location.

**Response:** Thank you for your advice. It is changed in the paper. The original sections 2.2 **Monitoring locations** and 2.3 **MAX-DOAS instrument and measurement** are grouped together as the new section 2.1 **Monitoring locations and instrument.** The original sections 2.1 **MAX-DOAS Methodology** and 2.4 **DOAS analysis** are grouped together as the new section2.2 **DOAS Spectral retrieval and determination of the troposphericVCD.**

**Changes in manuscript:** See section 2.

20、Figure 6 . manuscript: oposphericVCDtion2.2 hen viewing azimuth and location.ion of Phys

**Response:** All the days measured are quality controlled. The quality control criteria are added in the paper.

**Changes in manuscript:** We excluded data for solar zenith angle (SZAs) larger than 75 ºbecause of the stronger absorptions of stratospheric species and a low signal-to-noise ratio. Data with a large root mean square (RMS) of the residuals (>10$^{-2}$) and large relative intensity offset were also excluded. (P8L6-8)

21、Page 8 Line 13 Please quantify in some way the relative change in solar radiation

and temperature compared to the days were peaks were not apparent.

**Response:** Thank you for your advice. It is added in the paper.

**Changes in manuscript:** The daily averaged intensity of solar radiation and temperature on November 4 and 7 were compared with data from the two periods of November 4 to 7 and October 1 to December 31, 2014 (**Fig. 8**). The differences in averaged solar radiation and temperature on November 4 and 7 compared to the period from October 1 to December 31, 2014, were 2.2%, -23.5% etc, respectively. (P12L1-4)

[Figure]

**Figure 8: Averaged intensity of solar radiation and temperature in the four periods of November 4, November 7, November 3 to 8, and October 1 to December 31, 2014. Error bars denote the standard deviations.**

22、Page 8 Line 21 Please define (or find a better descriptor for) "good dispersion conditions".

**Response:** Thank you for your advice. It is defined in the paper.

**Changes in manuscript:** The value on November 6, 2014 was probably caused by the good dispersion conditions under the northwest winds with speeds of more than 3.5 m s$^{-1}$, with the air mass mainly originating from the clean northwest area. (P12L13-14)

23、Page 9 Line 14 What type of relevant pollution sources do these cities have? Primary and/orsecondary? Are there many industry and/or vehicular sources?

**Response:** Thank you for your question. The type of relevant pollution source Tangshan, Baoding, Shijiazhuang and Tianjin is primary source. And there are a lot of industry in those cities. For Beijing, the main source is the vehicular source, especially in the urban area (Lin et al., 2009; Lin et al., 2012; Shao et al., 2006; Tang et al., 2015; Wei et al., 2016). The reference is added in the paper (sec. 2.1).

**Changes in manuscript:** Relevant pollution sources in Tangshan, Baoding, Shijiazhuang, and Tianjin are primary pollution hotspots. In Beijing, vehicles are the predominant pollution source, especially in the urban areas (Lin et al., 2009; Lin et al., 2012; Shao et al., 2006; Tang et al., 2015; Wei et al., 2016). (P6L12-14)

24、Page 9 Line 16 You may want to state explicitly that there are fewer, smaller or less polluted cities in the Northern region here. Are the lower VCDs due to just dispersion or is it also chemical aging, et

**Response:** Thank you very much for your advice. There are less polluted cities in the Northern region. The northern is clean with low VOC emissions, so there are few precursors of HCHO transported to the

measurement station. The lower HCHO VCDs under such conditions are mainly due to less VOC precursors of HCHO.

**Changes in manuscript:** The northern cities are clean with low VOC emissions, and thus few precursors of HCHO were transported to the measurement station in the north wind. The lower HCHO VCDs under such conditions are mainly due to fewer VOC precursors of HCHO. In summary, the wind from this area prominently contributes to the dispersion of the pollutants.

25、Page 9 Line 17 and 18 Please explain the dependence of the VCDs on wind speed under different wind directions.

**Response:** Thank you very much for your suggestion. It is added in the paper.

**Changes in manuscript:** In terms of the dependence of HCHO on wind speed, the HCHO VCDs decrease along with the increasing wind speed under the northerly fast and clean wind, which results in the rapid dissipation of the pollution. Under the southerly wind, the HCHO VCDs increase with increasing wind speed. Thus, transport from the south polluted air to the observation site occurs more easily under southerly winds with relatively high wind speeds.

26、Page 10 Line 6 What are the errors on each of the VCD values. Are they statistically different?

**Response:** Thank you very much for your advice. According to the advice of review 2, we think it is more reasonable to use the standard deviation to represent the error bars. It is changed in the paper. And the figure 9 is changed to figure 11 due to some new figures were added. And we add the standard deviation on each of VCD values. It is added in the paper.

**Changes in manuscript:** The average HCHO VCDs were $9.65 \times 10^{15}$, $5.99 \times 10^{15}$, and $8.65 \times 10^{15}$ molec cm$^{-2}$ before, during, and after APEC, with fitting errors of 9.39%, 10.12%, and 9.74%, respectively.

27、Page 10 Line 17. This sentence is too vaguely written. Also, depending on whether the differences between the peak values are statistically significant, depending on the expected errors and the significance of the wind direction change, you may not have sufficient evidence to support this conclusion. Also consider that during APEC time the conditions were not only northerly winds but also higher wind speeds, which you state earlier in the paper tends to reduce the VCDs (which should be explained for clarity). Same comment for the sentence on lines 19 and 20.

**Response:** Thank you very much for your advice. According to the question 43, we plot the VCD data against the wind speed and direction for the pre-APEC, during-APEC, and post-APEC periods, seperately. And we re-organized our discussion in the section 3.2. Please see the modified paragraph in the following **.**

**Changes in manuscript:** As the measurement station is located in the northern suburban area of Beijing, the effects of the control measurements, which were mainly implemented in the urban areas, on HCHO were only observed at the station when dominant southerly winds occurred. We thus plotted the dependence of HCHO VCDs on the wind speed and directions in **Fig. 12d–f** for the pre-APEC, APEC, and post-APEC periods. **Fig. 12d–f** indicate that the averaged HCHO VCDs under south winds during APEC were about $6.46 \times 10^{15}$ molec cm$^{-2}$, which was considerably lower than 10.29, 6.46, and $9.20 \times 10^{15}$ molec cm$^{-2}$ in the pre-APEC and post-APEC periods. In addition the peak values due to transport from the south urban area on November 4, 2014 and November 7, 2014 during APEC shown in **Fig. 11** were 25.75% and 18.3% lower than the peak values under similar wind fields in the pre-APEC and post-APEC periods. In general, the HCHO values under the dominant southerly wind field were considerably lower during APEC than the pre-APEC and post-APEC periods. The

phenomenon implies that the control measures had a certain effect on reducing the concentration of HCHO. This suggests that the implementation of control measures during the APEC summit reduced the concentrations of NO$_2$ and aerosols (Liu et al., 2016; Zhang et al., 2017).

[Figure]

**Figure 12: Wind roses in (a) the "pre-APEC", (b) the APEC, and (c) the "post-APEC" periods. Dependence of HCHO VCDs (10$^{15}$ molec cm$^{-2}$) on wind directions for different wind speeds in the pre-APEC (d), during the APEC (e), and post-APEC (f) periods.**

28、Page 10 Line 25 Some basic equations on HCHO chemistry in the introduction
section would be very helpful for the reader by the time they get to this point
in the paper.
**Response:** Thank you very much for your suggestion. The equations are listed in the introduction, as the response to question 6.
**Changes in manuscript:** Atmospheric photochemical reactions are related to the intensity of solar radiation as indicated in equation 1.

29、Page 11 Line 3 Are you suggesting that this peak in the diurnal variation is due primarily to secondary production of formaldehyde rather than direct emissions? Since the most light is available mid-afternoon and local direct emissions are relatively smaller compared to secondary production? Please make this clearer to the reader.
**Response:** Thank you for your question. Yes, we conclude this peak in the diurnal variation is due primarily to secondary production of formaldehyde rather than direct emissions. It is described in the paper.
**Changes in manuscript:** Since most light is available in the early afternoon and local direct emissions are relatively smaller compared to secondary production, the secondary production of formaldehyde primarily caused the peak at 14:00.

30、Page 11 Are your conclusions that the diurnal variability is driven by variation in light
levels rather than diurnal variations in emissions? If light measurements are available,
you could try correlating the light intensity with the VCDs.

**Response:** Thank you for your advice . Anderson, et al., 1996; Lee, et al., 2015; Pinardi, et al., 2013 report that the increased HCHO at early afternoon implies that photo-oxidation of VOCs was very rapid due to the peak solar irradiance at this point in the day. We also compare the correlating of the light intensity with the VCDs, but the result shows poor correlation. The reason should be that the lifetime of HCHO and the photochemical reaction rate of VOC to generate HCHO contribute to non linear dependence of HCHO VCDs on light intensity. In order to clarify this point, we modified the manuscript accordingly. Please see the modifications below.

[Figure]

**Changes in manuscript:** Since most light is available in the early afternoon and local direct emissions are relatively smaller compared to secondary production, the secondary production of formaldehyde primarily caused the peak at 14:00. The diurnal variation in VOC emissions could also play a role in the diurnal variation of HCHO. However, the typical life time of VOCs can reach several days. The diurnal variations in VOC emission are unlikely to change the abundance of atmospheric VOCs. Therefore, diurnal variation in photo reaction rate could be a dominant driving factor. Other smaller peaks appeared in the evening during another period of busy traffic (16:00–18:00 LT), which might be caused by primary pollution sources, e.g., exhaust fumes from vehicles. Thus, the diurnal variations in HCHO during all three episodes were similar to the typical patterns of secondary sources as reported in Anderson, et al (1996), Lee, et al (2015), Pinardi, et al (2013).

31、Page 11 Line 5 Where can the reader see evidence of similar diurnal trends in the secondary sources?

**Response:** Thanks for your question. Anderson, et al., 1996; Pinardi, et al., 2013 show typical HCHO diurnal variation of secondary sources, and my results are similar to what they reported.

**Changes in manuscript:** Thus, the diurnal variations in HCHO during all three episodes were similar to the typical patterns of secondary sources as reported in Anderson, et al (1996), Lee, et al (2015), Pinardi, et al (2013).

32、Page 11 Line 6 Many of the VCDs in the during, before and after APEC periods are equal within error. Are you referring to only the afternoon peaks HCHO values? The peak during APEC value appears to be equal within error with some of the highest post APEC values.

**Response:** Thank you very much for your question. Many of the VCDs in the before and after APEC periods are equal, but those all are higher than the value during the APEC. Please notes that the error bars in Fig. 11(Fig. 13 now) denote the standard deviation of HCHO VCDs, but not errors.

33、Page 11 Line 7 What are the actual values with associated errors and are they statistically different?

**Response:** Thank you very much for your question. The averaged VCD fitting errors of evening rush hours after and before APEC due to DOAS fit error here are 9.64% and 9.80%. Systematic error of the HCHO VCDs calculated by the geometric approximation is mostly smaller than 6% for the 15 ° elevation angle.

34、Page 11 Line 8 Please explain your reasoning.

**Response:** Thank you very much for your suggestion. After consideration, we decided to delete the conclusion that " The absolute HCHO values during the APEC period are obviously lower than those in the pre-APEC and post-APEC periods. The averaged HCHO during evening rush hours after APEC was higher than that before APEC. This finding is an interesting phenomenon, which may be related to some measures taken before the APEC.". Because this difference is small within the uncertainty range, the explanation is not reasonable.

35、Page 11 Line 14 Where were the in-situ ozone measurements located relative to the MAX-DOAS measurements? Put this information in methodology.

**Response:** Thank you for your advice. The MAX-DOAS instrument was deployed on the balcony (without a roof) of a classroom on the 4th floor in the laboratory building in the campus of UCAS (116.67 °E, 40.4 °N). And the UCAS supersite is on the top floor of the laboratory building, which is about ten meters away from MAX-DOAS. Ozone (O3) was measured by UV photometry (model 49i; Thermo Scientific), which is in the UCAS supersite. And I added the corresponding content in the article.

**Changes in manuscript: Sec.2.1:** The MAX-DOAS instrument was deployed on the balcony (without a roof) of a classroom on the $4^{th}$ floor in the laboratory building in the campus of UCAS (116.67 °E, 40.4 °N). The UCAS supersite is on the top floor of the laboratory building, which is about 10 m away from the MAX-DOAS instrument. Nitrogen oxide (NO, $NO_2$, and NOx) was measured by chemiluminescence (model 42i; Thermo Scientific), and ozone ($O_3$) was measured by UV photometry (model 49i; Thermo Scientific). These gas analyzers had precision values of 0.5 ppb and 0.4 ppb, respectively.

36、 Section 3.3 would benefit from a reorganization. Perhaps put information about primary versus secondary sources and the correlations first before making conclusions about diurnal trends.

**Response:** We carefully think about the suggestion, but we think our current typesetting is more logical. Firstly, the source of HCHO was implied by the observation of diurnal variation of HCHO, and then the correlation analysis was used to further support the speculation.

37、Section 3.4 How are VCDs calculated from the model output (i.e., what vertical height interval was integrated from the modeled vertical profile?) Section 3.4 Explain in more detail why the model poorly captures the local emissions. Could the lack of heterogeneous reaction in the model be contributing to the underestimation of the low HCHO values? Page 12 Line 14. Since the grid size seems to have little impact on the quality of the model output, is the "worse constraint" due to poor or outdated emission inventories local sources in this area? Are the highway emissions included in model calculations? How accurate is the emission inventory of the highway if it's included in the model?

**Response:** The vertical discretisation uses 60 levels up to the model top at 0.1 hPa (65 km) in a

hybrid sigma-pressure coordinate. The vertical extent of the lowest level is about 17 m; it is 100m at about 300m above ground, 400–600m in the middle troposphere and about 800m at about 10 km in height (Flemming, J., Huijnen, V., Arteta, J., Bechtold, P., Beljaars, A., Blechschmidt, A.-M., Diamantakis, M., Engelen, R. J., Gaudel, A., Inness, A., Jones, L., Josse, B., Katragkou, E., Marecal, V., Peuch, V.-H., Richter, A., Schultz, M. G., Stein, O., and Tsikerdekis, A.: Tropospheric chemistry in the Integrated Forecasting System of ECMWF, Geosci. Model Dev., 8, 975–1003, doi:10.5194/gmd-8-975-2015, 2015.). HCHO VCD is calculated by an integration of the modeled vertical profiles of HCHO.

The model doesn't consider the heterogeneous reaction. So the lack of heterogeneous reaction in the model could contribute to the underestimation of the low HCHO values.

The actual emission totals for 2008 inventory including anthropogenic, biogenic and natural sources and biomass burning. So highway emissions are considered in the 2008 inventory. Due to the establishment of the UCAS from 2013 and the holding of APEC meeting in 2014, the economy near the UCAS has grown rapidly, and the traffic flow has increased significantly in recent years. Thus, it could underestimate the highway emissions by using the 2008 inventory.

**Changes in manuscript:** The underestimation of the low HCHO values by the CAMS model compared to the MAX-DOAS measurements could be attributed to the lower constraint of local emissions in the model near the UCAS measurement station, and the lack of heterogeneous reactions in the model could also contribute to the underestimation of the low HCHO values. The China National Highway 111 is nearby and runs from north to south. The actual emission totals for the 2008 inventory included anthropogenic, biogenic and natural sources, and biomass burning, thus, highway emissions were considered in the 2008 inventory. However, due to the establishment of the UCAS from 2013 and the holding of the APEC meeting in 2014, the economy near the UCAS had grown rapidly, and the traffic flow had increased significantly in recent years. Thus, the use of the 2008 inventory could underestimate the highway emissions.

38、Page 13 Line 8 When you say "the primary HCHO is dominant" do you mean that the dominant contribution to the HCHO VCDs is the "local" primary emissions of HCHO?
Edit for further clarity.
**Response:** Thank you very much for your advice. Yes, "the primary HCHO is dominant" mean that the dominant contribution to the HCHO VCDs is the "local" primary emissions of HCHO. It is changed in the paper to make it clear.
**Changes in manuscript:** Thus, when the secondary source of HCHO is reduced, namely the "local" primary emissions of HCHO predominantly contribute to the HCHO VCDs, the difference between the MAX-DOAS observation and CAMS model is obvious.

39、Page 13 Line 14 Your conclusion is not necessarily sufficiently supported given the small R2 value and "reasonably" (too vague) would have to be defined before it is clear whether the data support this statement sufficiently.
**Response:** Thank you for your advice. R represents the ratio of HCHO VCDs in the morning (8:00LT)and noon (14:00LT). If R from the model is close to obtained from the MAX-DOAS, it indicates that the trend of diurnal variation of HCHO from the model simulation and MAX-DOAS observation is consistent. In other word, therefore the model can reasonably simulate the systematic diurnal variation of HCHO.

**Changes in manuscript:** R represents the ratio of HCHO VCDs in the morning (8:00 LT) and noon (14:00 LT). If $R_{Model}$ is close to $R_{MAX-DOAS}$, it indicates that the trend in diurnal variation of HCHO from the model simulation and MAX-DOAS observation is consistent, suggesting that the model can reasonably simulate the systematic diurnal variation in HCHO.

40、 Page 14 Line 8 You may want to add that, in contrast, correlation with NO2 was lower and what that implies. If VCDs are calculated from the 10° and 30° spectra, how do the values compare to the 15° spectra VCDs? Given that the geometric approximation becomes
worse under high aerosol conditions and these VCDs would be expected to diverge in
that case, comparison with the 10° and 30° spectra may be a good measure of the
validity of your use of the geometric approximation.

**Response:** Thank you very much for your advice. In order to constrain the systematic error of the grometric approximation, we compare HCHO VCDs calculated with the geometric approximation with those retrieved using PriAM profile inversion algorithm. The discussion is added in the manuscript as following.

**Changes in manuscript:** a. The systematic error of the HCHO VCDs calculated by the geometric approximation depends on the layer height of the TGs and aerosols. To evaluate the systematic error of the geometric approximation, we calculated more exact tropospheric HCHO VCDAMF using the PriAM inversion algorithm (Wang et al, 2017b). HCHO VCDgeo at elevation angles at 15° and 30° are usually obtained from the geometric approximation. The relative differences (Diff) between VCDAMF and VCDgeo for HCHO were calculated by Eq. (9):

$$Diff = \frac{VCD_{geo} - VCD_{AMF}}{VCD_{AMF}} \tag{9}$$

In Fig. 6, the average relative differences for elevation angles of 15° and 30° are shown as a function of the effective cloud fractions (eCF), as $0 < eCF \le 1$, $0 < eCF \le 0.3$, $0.3 < eCF \le 0.7$, and $0.7 < eCF \le 1.0$. The cloud fractions (eCF) are downloaded from the ECMWF CAMS model. It can be seen that the biases caused by the use of the geometric approximation are generally much smaller at EA=15° than at EA=30°, with the Diff being mostly smaller than 6% for the 15° elevation angle of and smaller than 16% for the 30° elevation angle in all periods. The bias for Diff caused by using the geometric approximation is about 2% (Ma et al., 2013; Wang et al., 2017c).

41、 Figure 11 Can you explain why the standard deviation of the pre-APEC time is so much smaller than the post-APEC period despite similar values?

**Response:** Thank you very much for your question. As we can see in figure 11, there are two obvious pollution process after APEC. The HCHO concentration shows a significant lifting process, which makes the standard deviation large.

42、 Figure 9 Since you show average values, how did the standard deviations of the averages compare to the retrieval errors? Are the larger of the two plotted as error bars?

**Response:** Thank you for your question. The standard deviations of the averages is larger than the retrieval errors. The fitting error is also added in the figure.

**Changes in manuscript:**

[Figure]

**Figure 13: Averaged diurnal variation in HCHO VCDs measured by MAX-DOAS in three episodes around APEC. The short cap width of the error bars denotes the one sigma standard deviations around the mean analysis values. The long cap width of the error bars denotes the fitting error.**

43、Figure 8 Why do moderate wind speeds appear to produce similar VCD values for all wind-directions. Also, why do southerly conditions appear to result in maximum VCDs occurred under the highest wind speeds given that you stated that high wind-speeds tend to reduce VCDs?

**Response:** Thank you for your question. I mean that transport from the south polluted air to observation site is easier under southerly winds with high wind speed.

**Changes in manuscript:** Under the southerly wind, the HCHO VCDs increase with increasing wind speed. Thus, transport from the south polluted air to the observation site occurs more easily under southerly winds with relatively high wind speeds.

Can you divide the VCD data into wind-speed and time of day and then see if there is a statistically significant reduction of the VCDs under non-Southerly wind conditions during APEC compared to before and after? That may help to determine how much the emissions controls impacted the VCDs independent of wind-direction.

**Response:** Thank you for your advice. We made the new plots following your suggestion (see attached figure) as Fig. 12 in the revised manuscript. We do not see the significant reduction of HCHO VCDs under non-southerly winds during APEC compared to before and after. However it is understandable. Because our station is in the north suburban area of Beijing city. The control measurements were mainly operated in the Beijing urban area. Therefore in order to evaluate the effects of control measures in the city, we need to compare HCHO observed at the suburban station under the southerly winds between different APEC periods, because pollutants in the city center can be transported to the measurement site under the southerly winds. Accordingly we modified the paragraph in Section 3.2

which has been given in the response to your point 27.

[Figure]

**Figure 12: Wind roses in (a) the "pre-APEC" , (b) the APEC and (c) the "post-parade" periods. Dependence of HCHO VCDs ($10^{15}$ molec cm$^{-2}$) on wind directions for different wind speeds in the pre-APEC (d), during the APEC (e) and post-APEC (f).**

**Response to RC2**

Technical Corrections:

1、Title: Consider adding VCDs after the word formaldehyde. Consider also including the APEC study to the title.

**Response:** Thank you very much for your suggestion. We have considered your advice, and it is added in the title.

**Changes in manuscript:** Ground-based MAX-DOAS observations of tropospheric formaldehyde VCDs and comparisons with the CAMS model at a rural site near Beijing during APEC 2014

2、General technical comment: when listing VCDs to route, please include the error values.

**Response: I don't understand your meaning. Do you mean adding the error after the VCDs in the text?**

3、Page 1 Line 22 Abstract: what are the units of HCHO and O3? VCDs? Mixing ratios?

**Response:** Thank you very much for your question. The units of HCHO are VCDs, and the units of O3 are Volume mixing ratio. It is added in the paper.

**Changes in manuscript:** Peak values of HCHO vertical column densities (VCDs) around noon and a good correlation coefficient $R^2$ of 0.73 between HCHO VCDs and surface $O_3$ concentration during noontime indicated that the secondary sources of HCHO through photochemical reactions of volatile organic compounds (VOCs) dominated the HCHO values in the area around UCAS.

4、Page 2 Line 8 What were the specific dates of the conference?

**Response:** Thank you very much for your suggestion. The specific dates of the conference are from November 5 to November 11, 2014. High emissions in Beijing and surround area were required to stop or limit their production during 3–12 November 2014.

**Changes in manuscript:** The 2014 Asia-Pacific Economic Cooperation (APEC) conference was held in the Huairou District of Beijing from November 5–11, 2014.

5、Page 2 Line 18 It is unclear what "traffic" and "regional" stations are?

**Response:** Thank you very much for your question. Wang et al., (2016a) selected five representative in-situ stations in different locations of Beijing, which represents different emission types and backgrounds. The traffic station is located in the Xizhimen Station of city center in Beijing with heavy traffic and traffic flow. The regional station is located in the suburbs of Beijing to reflect the impact of urban development on the suburban environment.

6、Page 3 lines 10 and 11 The meaning of this sentence is unclear.

**Response:** Thank you very much for your remind. It is changed to make it clear in the paper.

**Changes in manuscript:** Being a short lifetime oxidation product, long-living VOCs, such as methane $(CH_4)$, contribute to the background levels of HCHO (Pinardi et al., 2013; Stavrakou et al., 2009; Vrekoussis et al., 2010).

7、Page 3 Line 24 I believe this should say tropospheric column densities, surface mixing ratios, and vertical profiles of aerosol extinction and trace gas mixing ratios.

**Response:** Thank you very much for your remind. It is changed in the paper.

**Changes in manuscript:** The information obtained from MAX-DOAS measurements includes tropospheric column densities, surface mixing ratios, and vertical profiles of aerosol extinction and trace gas mixing ratios.

8、Page 4 Line 18 Consider changing to "derived from the DOAS spectral analysis [of the measured spectra]"

**Response:** Thank you very much for your advice. It is changed in the paper.

**Changes in manuscript:** In the first step, the differential slant column densities (dSCDs) were derived from the DOAS spectral analysis with a so-called FRS and measured in a small sun zenith angle at 90° elevation around noon (Hermans et al., 2003; Hönninger and Platt, 2002; Kraus, 2006).

9、Page 6 Line 22 In this sentence and figure 4 you use different terms for the blue and red lines: blue (measured, derived ) red (calculated, retrieved, fitted).

**Response:** Thank you very much for your suggestion. It is changed in the paper. Now figure 4 is changed to figure 5.

**Changes in manuscript:** Figure 5: Example of a DOAS fit of a spectrum to retrieve the slant column densities of HCHO; the red and blue curves indicate the fitted absorption structures and the derived absorption structures from the measured spectra, respectively.

Pick one term for each and ensure that the meaning of the term is clear. What about the contribution of the residual to the blue line?

**Response:** Thank you very much for your remind. The residual represents the remaining structure after the measured spectrum (blue line) minus the fitted absorption structures (red line). The smaller the residual, the better the spectral fit

10、Page 10 Line 4 the sentence needs editing for greater clarity and to appropriately describe figure 9.

**Response:** Thank you very much for your suggestion. It is changed in the paper.

**Changes in manuscript:** The result shows a "fluctuating effect" with the HCHO VCDs increasing abruptly over several days and dropping sharply for a few days during the APEC summit.

11、Page 13 Line 24 HCHO were also studied before and after APEC, were they not?

**Response:** Thank you very much for your suggestion. It is changed in the paper.

**Changes in manuscript:** We studied the tropospheric HCHO VCDs at the UCAS site in Huairou District, Beijing around the APEC summit based on the MAX-DOAS measurements from October 1, 2014 to December 31, 2014.

12、Table 1 There are small spacing and English errors.

**Response:** Thank you very much for your remind. It is changed in the Table 1.

**Changes in manuscript:**

| Spectrometer | | Azimuth | Elevation | Temperature | Location | | Measuring time |
|---|---|---|---|---|---|---|---|
| Name | Maya (Ocean Optics) | 0 ° | 3 °; 5 °; 10 °; 15 °; | 20 ℃ | Site | Yanxi Lake campus of UCAS | 6:30-18:30 |
| Spectral range | 290– 420 nm | | 30 °; 90 °; | | Longitude | 116.67 °E | |
| FWHM | 0.5 nm | | | | Latitude | 40.4 °N | |

13、Clarity of Figure 9 may be improved by lines or boxes that indicate the afternoon period.

**Response:** I think maybe you give the wrong number of the figure.

14、Figure 6 If relative humidity is not discussed in the results or discussion, perhaps remove it from the figure to have more space to expand the more relevant data.

**Response:** Thank you very much for your suggestion. We have considered your advice, and it is removed in the Figure 6.

**Changes in manuscript:**

[Figure]

15、Supplement: More helpful analysis may be achieved by dividing the regressions into bins that do not all include zero cloud fraction. For example, are different trends observed for eCF 0-0.3, 0.3-0.5, 0.5-0.7 etc.?

**Response:** The figure followed shows the result of dividing the regressions into bins of eCF. And the figure also supports the conclusions in our text. The figure is added in the supplement.

**Changes in manuscript:**

[Figure]

**Figure S3: Correlation between HCHO VCDs retrieved from the MAX-DOAS measurements and those obtained from the CAMS model data for 0<eCF ≤1 (a), 0<eCF≤0.3(b), 0.3<eCF ≤ 0.7 (c), and 0.7<eCF≤1.0 (d) at 8:00 LT from October to December 2014.**

[Figure]

**Figure S4: Correlation between HCHO VCDs retrieved from the MAX-DOAS measurements and those obtained from the CAMS model for 0<eCF ≤1 (a), 0<eCF≤0.3(b), 0.3<eCF ≤0.7 (c), and 0.7<eCF≤1.0 (d) at 14:00 LT from October to December 2014.**

Major

(1) Use of the Geometric Approximation Only

P5, 8: Your comment, ": : :it has lower systematic errors because of the geometrical approximation" needs to be backed up. Do you have a reference, maybe a short explanation of your reasoning? You need to somehow prove to me that it is better to use a geometric approximation of the VCD vs. one of several RTM/inversions approaches. You did not do this here, but rather allude to some other studies. I am not convinced that the geometric approximation is the best? Prove me wrong?

**Response:** Thank you for your question. This sentence is not very clear so that caused some misunderstandings for you. We make changes in the paper to make it clearer. 1) we mean that: According to Ma et al., 2013 and Wang et al., 2017c, they found the systematic error is larger for larger elevation angles and larger RAA. So this study uses the geometric approximation method to determine HCHO VCDs at an elevation angle of 15 ° to avoid surface obstacles on light paths along the line of sight, at the same time, it has lower systematic errors at 15 ° than at 30 °. 2) And we add the discussion of the error budgets for geometric approximation in section 2.3. It also shows the systematic errors at 15 ° is smaller than at 30 °by using the geometrical approximation.

**Changes in manuscript:** Lower systematic errors were achieved at 15 ° than at 30 ° by using the geometrical approximation (discussed in Section 2.3 below).

Alternatively, you could provide a comparison of your VCDs with RTM-inversion derived HCHO VCDs. I would also suggest that you provide a comparison of your HCHO VCDs to those measured via satellite. This would give me more confidence in your conclusions

**Response:** Thank you very much for your suggestion. We have considered your advice, and I add the comparison of geometrical approximation with inversions approaches at section 2.3.

**Changes in manuscript:**

**2.3 Error budgets**

The following error sources were considered as the error estimates for the MAX-DOAS results:

a. The systematic error of the HCHO VCDs calculated by the geometric approximation depends on the layer height of the TGs and aerosols. To evaluate the systematic error of the geometric approximation, we calculated more exact tropospheric HCHO VCDAMF using the PriAM inversion algorithm (Wang et al, 2017b). HCHO VCDgeo at elevation angles at 15 ° and 30 ° are usually obtained from the geometric approximation. The relative differences (Diff) between VCDAMF and VCDgeo for HCHO were calculated by Eq. (9):

$$Diff = \frac{VCD_{geo} - VCD_{AMF}}{VCD_{AMF}} \qquad (9)$$

In Fig. 6, the average relative differences for elevation angles of 15 ° and 30 ° are shown as a function of the effective cloud fractions (eCF), as 0<eCF ≤1, 0<eCF≤0.3, 0.3<eCF ≤0.7, and 0.7<eCF≤1.0. The cloud fractions (eCF) are downloaded from the ECMWF CAMS model. It can be seen that the biases caused by the use of the geometric approximation are generally much smaller at EA=15 ° than at EA=30 °, with the Diff being mostly smaller than 6% for the 15 ° elevation angle of and smaller than 16% for the 30 ° elevation angle in all periods. The bias for Diff caused by using the geometric approximation is about 2% (Ma et al., 2013; Wang et al., 2017c).

b. The fitting error of the DOAS fit is derived from the dSCD fitting error to VCD error by using geometric approximation, as

$$VCD_{fittingerror} = \frac{VCD_{error}}{VCD} = \frac{\sqrt{2(dSCD_{\alpha \neq 90° \ fittingerror}^{2} + dSCD_{\alpha = 90° \ fittingerror}^{2})}}{2 \ (\frac{1}{\sin \alpha} - 1) \times VCD} \qquad (10)$$

and the hourly average of the HCHO VCD fitting error was from 3.61% to 27.19% for the entire period.

c. Cross section error also constitutes one of the error sources. Some previous research reported that cross section errors of $O_4$ (aerosols) and HCHO are 5% and 9%, respectively (Bogumil et al., 2003; Meller and Moortgat, 2000;Thalman and Volkamer, 2013; Vandaele et al. 1998 ). Wang et al (2017b) estimated the errors related to the temperature dependence of the cross sections, and the corresponding systematic error of HCHO was estimated to up to 6%.

Since the three errors are mainly independent, the total error can be calculated by combining all the above error sources, adding up to about 12% on average.

(2) Emission Totals from 2008 for CAMS model P7, l12: You use emission totals from 2008. Your year of study is 2014, that is a difference of 6 years and a lot can change. Why didn't you use a more recent emission inventory? Is there one? If so, why didn't you use it?

**Response:** Thank you for your question.   The simulation work of the model is made by the ECMWF, and we download the data from CAMS real-time products in ECMWF (http://apps.ecmwf.int/datasets/data/cams-nrealtime/levtype=sfc/). Besides, the MAX-DOAS data can verify the model. We also make the conclusion that inventory needs to be updated according to our comparative study. Annual emissions from anthropogenic, biogenic and natural sources and biomass burning for 2008 in Tg for a composition Integrated Forecasting System (C-IFS) (CB05) run at T255 resolution (Flemming et al., 2014). The 2008 global emissions is used as a total amount of emissions to

assimilate data in the C-IFS model. Then, for their near real-time data, they will be added to the latest satellite observation data for assimilation.

(3) HCHO VCD error

P9, 111-120: It is likely that when the wind comes from the south it is more polluted than when the wind comes from the north. However, an average HCHO VCD of 7.57*1015 vs. 6.64*1015 is hardly conclusive. This is a 14% difference. WHAT IS THE ERROR OF YOUR VCD? I would estimate that is it a least 10%, likely over 20%. As such, your statistics here are weak. Please define the error of your VCDs and then re-word this section. For example, in Figure 9, you have error bars on your VCDs, but no mention of how you calculate them [they also look very low to me].

**Response:** Thank you for your advice. Although the uncertainty of HCHO VCD is about 6% for the 15 ° elevation angle, the uncertainty is comparable to the systematic difference of HCHO under different wind fields. However, uncertainty effects on systematic bias can be averaged as zero for a long-term measurements, therefore the systematic differences of HCHO VCDs still considerably indicate that more pollutants are transported from the southern region.

(4) Figure 12 – Correlation Analysis

I understand what you are trying to achieve here. However, I am not sure why you choose the period Oct 26 – Nov 20, 2014? This seems random? Why not use all your data?

**Response:** Thank you for your question. Because in the previous study around APEC, the period used was from October 26 to November 20, 2014, so the analysis here we used the corresponding period. However we have considered your advice, and I use all the data for the correlation analysis. The change is made in the paper. The new correlation analysis indicates that the correlation coefficients between HCHO VCD and NO2 VCD at rush hour and between HCHO VCD and O3 during the noon time are slightly reduced. However, the results still show high correlation between HCHO VCD and O3 during the noon time and low correlation between HCHO VCD and NO2 VCD at rush hour.

The NO2 VCD is not described. Is it VCDgeo? Is it data from the same instrument and time? Did you also compare your O3 with the 7-9 & 16-18 periods? You don't have to show the plot but I would like to know the R of that? Hopefully it is very low to prove your point. Similarly, did you compare the NO2 VCDs with the HCHO VCDs from 11-14 period. You need a more complete assessment here to really prove your point.

**Response:** Thank you for your advice. The NO2 VCD is VCDgeo and the data is from the same MAX-DOAS instrument and time. All the suggested comparisons are added in the paper and please see the following changes.

**Changes in manuscript:**

Determining pollution sources is crucial to controlling air pollution. Three time intervals were used for determining the main HCHO sources. The first interval was defined as noontime from 11:00–14:00 and is associated with strong photochemical reactions. The second and third intervals were defined as the morning rush hour from 7:00–9:00 and the evening rush hour from 16:00–18:00. To further determine whether the pollution sources of HCHO at UCAS were primary or secondary formations from other VOCs, the correlations of HCHO with the primary pollutant $NO_2$ or secondary pollutant $O_3$ were analyzed (Anderson et al, 1996; Possanzini et al., 2002). Surface $O_3$ data were obtained from *in situ*

measurements in the UCAS supersite, and troposphere $NO_2$ VCD data were retrieved from the same MAX-DOAS measurements using geometric approximation. The linear correlations of noontime average HCHO VCD with $NO_2$ VCD and $O_3$ from 11:00–14:00 and rush hour average HCHO VCD with $NO_2$ VCD and $O_3$ from 7:00–9:00 and 16:00–18:00 are shown in **Fig. 14**. Direct analysis of the data indicates that noontime average HCHO had a higher correlation coefficient with $NO_2$ VCD and $O_3$ than rush hour. This implies that a small amount of HCHO comes from the traffic emissions during rush hour. A good correlation coefficient $R^2$ of 0.73 was found between HCHO VCD and $O_3$ during the noontime, which indicates that the main source of HCHO was from secondary photo-oxidation formation at noon. In contrast, a correlation coefficient of 0.38 between HCHO VCD and $NO_2$ VCD during noontime was better than during rush hour ($R^2$=0.06), which may be due to the contribution of vehicle emissions to HCHO precursors. A longer $NO_2$ lifetime with less dispersion efficiency in winter and HCHO from continuously generated photo-oxidation contributed to the higher correlation between HCHO VCD and $NO_2$ VCD at noon higher than during rush hour. The transport of $NO_2$ and VOC may constitute one of the causes. The VOCs from transport generate HCHO due to strong photo-oxidation at noon. This result indicates that secondary photo-oxidation formation of HCHO from other VOCs should be the dominant source at UCAS.

[Figure]

**Figure 14: Scatter plots and linear regressions (a) of noontime average HCHO VCD measured by MAX-DOAS against $O_3$ VMRs measured by a stationary ozone monitoring instrument, and (b) rush hour average HCHO VCD against $NO_2$ VCD measured by MAX-DOAS from October to December 2014.**

What happens if the R value for O3 and 7-9&16-18 periods is also high? I believe you have something here but be careful about how you present it. I also need to know

exactly where your O3 monitor is, is it at ground-level?

**Response:** Thank you for your question. The R2 value for O3 and HCHO at 7-9&16-18 periods is 0.03. The MAX-DOAS instrument was deployed on the balcony (without a roof) of a classroom on the 4th floor in the laboratory building in the campus of UCAS (116.67 °E, 40.4 °N). And the UCAS supersite is on the top floor of the laboratory building, which is about ten meters away from MAX-DOAS. Ozone (O3) was measured by UV photometry (model 49i; Thermo Scientific), which is in the UCAS supersite. And we add the corresponding content in the revised manuscript.

(5) Assumption that the HCHO VCD is the correct result

On P12, l3 you state that the CAMS model UNDERESTIMATES : : :.

How do you know this? How do you know the MAX-DOAS result is the correct result and better than the CAMS model? What other VALIDATION do you have? Did you compare it to the satellite data; ground-data extrapolated to a column {see comment 1}? You may be right, but you may also be wrong. I am not convinced, especially without any error analysis of your HCHO VCDs or CAMS model. I would say that your CAMS model could be really off since it uses emission totals from 2008. Maybe the emission estimates in the model for 2008 are simply much lower than the 2014 values?

You allude to this on P12, l14-15, right?

**Response**: Thank you for your question. According to your advice, we evaluate the systematic error of the geometric approximation by comparing the VCD calculated using the geometric approximation and those retrieved using a PriAM profile inversion algorithm.. The new discussion is added in Section 2.3. The result shows that the systematic error is less than 6% for the elevation angle of 15 degrees. Besides, satellite retrievals of HCHO have more problem than MAX-DOAS measurements. MAX-DOAS is an usual technique to validate the HCHO satellite data (cite: De Smedt, I., Stavrakou, T., Hendrick, F., Danckaert, T., Vlemmix, T., Pinardi, G., Theys, N., Lerot, C., Gielen, C., Vigouroux, C., Hermans, C., Fayt, C., Veefkind, P., Müller, J.-F., and Van Roozendael, M.: Diurnal, seasonal and long-term variations of global formaldehyde columns inferred from combined OMI and GOME-2 observations, Atmos. Chem. Phys., 15, 12519-12545, doi:10.5194/acp-15-12519-2015, 2015.). In addition MAX-DOAS retrievals of HCHO have been well proved and evaluated in the previous study. Wang et al., 2017b retrieved tropospheric HCHO VCDs and vertical profile in Wuxi from 2011 to 2014, and the DOAS fit setting derived from the formaldehyde slant column measurements during CINDI: intercomparison and analysis improvement. Therefore MAX-DOAS results of HCHO are valuable and sufficiently confident to be used for validation of model simulations.

For the old emission inventory, the inventory is used by the operational CAMS model. We agree it could be lower than the current emission. The conclusion is also our finding by comparing MAX-DOAS measurements with the model data.

P12, l3: What do these ranges mean? Is it due to different grid-sizes?

**Response:** Thank you for your question. These ranges are due to the different grid-sizes. We do some change in the paper to make it clear.

**Changes in manuscript:** On average, the CAMS model underestimated HCHO VCDs by 1.56–2.02 × $10^{15}$ molec cm$^{-2}$ and 1.27–2.12 × $10^{15}$ molec cm$^{-2}$ compared to the MAX-DOAS measurements at 8:00 LT and 14:00 LT, respectively, due to different grid-sizes.

(6) RMAX-DOAS vs Rmodel

P13, l10: You R concept is interesting. Based on this I would think that R(DOAS) should be higher than R(model) for cases when the temp is cold (and secondary is HCHO is lower than predicted via the model), do you see this? Alternatively, if primary HCHO emissions are under predicted in the model R(DOAS) again would be higher than R(model) right? So what does this R concept really tell us? A graph like Figure 15, does not tell me much? However, if you separate out case studies maybe you get some more information.

P14, l21-23: If the CAMS model underestimates primary sources of HCHO then R(DOAS) > R(model) but "under a situation with a low temperature when the production rate of secondary HCHO is relatively low" won't the CAMS model also underestimate the secondary HCHO production also causing R(DOAS) > R(model) as well? What is the assumed temp in the model, or does it use real-time met-data? How do we know what is the problem, is it a problem with the assumed temp, if so can you adjust that to check? OR is it a problem with the emissions inventory (perhaps a bigger issue). Again, the above concept seems to have merit, but you need to develop this and explain it further, because I am somewhat confused. Also, despite your analysis I have no feeling as too how much HCHO is secondary and how much is primary (and isn't that what the R calculations are for?).

**Response:** Thank you for your suggestions. Here are some explanations for your questions. We agree on your conclusion, if there is big bias in the model simulations of the secondary production of HCHO, it can also cause deviations of R(DOAS) and R(model).

Following your suggestion:

1) we separate the plots in the periods of October to November and for December (see below). But there is no significant difference between the two periods.

[Figure]

[Figure]

**Scatter plots and linear regression of** $R_{Model}$ **against** $R_{MAX\text{-}DOAS}$ **(see the text) from October to December 2014 (a) and from October to November and for December due to the changing of temperature (b). There are not significant differences between the two periods.**

2) we check the source of meteorological data in the model. The CAMS global real-time production system uses all the meteorological observations from the ECMWF numerical weather prediction system, which is extracted from satellite real-time meteorological data. We also compared the temperature in the model with in-situ measurements. The results are shown in the response according

to your point "P12, l124" in the minor comment. Generally good agreement can be seen. Therefore the model simulations could predict the secondary formation of HCHO well, but it can't be confirmed.

Based on the two further analysis, we noticed that the diurnal variation of HCHO is a mixed effect of primary emission, secondary formation, and probably also meteorology. It is impossible to gain the conclusion that which is the factor which causing the deviation of R(DOAS) and R(model). Therefore the R comparisons only generally evaluate the quality of model simulations on diurnal variations of HCHO. As you asked, both underestimation of primary emission and overestimation of secondary emission by model simulations can cause the similar fact that R(DOAS)>R(model). We can not firmly conclude which is the reason. And the method can't give quantified conclusion of HCHO source. Therefore we add a clarification in the revised manuscript.

**Changes in manuscript:** It needs to be noted that the diurnal variation in HCHO is the result of the combined influence of primary emissions, secondary formation, and meteorology. We found that $R_{MAX-DOAS}$ was generally larger than $R_{Model}$. However, it was impossible to determine the factor causing the deviation in $R_{MAX-DOAS}$ and $R_{Model}$. Therefore, the R comparisons generally only evaluate the quality of the model simulations on diurnal variations in HCHO.

**Minor**

P2, l3: {Q} Is the correlation coefficient (R=0.83)? If so, say (R=0.83, not ~0.83)

**Response:** Thank you for your suggestions. Correlation between HCHO VCDs retrieved from the MAX-DOAS measurements and those obtained from the CAMS model at 8:00 LT and 14:00 LT from October to December 2014 in different grids were compared. The correlation coefficient R is more than 0.83, So we use ~0.83. And it is changed in the paper.

**Changes in manuscript:** The HCHO VCDs of the CAMS model and MAX-DOAS were generally consistent with a correlation coefficient $R^2$ greater than 0.69..

P2, l14: {Q} How is "APEC blue" defined? Perhaps a brief statement of how the actual reduction strategies were defined and the defined APEC levels would be useful? Is there an APEC-red for example?

**Response**: Thank you for your suggestions. For the sake of guaranteeing the smooth convening of the APEC meeting, China took a series of effective measures which played a prominent role in improving the air condition in Beijing and surrounding regions. As a result, a better quality environment emerged, which we called "APEC-Blue". This is reported in the Chinese website (https://baike.so.com/doc/7519682-7792600.html).
The actual reduction strategies were added in the paper to make it clear.

**Changes in manuscript:** Since November 1, 2014, parts of the Jing-Jin-Ji region and surrounding areas had begun to implement an emission reduction plan according to the APEC conference air quality assurance policy. Formal emission reduction measures were implemented in the Jing-Jin-Ji region and surrounding areas from November 3 and included limiting the production of factories, shutting down construction sites, implementing traffic restrictions based on even- and odd- numbered license plates, and improving road cleaning (Wang et al., 2016). In response to the possible adverse weather conditions from November 8–10, the "enhanced emission reduction measures" were implemented in the Jing-Jin-Ji region and surrounding areas from November 6. These various efforts coupled with relatively favorable weather conditions than previous years resulted in the emission reduction measures

having significant effects. Based on estimations, all types of main pollutants were reduced by over 40% in Beijing and by over 30% in other provinces, through these measures (Wang et al., 2016).

P2, l16-17: {Q} Do you or the authors of the Wang et al. make any conclusion as to why the O3 rose to 189%? Does this have to do with being in a NOx-limited or VOC-limited regime?

**Response**: Thank you for your question. Wang et al.,2016a gave the reason that the O3 in urban and suburban areas of Beijing is mostly in the control area of VOCs. The possible reason for the increase of O3 is that the emission control measures of NOx are greater than the emission control measures of VOCs, which leads to the weakening of the inhibition of O3 formation by NOx, resulting in significant increasing of O3 concentration. And it is added in the paper to make it clear.(Besides, the introduction is reorganization to make it more logical. )

**Changes in manuscript:** Wang et al (2016a) selected five representative *in situ* stations in different locations in Beijing and found that average concentrations of $SO_2$, $NO_2$, $PM_{10}$, and $PM_{2.5}$ decreased by 61.5%, 40.8%, 36.4%, and 47.1%, respectively, whereas the average concentration of $O_3$ increased by 101.8%, compared with the same period over the last five years ($PM_{2.5}$ since 2013). $O_3$ in urban and suburban areas of Beijing is mostly in the control area of volatile organic carbons (VOCs). The possible reason for the increase in $O_3$ is that the emission control measures of NOx are greater than the emission control measures of VOCs, which leads to the weakening of the inhibition of $O_3$ formation by NOx, resulting in significant increases in $O_3$ concentration.

P2, l25-P3, l1: {Q} What were Zhang's conclusions (briefly)?

**Response**: Thank you for your advices. We have considered your advice, and we add the Zhang's conclusions to make it clear. During the APEC conference period, the average concentration of PM2.5 was 37.7 ±35.4 mg/m3, which was 48% and 54% lower than that of BAPEC and AAPEC period, respectively. Compared with ultrafine particles (<100 nm), the number concentration of accumulation mode and coarse mode particles experienced more significant decreases by 47% and 68%, indicating that particles with larger sizes were better controlled during the APEC period.

**Changes in manuscript:** Zhang et al (2017) analyzed the characteristics of aerosol size distribution and the vertical backscattering coefficient profile during the 2014 APEC summit using lidar observation. Particles with larger sizes were better controlled during the APEC period, with the number concentration of accumulation mode and coarse mode particles experiencing more significant decreases of 47% and 68% (Zhang et al., 2017).

P3, l1-2: ADD {REFS} for the published studies here.

**Response**: Thank you for your advice. The REFS were added in the paper.

**Changes in manuscript:** Published studies have focused mainly on the effects of commonly measured gas pollutants, particulate matter, and aerosols, but not HCHO (Cheng et al., 2016; Fan et al., 2016; Huang et al., 2015; Li et al., 2016; Liu et al., 2016; Meng et al., 2015; Tang et al., 2015; Chen et al., 2015; Wang et al., 2016a; Wang et al., 2016b; Wang et al., 2017a; Wei et al., 2016).

P3, l5: {REF} is not in your final reference list.

**Response**: Thank you for your remind. We are so sorry for making this mistake. The REF is added in the final reference list..

**Changes in manuscript:** Fried A, Cantrell C, Olson J, Crawford J H. Detailed comparisons of airborne formaldehyde measurements with box models during the 2006 INTEX-B and MILAGRO campaigns: potential evidence for significant impacts of unmeasured and multi-generation volatile organic carbon compounds, Atmos. Chem. Phys., 11, 9887-9957, 2011.

P3, l14-15: HO2 and OH are radicals not ions, please correct this.

**Response**: Thank you for your remind. We are so sorry for making this mistake. It is corrected.

**Changes in manuscript:** As an active gas, HCHO can be photolyzed to generate $HO_2$ free radicals. $HO_2$ rapidly and radically reacts with NO to generate OH, which can influence the oxidation ability of the atmosphere.

P3, l22: fix {REF}, you mean Honninger et al., 2004 right?

**Response**: Thank you for your remind. We are so sorry for making this mistake. Yes, it is corrected in the paper.

**Changes in manuscript:** A type of passive differential optical absorption spectroscopy system, called Multi-axis Differential Optical Absorption Spectroscopy (MAX-DOAS), has been used over the past decade to measure tropospheric trace gases (Honninger et al., 2004; Wagner et al., 2004; Sinreich et al., 2005; Wagner et al., 2007; Vigouroux wt al., 2009).

P4, l6: I would call it the Beer-Lambert Law

**Response**: Thank you for your advice. It is corrected in the paper.

**Changes in manuscript:** MAX-DOAS, which is an optical remote-sensing technology that records the spectra of scattered sunlight at different elevation angles, can be used to quantitatively measure trace gases based on Beer-Lambert Law (Hönninger and Platt, 2002; Bobrowski et al., 2013; Roozendael et al., 2003; Trebs et al., 2004; Hönninger et al., 2004; Wagner et al., 2004).

P4, l9: {Q} Were clouds a factor? How often was it cloudy? Was the data pre-screened in any way?

**Response**: Thank you for your question. In this paper, the effects of different cloud coefficients on MAX-DOAS inversion VCDs and the HCHO VCDs from MAX-DOAS and CAMS model under different cloud coefficients are both compared. It is found that the cloud coefficient has negligible influence on it. During the entire APEC period, it is basically sunny and cloudless weather. The data is pre-screened. The spectrum with too small a light intensity and an excessive integration time are removed.

Figure 1: Change the colour red on your figure, it is hard to read.

**Response**: Thank you for your advice. It is corrected in the paper.

**Changes in manuscript:**

[Figure]

P5, l21: : : :some point sources (e.g. XX and YY). Add some key examples, factories
or power plants?

**Response**: Thank you for your advice. Some point sources here mean stationary sources from the rural settlement. They are not factories and power plants. And it is added in the paper.

**Changes in manuscript:** The site is mainly influenced by emissions from vehicles on China National Highway 111 that runs from the north and south as well as some stationary sources from the rural settlements across the highway (Zhang et al., 2017).

Figure 2: fix the text on your figure (e.g. spectrograph as one word)

**Response**: Thank you for your suggestion. It is changed in the paper.

**Changes in manuscript:**

[Figure]

P5, l24: {C} change stepping motor to stepper motor?

**Response**: Thank you for your advice. It is changed in the paper.

**Changes in manuscript:** This system comprises a telescope, stepper motor, spectrometer, and

computer.

P6, l2: {Q} Why was the temp set to 20C?

**Response**: Thank you for your question. The changing ambient temperature in China is from -15 ℃ to 35 ℃ in a year. And the weather in spring and autumn is a little longer in Beijing with the temperature around 20 ℃. So we set the temp as 20 ℃ to make sure a stable temperature in all seasons.

**Changes in manuscript:** The spectrometer was placed in a temperature-controlled box at 20℃ to ensure that the spectrograph could work at a stable temperature under the changing ambient temperature from -15 ℃ to 30 ℃ in China.

P6, l10: replace scanning times with SCANS

**Response**: Thank you for your advice. It is changed in the paper.

**Changes in manuscript:** Each measurement had an average of 100 SCANS, and the integration time was adjusted automatically based on the light intensity.

Figure 3: replace a1, a2, etc. with a3 a30 a90 etc. {you don't need to number each one, simply add elevation angles to the alpha directly}

**Response**: Thank you for your advice. It is changed in the figure 3.

**Changes in manuscript:**

[Figure]

Table 1: fix text .. Longitude – one word, {Q} What is the MAYA? Is that Ocean Optics? If so, add that.

**Response**: Thank you for your advice. It is changed in the table. And Maya is a Ocean Optics spectrometer (https://oceanoptics.com/product/maya2000-pro-custom/). The briefly introduction of Maya is added in the paper.

**Changes in manuscript:**

| Spectrometer | Azimuth | Elevation | Temperature | Location | Measuring |
|---|---|---|---|---|---|

| | | | | | | | time |
|---|---|---|---|---|---|---|---|
| Name | Maya (Ocean Optics) | 0 ° | 3 °; 5 °; 10 °; 15 °; 30 °; 90 °; | 20 ℃ | Site | Yanxi Lake campus of UCAS | 6:30-18:30 |
| Spectral range | 290–420 nm | | | | Longitude | 116.67 °E | |
| FWHM | 0.5 nm | | | | Latitude | 40.4 °N | |

The spectrometer was produced by Ocean Optics and was named Maya (https://oceanoptics.com/product/maya2000-pro-custom/). The spectrometer covers the range of 290 nm to 420 nm, and its instrumental function is approximated as a Gaussian function with a full width at half maximum (FWHM) of 0.5 nm.

P6, 19: replace Doasis with DOASIS

**Response**: Thank you for your advice. It is changed in the paper.

**Changes in manuscript:** The ring structure (Fish and Jones, 2013), which is used to account for rotational Raman scattering effects, was calculated using DOASIS software (Kraus, 2006) based on the FRS and was included in the fit.

P6, l22: replace [derived] with [measured] {as you did in your Figure4}

**Response**: Thank you for your advice. I changed the description of figure 4 here. Now figure 4 is changed to figure 5 because some new figure is added.

**Changes in manuscript:** Figure 5: Example of a DOAS fit of a spectrum to retrieve the slant column densities of HCHO; the red and blue curves indicate the fitted absorption structures and the derived absorption structures from the measured spectra, respectively.

Figure 6 shows the period of 3-8 November. Why didn't you use the period of 3-12 November (the whole APEC period)?

**Response**: Thank you for your question. This data is used to support the analysis of the transport event. So the meteorological data for the time period corresponding to the transport event is displayed, which is the period of 3 to 8 November, 2014.

P8, l11: You describe 2 peaks on Nov 4 and Nov 7, but what about Nov 3, as seen on Figure 6 that actually has the HIGHEST HCHO VCDs?

**Response**: Thank you for your advice. Two daily averaged HCHO VCD peaks were on Nov 4 and Nov 7, and the rise process of Nov 4 is from the evening of Nov 3.

Figure 7: perhaps replace UTC time with LT for consistency.

**Response**: Thank you for your suggestion. I changed in the paper.

**Changes in manuscript:**

[Figure]

Figure 9: Error bars equal retrieval error. {Q} How is this calculated?

**Response**: Thank you for your question. We have considered your question, and we think it is more reasonable to use the standard deviation to represent the error bars. It is changed in the paper. And the figure 9 is changed to figure 11 due to some new figures were added. And about the retrieval error, we discuss at the section 2.3.

**Changes in manuscript:**

[Figure]

**Figure 11: Daily averaged values of HCHO VCDs from October 26, 2014 to November 20, 2014. Error bars denote standard deviations.**

P10, l19-20: Is there any way to determine which is more important, the control measures of the meteorology? Perhaps a longer term study? Please comment.

**Response**: Thank you for your suggestion. Comparisons of transports events between from the polluted south area and clean north area indicate meteorology condition can vary HCHO amounts by about 50%. However meteorology condition is not under control. Reduction of HCHO emission in the south polluted area can be estimated by ~20% due to control measures of emissions. The significant effects of control measures are important for improving air qualities, especially under a meteorology condition which obstructs depositions of pollutants.

P11, l7-8: Could this have to do with a change in NOx-limiting vs. VOC-limiting cases? Please advise.

**Response**: Thank you for your advice. We carefully think about your advice and do some research on previous literature. Wang et al., 2009 found that ozone formation is mainly controlled by VOCs in the near-suburbs of Beijing City and its high-value ozone areas in the downwind direction. In suburban counties and rural areas, the sensitivity of ozone generation to NOx becomes important. And the UCAS is located in the outer suburbs of Beijing, in other word, the UCAS belongs to the NOx-limiting area. During APEC, the NOx concentration gradually decreases due to the control measurement. As a results, the HCHO decreases. After APEC, control measurements are abolished, HCHO concentration is increased with the increasing of NOx. There should be not a change in NOx-limiting vs. VOC-limiting cases. So we can't draw the exact conclusion. On the other hands, according to the recommendation of reviewer 1, we seriously discussed it. The SNR in evening is low that makes the data not very credible. So I decided to remove this part from the text.

P11, l14: Where was the surface O3 measurement location exactly? What type of NO2 VCD was it, geo-approximated, same instrument and location? Please describe.

**Response**: Thank you for your advice. The MAX-DOAS instrument was deployed on the balcony (without a roof) of a classroom on the 4th floor in the laboratory building in the campus of UCAS (116.67 °E, 40.4 °N). And the UCAS supersite is on the top floor of the laboratory building, which is about ten meters away from MAX-DOAS. The O3 was measured by UV photometry in the UCAS supersite. The NO2 VCD was obtained from the MAX-DOAS observation by using geo-approximated. The description was added in the paper.

**Changes in manuscript:** The MAX-DOAS instrument was deployed on the balcony (without a roof) of a classroom on the 4[th] floor in the laboratory building in the campus of UCAS (116.67 °E, 40.4 °N). The UCAS supersite is on the top floor of the laboratory building, which is about 10 m away from the MAX-DOAS instrument. Nitrogen oxide (NO, $NO_2$, and NOx) was measured by chemiluminescence (model 42i; Thermo Scientific), and ozone ($O_3$) was measured by UV photometry (model 49i; Thermo Scientific). These gas analyzers had precision values of 0.5 ppb and 0.4 ppb, respectively.

**Sec. 3.3 :** Surface $O_3$ data were obtained from *in situ* measurements in the UCAS supersite, and troposphere $NO_2$ VCD data were retrieved from the same MAX-DOAS measurements using geometric approximation.

P11, 118: Too many significant figures!

**Response**: Thank you for your advice. It is changed in the paper.

**Changes in manuscript:** Direct analysis of the data indicates that noontime average HCHO had a higher correlation coefficient with $NO_2$ VCD and $O_3$ than rush hour. This implies that a small amount of HCHO comes from the traffic emissions during rush hour. A good correlation coefficient $R^2$ of 0.73 was found between HCHO VCD and $O_3$ during the noontime, which indicates that the main source of HCHO was from secondary photo-oxidation formation at noon. In contrast, a correlation coefficient of 0.38 between HCHO VCD and $NO_2$ VCD during noontime was better than during rush hour ($R^2$=0.06), which may be due to the contribution of vehicle emissions to HCHO precursors. A longer $NO_2$ lifetime with less dispersion efficiency in winter and HCHO from continuously generated photo-oxidation contributed to the higher correlation between HCHO VCD and $NO_2$ VCD at noon higher than during rush hour. The transport of $NO_2$ and VOC may constitute one of the causes. The VOCs from transport generate HCHO due to strong photo-oxidation at noon.

P12, l124: What is the assumed temp in the model for Dec 1, 2014 then?

**Response**: Thank you for your asking. We download the temp data of model at 2 meter and compare with the temp from in-situ instrument. The results show that the temp in the model also plummeted in December 1, 2014, and fell below 0 ℃.

[Figure]

**Figure: Hourly averaged temperature in CAMS model (grid of 0.125 °× 0.125 °and 0.25 °× 0.25 °)at 2 metre and in-situ observations at 8:00 (a) and 14:00 LT (c) from October 29 to December 31, 2014.**

P13, l16-20: Briefly state what associated errors clouds could pose. In l19 you say a slight variety (variation), give an error estimate please.

**Response**: Thank you for your suggestion. It is added in the paper.

**Changes in manuscript**: First, clouds can affect atmospheric radiative transport and thus influence optical paths. Furthermore, the atmospheric absorber densities [by (photo-)chemistry or convective transport] are potentially altered due to the changes in optical paths (Grats, ea et al. 2016). Second, AMFs calculated by geometrical approximation could be significantly biased from the reality under cloudy conditions (Brinksma, et al., 2008).

REF: Gratsea, M., Vrekoussis, M., Richter, A., Wittrock, F., Schonhardt, Anja., Burrows, J., Kazadzis, S., Mihalopoulos, N., Gerasopoulos, E.: Slant column MAX-DOAS measurements of nitrogen dioxide,

formaldehyde, glyoxal and oxygen dimer in the urban environment of

Athens, Atmos. Environ., 135,118-131,2016.

P14, l16: Where does this number come from and what dates? It is not the same as

Figure 12 and it is not mentioned anywhere else in your paper. Is it a typo? Please

advise.

**Response**: Thank you for your remind. There is mistake. This number (0.87) is the correlation coefficient $R^2$, and the 0.934 in figure 12 is the correlation coefficient R. We redraw the figure 12 by using all the data from October to December, 2014. And the figure 12 is changed to figure 14 due to some new figures were added. The new correlation coefficient $R^2$ of average HCHO VCDs with $O_3$ is 0.73. It is changed in the paper.

**Changes in manuscript:** A good correlation coefficient $R^2$ of 0.73 was found between HCHO VCD and $O_3$ during the noontime, which indicates that the main source of HCHO was from secondary photo-oxidation formation at noon.

P14,l20: Why the range? Grid sizes, I assume?

**Response**: Thank you for your question. The range is mainly due to two different time periods(8:00 and 14:00 LT) and three different grid points. In order to make it clear, I calculate the averaged value and change it in the paper.

**Changes in manuscript:** The CAMS model underestimated HCHO VCD by about $1.63 \times 10^{15}$ molec $cm^{-2}$ on average compared to the MAX-DOAS measurements.

**Response to Report#1**

P2, L1 {suggest 38% and 31%, as I don't think 4 significant figures is appropriate}

**Response:** Thank you very much for your suggestion. We considered the errors attributed to them and the correct number of significant figures. It was changed as ~38%±20%, and ~30%±24%.

**Changes in manuscript:** During the period of the APEC conference, the average HCHO VCDs were ~38%±20%, and ~30%±24% lower than that during the pre-APEC and post-APEC periods calculated at 95% confidence limit, respectively.

P2, L15: …emission reduction measurements {is it measurements or strategies or maybe measurement strategies? I would use emission reduction strategies in accordance…}

**Response:** Thank you very much for your advice. We used emission reduction strategies here.

**Changes in manuscript:** Some provinces, including Beijing, Tianjin, Hebei, Shanxi, Inner Mongolia, and Shandong implemented different emission reduction strategies in accordance with the air quality assurance plan (Wang et al., 2016a).

P3, L4-L7: {So the O3 increased by 101%, you could also say that the O3 level approximately doubled over this period then, right?}

**Response:** Thank you for your question. The O3 increased by 101%, which means the O3 level approximately doubled over this period then the same period over the last five years. It was changed in the manuscript.

**Changes in manuscript:** Wang et al, (2016a) selected five representative *in situ* stations in different locations in Beijing and found that average concentrations of $SO_2$, $NO_2$, $PM_{10}$, and $PM_{2.5}$ decreased by

62%, 41%, etc, respectively, whereas the average $O_3$ level approximately doubled over this period than the same period over the last five years ($PM_{2.5}$ since 2013)

L9 : {do you mean that there is less O3 titration due to NOx, if so re-word this? Not sure if inhibition of O3 is the best wording here?}

**Response:** Thank you for your advice. We re-worded this to make it clear.

**Changes in manuscript:** $O_3$ production rate depends on the ratios of volatile organic carbon (VOC) and NOx. The urban and suburban areas of Beijing are controlled by the NOx saturated condition of $O_3$ production. Since emission control measures are mainly focus on NOx, but not VOCs, decrease of NOx can cause significant increases of $O_3$ concentration (Wang et al., 2016a).

L10-12: {sorry but I am not sure what the difference is between traffic, urban, suburban and regional stations, define it or at least reference it please}

**Response:** Thank you for your question. At the beginning we introduced five representative in suit station, which were City Background Station, Regional Station, Suburban Station, City Station, and Transport Station. We defined it to make it clear.

**Changes in manuscript:** Wang et al, (2016a) selected five representative *in situ* stations in different locations in Beijing, which were Miyun Reservoir Station (City Background Station), Yuzhan Station (Regional Station), Changping Station (Suburban Station), Olympic Sports Center Station (City Station) and Xizhimen North Street Station (Transport Station), and found that average concentrations of $SO_2$, $NO_2$, $PM_{10}$, and $PM_{2.5}$ decreased by 62%, 41%, etc, respectively, whereas the average $O_3$ level approximately doubled over this period than the same period over the last five years ($PM_{2.5}$ since 2013).

Although the traffic and urban stations produce a lot of pollution due to motor vehicle emissions, the $NO_2$ concentrations of suburban and regional stations significantly dropped (47%) compared with the traffic and urban stations (23%) as a result of the control measures.

L22-23: .. experiencing significant descreases of 47% and 68% versus what? {…versus the average 2008 values, state what you mean here}

**Response:** Thank you for your question. It is comparison between before, during and after APEC. It was added in the manuscript to make it clear.

**Changes in manuscript:** Particles with larger sizes were better controlled during the APEC period, with the number concentration of accumulation mode and coarse mode particles experiencing more significant decreases of 47% and 68% than before and after the APEC period (Zhang et al., 2017).

P8, L5-L6: {define what large RMS values are, what was your threshold please?}

**Response:** Thank you very much for your advice. The threshold $>10^{-2}$ was added to make it clear..

**Changes in manuscript:** Data with a large root mean square (RMS) of the residuals ($>10^{-2}$) and large relative intensity offset were also excluded.

P9, L7: {do your lower EAs have obstacles in their paths, if so, what is the use of them? what about using the geometric approximation for the lower EAs, that doesn't work either right? You have a lot of unused data in your data set (ie. 3, 5, 10 degrees). Perhaps later to get a VCD from RTM?}

**Response:** Thanks for the suggestion! We checked the data and found the lower EAs are not obstructed.

However the sensitivity height at a lower EA is too close to the surface. In order to derive the HCHO VCD in the boundary layer, the geometric light path at 15 ° is a good approximation in the boundary layer. In a following study, we will try to retrieve HCHO profiles from the full sequence for the comparisons with model simulations.

**Changes in manuscript:** This study used the geometric approximation method to determine HCHO VCDs at an elevation angle of 15 °. The geometric light paths at 15 ° and 30 ° are good approximations in the boundary layer. However lower systematic errors were achieved at 15 ° than at 30 ° by using the geometrical approximation (discussed in Section 2.3 below).

L15: {what is TG?}

**Response:** Thank you very much for your question. The TGs here means trace gases. It was changed in the manuscript.

**Changes in manuscript:** The systematic error of the HCHO VCDs calculated by the geometric approximation depends on the layer height of the trace gases and aerosols.

P9-10, l23-l1 : {from your figure 6, it appears that the VCDgeo > VCDamf for 30 degrees, but VCDamf>VCDgeo at 15 degrees, is there any explanation for this?}

**Response:** Thanks for the asking! The effect shown in Fig. 6 is from RTM simulations and consistent with the previous researches based on RTM simulations (Wang et al., 2017c, Ma et al., 2013, and Shaiganfar et al., Atmos. Chem. Phys., 11, 10871–10887, 2011).

This phenomenon is mainly due to the layer height of HCHO and aerosol, and relative azimuth angles.

L6: {any explanation for the large range in the fitting error?}

**Response:** Thanks for the question. The absolute fit error is almost constant value of ($1.2*10^{15}$ molecules cm$^{-2}$) at different HCHO dSCDs. The relative error is given here. The relative error also depends on the HCHO dSCDs. The large variation range of the relative error is due to the variation of HCHO dSCD. In order to clarify this point, we add the absolute fit error in the manuscript.

**Changes in manuscript:** and the hourly average of the HCHO VCD fitting error was from 4% to 27% for the entire period with the absolute fit error is ~$1.2 \times 10^{15}$.

L12: {if the total error is 12% on average, what is the range?}.

**Response:** Thank you very much for your suggestion. The range of the total error is 3%-21%. It was added in the manuscript

**Changes in manuscript:** Since the three errors are mainly independent, the total error can be calculated by combining all the above error sources, adding up to about 7% - 28% with 17% on average.

L16: {you defined what CAMS stands for in the intro, but I would also repeat it here since it has been 9 pages since you introduced it}

**Response:** Thank you very much for your suggestion. It was added in the manuscript

**Changes in manuscript:** Copernicus Atmosphere Monitoring Service (CAMS), which is managed by ECMWF, publicly provides generally reliable atmospheric information.

L23 : TM5 chemical transport model (CTM)?

**Response:** Thank you very much for your advice. It was added in the manuscript to make it clear.

**Changes in manuscript:** In the simulation of HCHO, chemistry originating from the Transport Model 5 (TM5) had been fully integrated into the C-IFS, in which only gas phase reactions of HCHO are included.

L21: {it looks like 3 peaks to me?}

**Response:** Thank you for your question. The increasing process of HCHO VCDs is in the periods of 3-5 and 6-9 November. And the peaks appeared on November 4, 2014 and November 7, 2014, in the two periods, respectively.

P13, L5: … correspond to a minimum in the average HCHO VCDs (6.64 * 1015 molec cm-2).

**Response:** Thank you very much for your advice. We considered the correct number of significant figures. It was changed as $6.6 \times 10^{15}$ molec cm$^{-2}$.

**Changes in manuscript:** In contrast, the northeast and north directions correspond to a minimum in the average HCHO VCDs ($6.6 \times 10^{15}$ molec cm$^{-2}$).

L6-7: {what about VOCs from natural sources?}

**Response:** Thank you for your advice. The nature sources of VOC should be much lower than the anthropogenic sources, especially in the winter season. We clarify the point in the manuscript as" the nature sources of VOC in the north should be much lower than the anthropogenic sources in the winter season. "

**Changes in manuscript:** The northern cities are clean with low VOC emissions, and the nature sources of VOC in the north should be much lower than the anthropogenic sources in the winter season. Thus few precursors of HCHO were transported to the measurement station in the north wind.

P14, L1: {I would say 38% and 31% here, be consistent in your significant figures please.}

**Response:** Thank you very much for your suggestion. According to P2.L1, we considered the errors attributed to them and the correct number of significant figures. It was changed as ~38%$\pm$20%, and ~30%$\pm$24%

**Changes in manuscript:** The average HCHO VCDs were $9.7 \times 10^{15}$, $6.0 \times 10^{15}$, and $8.6 \times 10^{15}$ molec cm$^{-2}$ before, during, and after APEC, with fitting errors of 9.4%, 10.1%, and 9.7%, respectively. A noticeable decrease of ~38%$\pm$20%, and ~30%$\pm$24% during APEC was found compared with before and after APEC, which was calculated at 95% confidence limit.

L10-L12: {how do you know this, what is your proof? }

**Response:** Thank you very much for your suggestion. We considered your advice, and the references were added.

As the measurement station is located in the northern suburban area of Beijing, the effects of the control measures, which were mainly implemented in the urban areas, on HCHO were only observed at the station when dominant southerly winds occurred (Fan et al., 2016, Li et al., 2015).
Li, W.T., G, Q.X., Liu, J.R., Li, L., Gao, W.K., Su, B.D.: Comparative Analysis on the Improvement of Air Quality in Beijing During APEC, 36(12), 4340-4347, 2015.

L19-L20: the O3 is surface but the NO2 is VCD, is this a potential source of error? If so, mention it somewhere.

**Response:** Thank you very much for your advice. We clarified the point as the following:

Here it needs to clarify that the O3 data is from the surface measurements, but NO2 and HCHO are the tropospheric VCD. Since NO2 and HCHO are mostly in the boundary layer, the effect of the discrepancy of measured layers on the correlation analysis is not significant.

**Changes in manuscript:** Here it needs to clarify that the $O_3$ data is from the surface measurements, but $NO_2$ and HCHO are the tropospheric VCD. Since $NO_2$ and HCHO are mostly in the boundary layer, the effect of the discrepancy of measured layers on the correlation analysis is not significant

P19. L23: {could state the per cent error here instead of exact value, since you did that earlier, but it is your choice}

**Response:** Thank you very much for your suggestion. We have considered your advice, and the per cent error is instead of exact value. It is changed in the manuscript.

**Changes in manuscript:** The CAMS model underestimated HCHO VCD by about 24% on average compared to the MAX-DOAS measurements.

**Response to Co-editor**

I am somewhat satisfied with the changes, except for one point. I had suggested in a previous communication that an error is needed on the numbers in the abstract (and elsewhere, where it is discussed: "During the period of the APEC conference, the average HCHO VCDs were ~40% and ~30% lower than that during the pre-APEC and post-APEC periods, respectively." You have reduced these numbers from 4 to 2 significant figures, good, but certainly, some simple statistics can be put on this that would indicate whether the observed change is statistically significant or not. After all, if the reductions are -40% +/- 40% and -30 +/- 40%, then they are not truly reductions are they. Use some simple statistics that tell us how much confidence there is in the observed reductions (if not their source).

**Response:** We have carefully redo statistics and resubmit in the revised manuscript and marked every change in red.

The changing in the manuscript follows the calculation results below:

The standard error of mean is calculated by the following formula:

$$s(\bar{\delta_x}) = \sqrt{\frac{1}{n(n-1)} \sum_{i=1}^{n} (x_i - \bar{x})^2}$$

Pre-APEC:

averaged HCHO VCD, marked as a = $9.7 \times 10^{15}$ molecules cm$^{-2}$ with standard error of mean $0.8 \times 10^{15}$ molecules cm$^{-2}$

During APEC

averaged HCHO VCD, marked as b = $6.0 \times 10^{15}$ molecules cm$^{-2}$ with standard error of mean $0.5 \times 10^{15}$ molecules cm$^{-2}$

Post- APEC

averaged HCHO VCD, marked as c = $8.6 \times 10^{15}$ molecules cm$^{-2}$ with standard error of mean $0.9 \times 10^{15}$ molecules cm$^{-2}$

Since the reduction of Pre-APEC and APEC is calculated as following:

U =(a-b)/a

Averaged reduction U = (9.7-6.0)/9.7≈38%

The relative uncertainty of reduction is

$$\frac{\sigma u}{u}=\sqrt{(\frac{\sigma_{a-b}}{a-b})^2+(\frac{\sigma_a}{a})^2}=\sqrt{\frac{\sigma_a^2+\sigma_b^2}{(a-b)^2}+(\frac{\sigma_a}{a})^2}$$

$$=\sqrt{\frac{0.8^2+0.5^2}{(9.7-6.0)^2}+\frac{0.8^2}{9.7^2}}\approx 27\%$$

Then the uncertainty of reduction is

$$\sigma_u=u\times\frac{\sigma_u}{u}=38\%\times 27\%\approx 10\%$$

So the difference of Pre-APEC and APEC is $U+2\sigma_u=38\%+20\%$ calculated at 95% confidence limit.

As the same method, the averaged reduction of Post-APEC is 30%, and the Uncertainty($\sigma_u$) is 12%,

So the difference of Post-APEC and APEC is ~30%±24% calculated at 95% confidence limit..

**Changes in manuscript:**

P2L1:    During the period of the APEC conference, the average HCHO VCDs were ~38%±20%, and ~30%±24% lower than that during the pre-APEC and post-APEC periods calculated at 95% confidence limit, respectively.

P14L3-4: The average HCHO VCDs were $9.7\times10^{15}$, $6.0\times10^{15}$, and $8.6\times10^{15}$ molec cm$^{-2}$ before, during, and after APEC, with fitting errors of 9.4%, 10.1%, and 9.7%, respectively. A noticeable decrease of ~38%±20%, and ~30%±24% during APEC was found compared with before and after APEC, which was calculated at 95% confidence limit.

P19L19-20: During the period of the APEC conference, the average HCHO was $6.0\times10^{15}$ molec cm$^{-2}$, which was ~38%±20%, and ~30%±24% lower than that during the pre-APEC and post-APEC periods calculated at 95% confidence limit, respectively.

And Other changes in manuscript:

P13, L1: HCHO VCDs considerably depend on wind directions, and the average HCHO VCDs were $8\times10^{15}$ molecules cm$^{-2}$ under the southerly wind (including southwest and southeast).

P13, L4-5: In contrast, the northeast and north directions correspond to a minimum in the average HCHO VCDs ($6.6\times10^{15}$ molec cm$^{-2}$).

P14, L14-18: **Fig. 12d–f** indicate that the averaged HCHO VCDs under south winds during APEC were about $6.5\times10^{15}$ molec cm$^{-2}$, which was considerably lower than 10.3, or $9.2\times10^{15}$ molec cm$^{-2}$ in the

pre-APEC and post-APEC periods. In addition the peak values due to transport from the south urban area on November 4, 2014 and November 7, 2014 during APEC shown in **Fig. 11** were 25% and 18% lower than the peak values under similar wind fields in the pre-APEC and post-APEC periods.

P16, L16-17: On average, the CAMS model underestimated HCHO VCDs by $1.6$–$2.0 \times 10^{15}$ molec $cm^{-2}$ and $1.3$–$2.1 \times 10^{15}$ molec $cm^{-2}$ compared to the MAX-DOAS measurements at 8:00 LT and 14:00 LT, respectively, due to different grid-sizes.

Thank you for taking care of our manuscript.

Kind regards,
Xin Tian
E-mail: xtian@aiofm.ac.cn

Corresponding author : Pinhua Xie, Jin Xu, Yang Wang
E-mail address: phxie@aiofm.ac.cn; jxu@aiofm.ac.cn; y.wang@mpic.de